# TempoPFN: Synthetic Pre-training of Linear RNNs for Zero-shot Time Series Forecasting

## Abstract

Foundation models for zero-shot time series forecasting face challenges in efficient long-horizon prediction and reproducibility, with existing synthetic-only approaches underperforming on challenging benchmarks. This paper presents TempoPFN, a univariate time series foundation model based on linear Recurrent Neural Networks (RNNs) pre-trained exclusively on synthetic data. The model uses a GatedDeltaProduct architecture with state-weaving for fully parallelizable training across sequence lengths, eliminating the need for windowing or summarization techniques while maintaining robust temporal state-tracking. Our comprehensive synthetic data pipeline unifies diverse generators—including stochastic differential equations, Gaussian processes, and audio synthesis—with novel augmentations. In zero-shot evaluations on the Gift-Eval benchmark, TempoPFN achieves top-tier competitive performance, outperforming all existing synthetic-only approaches and surpassing the majority of models trained on real-world data, while being more efficient than existing baselines by leveraging fully parallelizable training and inference. We open-source our complete data generation pipeline and training code.

## 1 Introduction

Recent advances in large language models have inspired foundation models for time series forecasting that enable zero-shot predictions across diverse datasets without fine-tuning (Ansari et al., 2024; Das et al., 2024; Woo et al., 2024; Auer et al., 2025). By treating historical observations as input context, these models democratize forecasting for non-experts and excel in data-scarce domains.

However, current approaches face critical limitations. Transformer-based models struggle with long-horizon forecasting due to quadratic complexity and error accumulation (Zeng et al., 2023). While non-linear RNNs maintain temporal state, they require sequential processing that limits scalability. Although some recent models attempt synthetic-only pre-training including ForecastPFN (Dooley et al., 2023), CauKer (Xie et al., 2024), and Mamba4Cast (Bhethanabhotla & Swelam, 2024) none reported state-of-the-art performance on the Gift-Eval benchmark. TabPFN-TS (Hoo et al., 2024), which adapts a tabular foundation model to time series, achieves strong Gift-Eval performance but does not release its synthetic pre-training data, limiting reproducibility and extensibility.

We introduce **TempoPFN** (see Table 1 and Figure 1), a time series forecasting foundation model using *linear RNNs with GatedDeltaProduct recurrence* (Siems et al., 2025) for parallelizable training and inference across the sequence length. We adopt the Prior-Data Fitted Network (PFN) framework (Müller et al., 2022), treating zero-shot forecasting as Bayesian inference approximated via

Table 1: Contributions of TempoPFN: the first fully open-source time series forecasting foundation model with competitive performance; with fully synthetic pretraining and fast training and inference.

| Criterion | Tirex | TabPFN-TS | Mamba4Cast | Chronos | TempoPFN |
|---|---|---|---|---|---|
| Fully open-source data pipeline | ✗ | ✗ | ✓ | ✗ | ✓ |
| Open-source training code | ✗ | ✗ | ✓ | ✓ | ✓ |
| Competitive with SOTA performance | ✓ | ✓ | ✗ | ✓ | ✓ |
| Fast training and inference | ✓ | ✗ | (✓) | (✓) | ✓ |
| Purely synthetic pretraining | ✗ | (✓) | ✓ | ✗ | ✓ |

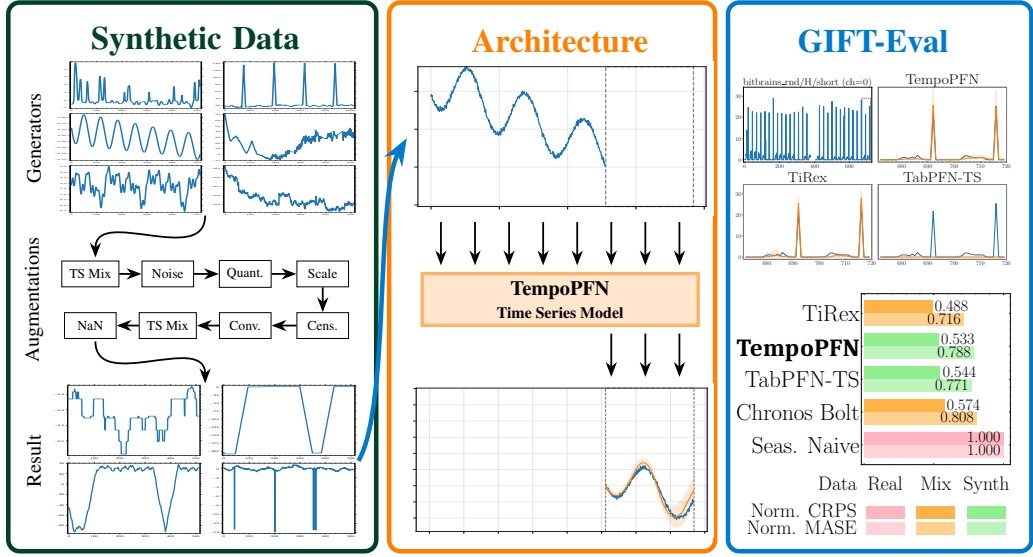

Figure 1: (Left) Synthetic Data Generation pipeline containing a mix of novel and existing time-series generators are augmented with a diverse set of augmentations to produce the time-series used for training. (Middle) The TempoPFN architecture produces coherent quantile predictions for all future time-stamps in parallel. (Right) TempoPFN obtains competitive performance on Gift-Eval despite being trained only on synthetic time-series.

in-context learning on a diverse synthetic prior (see Appendix A.1 for a detailed background). Unlike TiRex (Auer et al., 2025) which argued that non-linear RNNs like sLSTM are necessary for time-series forecasting due to their state-tracking capabilities we find that linear RNNs based on the GatedDeltaProduct recurrence are sufficient, in line with recent research demonstrating how linear RNNs can perform state-tracking (Grazzi et al., 2025). As detailed in Appendix B.1, DeltaProduct applies orthogonal rotations via multiple online gradient steps, enabling superior state-tracking compared to diagonal SSMs. Our synthetic data pipeline unifies diverse generators with novel augmentations, ensuring exclusive synthetic pre-training to prevent benchmark leakage. Unlike TabPFN-TS, we open-source our complete data generation pipeline and training code as a basis for future research (available at `https://anonymous.4open.science/r/TempoPFN_ICLR2026-E216/README.md`). In summary, our contributions are:

- The **TempoPFN** architecture, to our knowledge, the first univariate time series foundation model based on *linear RNNs with GatedDeltaProduct recurrence*. Our architecture and input representation allows the prediction of all future time-stamps in parallel, producing coherent quantile forecasts, without patching or windowing heuristics. We further propose a state-weaving mechanism for linear RNNs that facilitates bidirectional information flow across horizons without overhead.

- We design a **synthetic data pipeline** combining existing and novel synthetic generators with a cascade of augmentations, ensuring diverse temporal structures without relying on real-world data, thereby eliminating benchmark leakage and mitigating privacy concerns associated with training on real-world data. We release a fully open-source synthetic data generation pipeline for time series forecasting that achieves competitive competitive performance on Gift-Eval.

- Compared to nonlinear RNNs and transformer time series foundation models, TempoPFN achieves **top-tier competitive zero-shot performance on Gift-Eval**, surpassing all other synthetic-only approaches and the vast majority of models trained on real-world data. This result is achieved without any non-linearity in the recurrence, demonstrating that linear RNNs as a scalable and powerful alternative to non-linear RNNs and transformers for time series foundation models.

## 2 BACKGROUND AND RELATED WORK

**Time Series Forecasting.** Time series forecasting aims to predict future values $y_{T+1:T+H}$ from historical observations $y_{1:T}$. Traditional methods such as ARIMA (Box & Jenkins, 1968) and ex-

ponential smoothing (Hyndman et al., 2008) typically produce point estimates, while probabilistic forecasting captures uncertainty by modeling the predictive distribution $p(y_{T+1:T+h} \mid y_{1:T})$. The advent of deep learning has significantly expanded this toolkit, introducing transformers (Vaswani et al., 2017) and modern recurrent architectures (Beck et al., 2024; Gu & Dao, 2023).

A major recent development is *zero-shot forecasting*, where models pre-trained on diverse corpora can directly predict unseen time series without requiring fine-tuning. This paradigm mirrors the transformative shift seen in natural language processing and computer vision, where foundation models enable efficient cross-domain adaptation without costly per-task training.

Most successful zero-shot approaches leverage transformer architectures. Chronos (Ansari et al., 2024), TimesFM (Das et al., 2024), and MOIRAI (Woo et al., 2024) use techniques such as patching, frequency-specific projections, and masked modeling to handle heterogeneous time series data. Building on this foundation, MOIRAI-MOE (Liu et al., 2025b) incorporates a sparse mixture-of-experts architecture to achieve token-level specialization and improved robustness. Among true zero-shot models, MOIRAI currently demonstrates state-of-the-art performance on Gift-Eval while carefully avoiding overlap with evaluation benchmarks.

An alternative approach focuses on synthetic data pretraining. ForecastPFN (Dooley et al., 2023) trains exclusively on synthetic distributions featuring multi-scale trends, seasonality, and Weibull noise, enabling Bayesian zero-shot inference through single forward passes. TimePFN (Taga et al., 2025) extends this framework to multivariate scenarios using Gaussian process kernels and linear coregionalization. Similarly, TabPFN-TS (Hoo et al., 2024) represents time series in tabular format and leverages TabPFNv2 (Hollmann et al., 2025), achieving competitive performance despite the limited availability of its underlying synthetic data.

Recent work has also revisited recurrent architectures for long-horizon forecasting. TiRex (Auer et al., 2025), currently the top performer on Gift-Eval, uses xLSTM (Beck et al., 2024) pre-trained on synthetic Gaussian processes, Chronos datasets, and carefully selected Gift-Eval subsets, enhanced with data augmentation techniques including amplitude modulation, censoring, and spike injection. In contrast, TempoPFN takes a different approach by exploiting linear RNNs with Gated-DeltaProduct mechanisms and negative eigenvalues, enabling fully parallelizable training without requiring patching or summarization while relying exclusively on synthetic pretraining data to eliminate real-world leakage concerns.

**Linear RNNs.** Recurrent neural networks have seen a resurgence in interest with the emergence of linear RNNs also known as state-space models. While non-linear RNNs are non-trivially parallelizable (Gonzalez et al., 2024), linear RNNs can be parallelized using a chunk-wise parallel form (Yang et al., 2024a) or an associative scan (Gu & Dao, 2023; Martin & Cundy, 2018). Formally, linear RNN layers transform input sequences $\boldsymbol{x}_{1:t} \in \mathbb{R}^l$ into outputs $\hat{\boldsymbol{y}}_{1:t} \in \mathbb{R}^p$ through the recurrence

$$\boldsymbol{H}_i = \boldsymbol{A}(\boldsymbol{x}_i)\boldsymbol{H}_{i-1} + \boldsymbol{B}(\boldsymbol{x}_i) \text{ with output } \hat{\boldsymbol{y}}_i = \text{dec}(\boldsymbol{H}_i, \boldsymbol{x}_i) \text{ for } i \in \{1, \ldots, t\} \tag{1}$$

where $\boldsymbol{A} : \mathbb{R}^l \to \mathbb{R}^{n \times n}$ parameterizes the state-transition matrix, $\boldsymbol{B} : \mathbb{R}^l \to \mathbb{R}^{n \times d}$ governs state inputs, and $\text{dec} : \mathbb{R}^{n \times d} \times \mathbb{R}^l \to \mathbb{R}^p$ produces the layer output. Variants such as Mamba (Dao & Gu, 2024), GLA (Yang et al., 2024a), and mLSTM (Beck et al., 2024) adopt diagonal transitions, while others explore richer parameterizations. More expressive formulations relax diagonal state-transition constraints, as seen in DeltaNet (Schlag et al., 2021; Irie et al., 2023; Yang et al., 2024b), TTT-Linear (Sun et al., 2024), RWKV-7 (Peng et al., 2025), B'MOJO (Zancato et al., 2024), and Titans (Behrouz et al., 2024).

Within this framework, we use **DeltaProduct** (Siems et al., 2025) as our token-mixing mechanism, which generalizes DeltaNet's non-diagonal transitions by expressing $\boldsymbol{A}(\boldsymbol{x}_i)$ as a product of $n_h$ generalized Householder transformations, enabling a rank-$n_h$ transformation of the matrix-valued hidden state $\boldsymbol{A}(\boldsymbol{x}_i) = \prod_{j=1}^{n_h} \left(\boldsymbol{I} - \beta_{i,j} \boldsymbol{k}_{i,j}\boldsymbol{k}_{i,j}^{\top}\right)$. For each token $\boldsymbol{x}_i$, the model generates $n_h$ normalized keys $\boldsymbol{k}_{i,j} = \psi(\boldsymbol{W}_j\boldsymbol{x}_i)/\|\psi(\boldsymbol{W}_j\boldsymbol{x}_i)\|_2$, values $\boldsymbol{v}_{i,j} = \boldsymbol{V}_j\boldsymbol{x}_i$, and coefficients $\beta_{i,j} = \phi(\boldsymbol{U}_j\boldsymbol{x}_i)$ using learnable matrices $\boldsymbol{W}_j, \boldsymbol{V}_j, \boldsymbol{U}_j$, SiLU activation $\psi$ (Hendrycks & Gimpel, 2016), and a sigmoid-based gating function $\phi$. Siems et al. (2025) found that increasing $n_h$ leads to significantly improved length-extrapolation, language modeling, and state-tracking on permutation tasks, all capabilities which are equally desirable for time-series forecasting.

## 3 TEMPOPFN

### 3.1 ARCHITECTURE

The TempoPFN architecture is designed to forecast univariate time series across a full prediction horizon in a single forward pass, as illustrated in Figure 2. It consists of four main stages: input representation, backbone, non-causality through state weaving, and prediction.

**Input representation.** TempoPFN uses an input representation in which history (timesteps + values) and future (timesteps) are concatenated into one token sequence enabling communication between future time-steps for coherent predictions. In contrast to TiReX, which presummarizes time-steps into windows of size 32, TempoPFN directly operates on the individual time-steps. Each time step $t_i$ is encoded using GluonTS (Alexandrov et al., 2020) time features (e.g., seasonality indicators, day-of-week, or index-based encodings) that are linearly projected into the embedding dimension of the model. For historical steps, observed values $y_i$ are projected via a linear layer, while missing values are handled by a learnable NaN embedding. The historical embedding is obtained by additively combining the value and the time-feature embeddings. For future time steps, only the time-feature embedding is used.

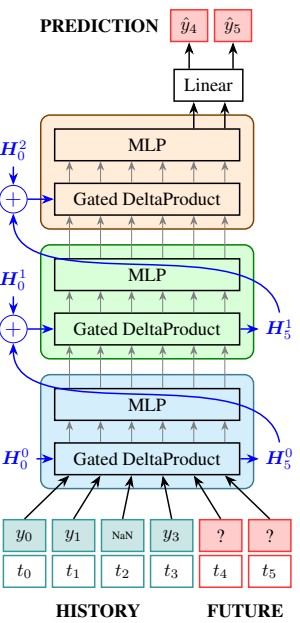

Figure 2: The **TempoPFN** architecture (3 blocks), using stacked GatedDeltaProduct blocks, learnable initial states $\boldsymbol{H}_0^i$ and state-weaving.

**Backbone.** The core of TempoPFN is a stack of 10 encoder layers, each based on the *Gated DeltaProduct block* from the flash-linear-attention library (Yang & Zhang, 2024), originally derived from the LLaMA architecture (Touvron et al., 2023). Each block consists of three components: (1) *token mixing* through a Gated DeltaProduct recurrence with short one-dimensional convolutions (kernel size 16–32), (2) *pre-normalization* applied before the recurrent unit to stabilize training, and (3) a gated MLP for channel-wise feature transformation. This design combines the parallelization advantages of linear recurrences with the expressivity of lightweight convolutional and feedforward operations. *Non-causality via state weaving.* Whereas DeltaProduct was originally developed for autoregressive language modeling, forecasting across a full horizon does not require causal masking. To exploit this property, we introduce *state weaving*. Specifically, the final hidden state of each layer $\boldsymbol{H}_t^i$ is added to the learnable initial state of the next layer $\boldsymbol{H}_0^{i+1}$. This mechanism enables bidirectional information flow across the entire sequence length without additional parameters or computational overhead through explicit bidirectionality (Hwang et al., 2024; Afzal et al., 2025). This allows future time-steps to attend to the entire history and future context, preventing the information bottleneck typical of causal RNNs during the prediction phase.

**Prediction.** At the output stage, embeddings corresponding to the forecast horizon are extracted from the final encoder block. These embeddings are passed through a linear projection head that outputs multiple *quantiles* of the predictive distribution, enabling probabilistic forecasting. Overall, this design allows to directly predict all future values given a history using a single forward pass.

### 3.2 SYNTHETIC DATA GENERATION

To train our time series foundation model, we generated a large and diverse dataset using 10 different synthetic generators. This approach combines established data generation techniques with novel methods to capture a wide spectrum of temporal patterns and behaviors. For a more comprehensive description of each generator refer to Appendix C.

**Existing Generators.** We adapted several established generators from prior work to ensure comprehensive coverage of common temporal patterns. The **ForecastPFN** generator (Dooley et al., 2023) composes multiplicative trend and seasonality components, combining linear and exponential growth terms with sinusoidal harmonics. The generator includes Weibull-distributed noise and

augmentations such as time warping, magnitude scaling, and spike injection, with filtering mechanisms to avoid extreme values. **KernelSynth**, following Ansari et al. (2024), samples univariate time series from Gaussian process priors with composite kernels. Base kernels include periodic (ExpSineSquared), stationary (RBF, RationalQuadratic), and noise (WhiteKernel) components, combined through addition or multiplication to yield smooth yet varied trajectories. We extended this approach with a broader **Gaussian Process** generator, as in Bhethanabhotla & Swelam (2024) that randomly combines kernels with greater functional diversity, producing wider ranges of stationary and non-stationary patterns. The **CauKer** generator (Xie et al., 2024) introduces causal dependencies by sampling from structural causal models (SCMs). Each node represents a Gaussian process with composite kernels and stochastic mean functions, while edges in a random DAG apply nonlinear transformations. We generate 21-channel multivariate series, treating each channel as an independent univariate signal to capture diverse, interdependent dynamics.

**Novel Generators.** We developed several new generators to fill gaps in existing approaches and capture specific temporal behaviors. **Sawtooth** creates ramp-like patterns with upward or downward slopes, enhanced with small linear trends and low-amplitude seasonal components to avoid overly idealized signals. **Step Function** produces piecewise constant series with configurable change-points, step sizes, and drift, using Gaussian smoothing at boundaries along with added noise, seasonality, and anomalies. For anomaly-rich data, we created two specialized generators. **Anomaly** produces baseline signals with periodic or clustered spikes, varying in magnitude regimes (constant, trending, cyclical, or correlated random) and timing patterns. **Spikes** emphasizes event-driven behavior by placing sharp spikes on flat baselines, with configurable shapes (V, inverted V, or plateau variants) arranged in bursty or evenly spread patterns. The **Sine Wave** generator provides clean oscillatory patterns with configurable period, amplitude, phase, and noise, offering fundamental periodic signals for learning basic oscillatory structures. To capture highly complex, real-world dynamics, we introduce **Audio-Inspired Generators** that use procedural audio synthesis techniques implemented with Pyo (Belanger, 2016). These generators model phenomena such as **Stochastic Rhythms** for event data, **Financial Volatility** with market shocks and clustering, **Network Topology** with traffic bursts and congestion, and **Multi-Scale Fractals** for self-similar patterns. Our most sophisticated contribution is the **stochastic differential equation** (**SDE**) generator, a flexible synthetic data generator based on a regime-switching, time-inhomogeneous Ornstein–Uhlenbeck (OU) process. The OU process follows the SDE $dy_t = \theta(t, r_t)\big(\mu(t, r_t) - y_t\big)\, dt + \sigma(t, r_t)\, dW_t$ where $\theta(t, r_t)$ is the mean reversion speed, $\mu(t, r_t)$ the time-varying mean, and $\sigma(t, r_t)$ the volatility. Each parameter depends on both time $t$ and a latent regime $r_t \in \{0, 1\}$ that evolves as a Markov chain. This framework enables parameters to shift abruptly across regimes while drifting smoothly over time through polynomial, sinusoidal, logistic, or piecewise-linear trends. Seasonal patterns are injected additively into both mean and volatility components, with amplitudes subject to gradual growth or decay. For enhanced realism, we optionally replace standard Brownian motion with fractional Brownian motion, introducing long-memory dynamics through the Hurst exponent $H \in (0, 1)$. Each simulated series undergoes global rescaling and shifting before additive Gaussian measurement noise is applied. This construction produces highly diverse temporal structures, capturing regime shifts, non-stationarity, periodicity, and measurement noise within a principled stochastic framework.

## 3.3 DATA AUGMENTATIONS

In addition to diverse synthetic time-series generators, our pipeline (Figure 3) also contains a mix of existing and novel augmentations to mix, transform, and distort the time-series for greater diversity.

**Augmentation Pipeline.** The offline pipeline applies transformations in a structured sequence. Base series undergo optional normalization (80% probability) using random scalers (Robust, MinMax, Median, or Mean). Early-stage TS-Mixup (Darlow et al., 2023) creates convex combinations of multiple source series with probability $p = 0.5$. The core augmentation step samples 2-5 distinct transformation categories with weighted probabilities: Invariances (0.6), Structure (0.6), Seasonality (0.5), Signal Processing (0.4), Discrete Effects (0.6), and Measurement Artifacts (0.3). From each selected category, one specific transformation is randomly chosen and applied in fixed global order. Optional stochastic convolution filtering (probability $p = 0.3$) applies 1-3 random 1D convolutions with randomized parameters. Late-stage TS-Mixup provides additional combination opportunities, followed by finishing transformations, including minor global scaling and low-magnitude Gaussian noise injection In the following, we provide details on the augmentations we implemented.

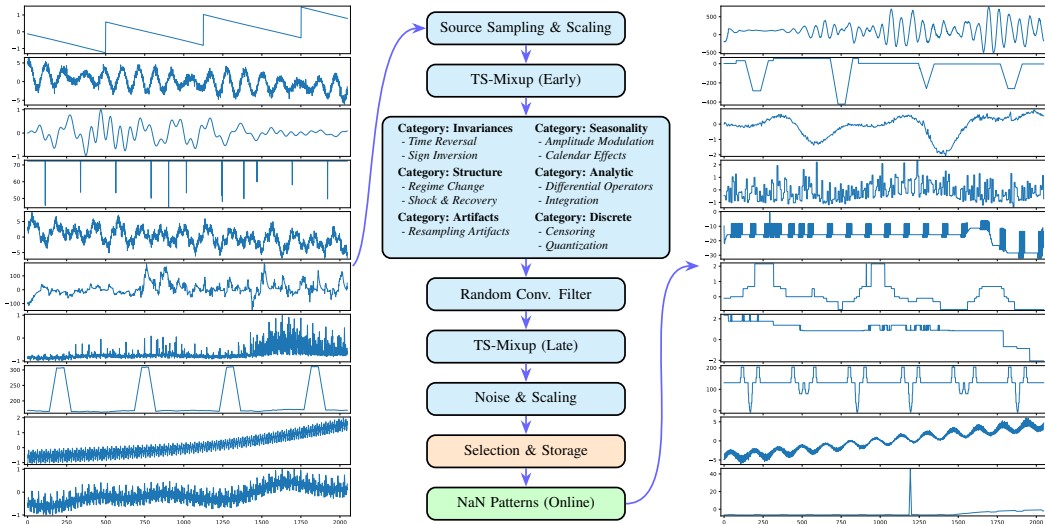

Figure 3: Synthetic data augmentation pipeline. Synthetic time-series undergo probabilistic transformations across six categorical groups (Invariances, Structure, Seasonality, Analytic, Discrete, Artifacts), with optional TS-Mixup combinations and stochastic convolution filtering. Competitive selection based on change scores ensures meaningful augmentations. Example outputs demonstrate the diversity of generated temporal patterns.

**Transformation Categories. Invariance transformations** promote robustness through temporal reversal ($\mathbf{x} \to \mathbf{x}_{T:1}$) and sign inversion ($\mathbf{x} \to -\mathbf{x}$), preserving temporal dependencies while testing directional conventions. **Structural modifications** inject non-stationarity via regime changes with piecewise affine transforms across random change-points, and shock-recovery dynamics using exponential decay impulses $I(t) = Ae^{-(t-t_0)/\tau}$ with randomized parameters.

**Seasonal effects** simulate real-world periodicities through calendar injections that apply multiplicative factors for weekend dips, month-end spikes, and holiday-like impulses using timestamp metadata. Amplitude modulation applies localized scaling to random segments, simulating time-varying volatility. **Signal processing** transformations include Gaussian smoothing followed by finite-difference operators (Sobel, Laplacian, higher-order derivatives up to 4th order) and numerical integration, with outputs rescaled to preserve original value ranges. Random convolution layers with highly randomized parameters (Dempster et al., 2020) provide additional signal transformation capabilities.

**Measurement artifacts** introduce realistic data collection imperfections: censoring clips values at random quantiles (similarly used by TiRex (Auer et al., 2025)), non-uniform quantization maps values to discrete levels using quasi-random Sobol sequences, and resampling artifacts downsample and upsample series with various interpolation methods.

**Combination strategies** We implement TS-Mixup (Ansari et al., 2024) to generate novel series through convex combinations of 2-10 source series, with mixing weights sampled from Dirichlet distributions and extend it with time-dependent mixing using smooth simplex path interpolation.

## 4 EXPERIMENTS

**Pre-training Setup.** TempoPFN's pre-training is conducted **exclusively on synthetic data**, ensuring no exposure to real-world benchmarks prior to evaluation. The training corpus consists of approximately 10 million time series from our generators, each with a maximum length of 2048. We train our main model (34.69M parameters) using a for a total of 4 million iterations with a batch size of 40. We use the AdamW optimizer (Loshchilov & Hutter, 2019) and quantile loss. We selected a 35M model (10 layers, 4 heads, 512 embedding dimension) for its strong performance and comparability to TiRex (Auer et al., 2025). To ensure robustness across sequence lengths, we randomly sample both the context length and historical window size during training. Complete training details are in Appendix D.

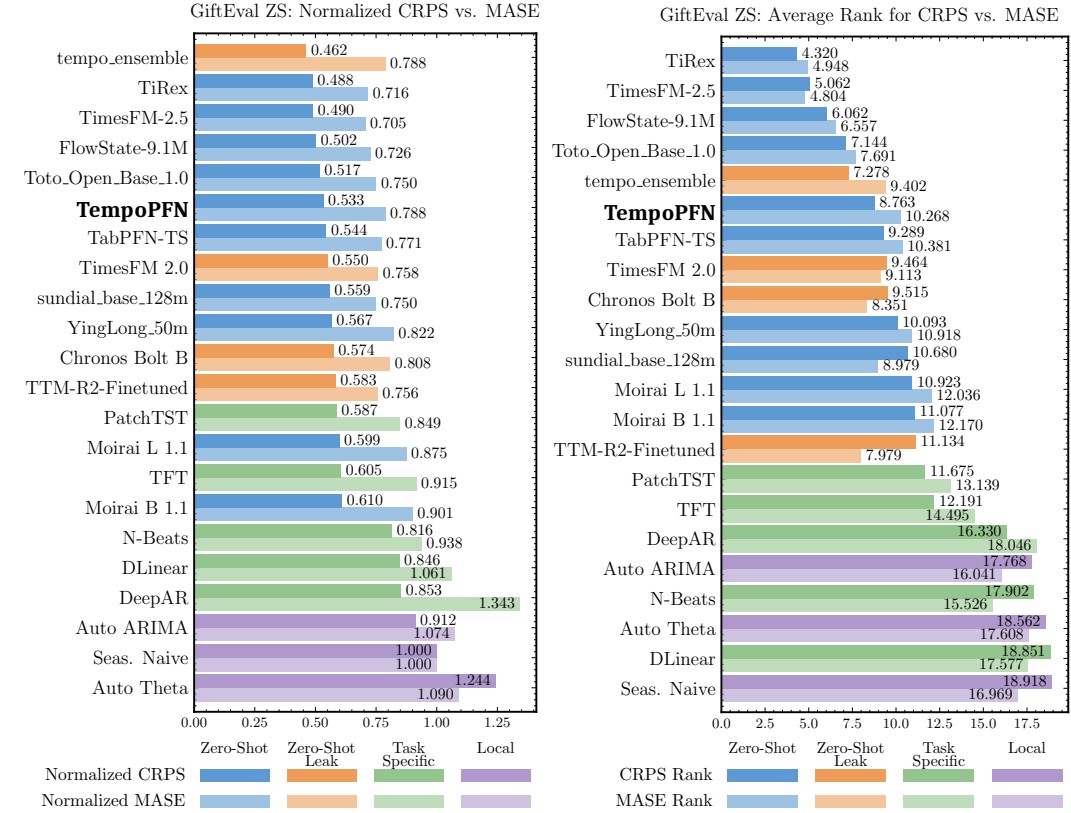

Figure 4: Comparison of TempoPFN performance (4M iterations), against other models on GiftEval benchmark. We compute both normalized and average ranks for CRPS and MASE. Colors represent the class of time series model.

**Quantitative Results.** We evaluate TempoPFN on the Gift-Eval benchmark, a comprehensive zero-shot forecasting suite covering diverse real-world datasets across domains and horizons. TempoPFN surpasses probabilistic performance of TabPFN-TS, the strongest synthetic-only baseline, with an overall CRPS of 0.537 (vs. 0.544). Its point-forecast accuracy is competitive, though slightly lower, with an overall MASE of 0.797 (vs. 0.771). Remarkably, despite relying solely on synthetic training data, TempoPFN matches or exceeds several leading models trained on real-data, including Chronos Bolt B (0.574/0.808), TimesFM 2.0 (0.550/0.758), and YingLong 50m (0.567/0.822), ranking 6th overall in CRPS and 5th in MASE. Figure 4 summarizes these quantitative comparisons against state-of-the-art baselines.

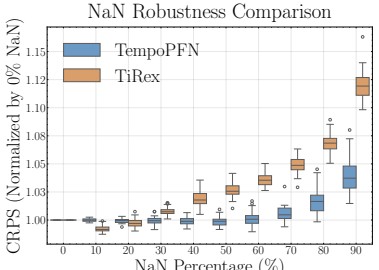

Figure 6: Normalized CRPS (relative to TempoPFN's CRPS at 0% NaNs) of TempoPFN and TiRex as a function of the percentage of missing values (NaN) in the data.

**Qualitative Results.** Figure 5 shows forecasting results on representative Gift-Eval series with varying temporal patterns (see Figure 21 for additional results). All models capture key trends and seasonality, but TempoPFN produces coherent predictive distributions without artifacts. Compared to TabPFN-TS, TempoPFN generates smoother uncertainty bounds while maintaining competitive point forecasts. This is likely a result of TabPFN-TS predicting all future time-steps in isolation while our architecture allows future time-steps to communicate. In many longer predictions made by TiRex (e.g. bizitops service), we find high frequency artifacts in the prediction of the quantiles which we hypothesize to be a result of the windowing done by TiRex which compresses the time-series into chunks of size 32 before applying the model and later upprojects them back to the original resolution. Since TempoPFN requires no windowing, we did not notice similar artifacts.

**Robustness to NaNs.** We now compare the robustness of TempoPFN and TiRex towards missing values (NaN) in the data. Figure 6 shows that both models exhibit a degradation in performance

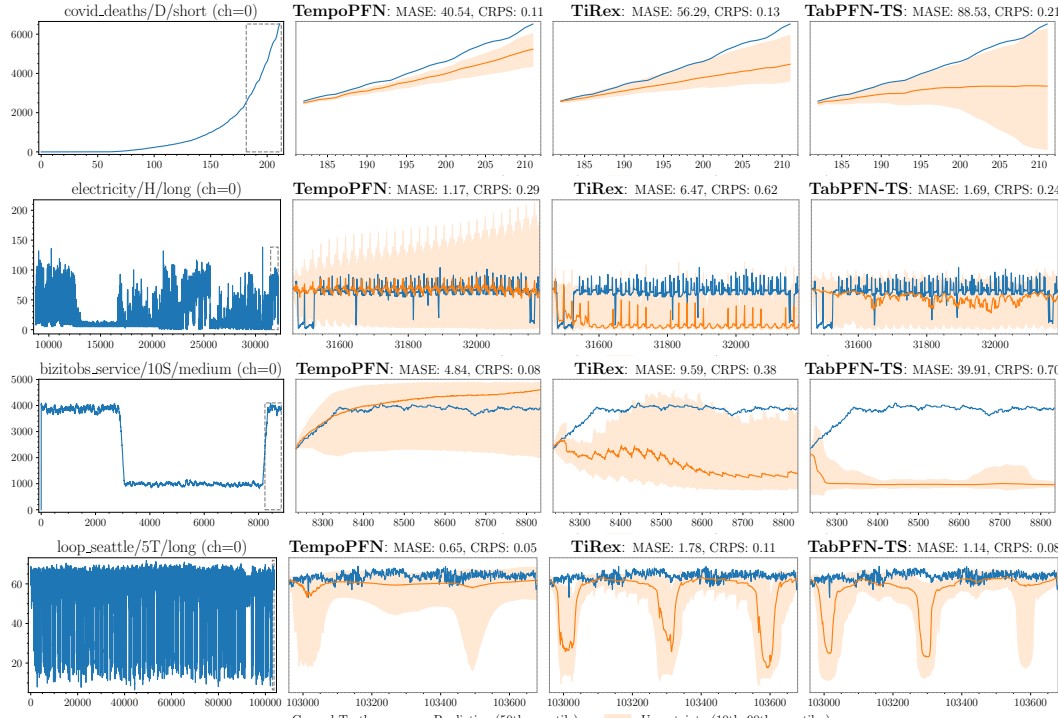

Figure 5: Qualitative comparison between TempoPFN, TiRex and TabPFN-TS on three series from the GIFT-Eval Benchmark. (Left) Total context with prediction window in dashed grey box. (Right) Predictions between TempoPFN, TiRex, and TabPFN-TS.

as the percentage of NaNs increases, however, TempoPFN is significantly more robust. While the normalized CRPS score (relative to TempoPFN's CRPS at 0% NaNs) for both models rises with more NaNs, TiRex's performance deteriorates more rapidly, with its median CRPS increasing by over 11% when 90% of the data is missing. In contrast, TempoPFN's median error increases by only about 4% under the same conditions, showcasing its superior stability and resilience when faced with incomplete time series data.

## 4.1 THE IMPORTANCE OF TEMPOPFN'S COMPONENTS

**Ablation Study Setup.** To manage the significant computational cost of pre-training, not all ablation experiments were conducted for a full 4M-iteration training schedule. Therefore, these studies are designed to provide strong directional evidence on the relative importance of each component, rather than to measure their full, converged performance impact.

**Ablating the synthetic time series generators.** To assess the individual contribution of each synthetic data source, we conducted an ablation study by retraining our model while excluding one synthetic time series generator at a time. As detailed in Table 3, the results reveal a clear hierarchy of importance, consistently observed across short, medium, and long-term forecasting horizons, with every generator proving beneficial for high performance. The highest impact data generator is our proposed SDE generator; its removal caused the most severe performance degradation, increasing the overall CRPS by 26% from 0.578 to 0.729. This highlights the importance of exposing the model to time series with mean-reverting and noisy, continuous-time dynamics. Significant, albeit smaller, performance losses were also observed upon removing generators responsible for complex seasonality (Cauker), abrupt changes (Step), and transient events (Spike), underscoring the necessity of a diverse pre-training corpus that captures a wide array of structural and stochastic patterns. In Table 7 in the appendix, we also compare our base model trained using all generators with models trained using a single generator at a time.

**Ablating the augmentation pipeline.** To quantify the impact of augmentations, we trained models with and without the full suite augmentation suite (see Table 2). Results on the GIFT-Eval bench-

Table 3: Ablation study of synthetic data priors using a leave-one-out methodology (500k iterations). The 'Ablation' column indicates the single prior excluded from the training mixture. Performance is measured by CRPS and MASE (lower is better). Rows are colored to indicate the performance impact on the overall CRPS when a prior is removed: High Impact ($> 25\%$ increase) and Medium Impact ($> 10\%$ increase). **N** = Novel prior (our contribution), **A** = Adapted from open-source.

| Ablation | Source | Gift-ZS Overall | | Gift-ZS Short | | Gift-ZS Medium | | Gift-ZS Long | |
|---|---|---|---|---|---|---|---|---|---|
| | | CRPS | MASE | CRPS | MASE | CRPS | MASE | CRPS | MASE |
| Base Model | – | 0.577 | 0.842 | 0.563 | 0.763 | 0.566 | 0.900 | 0.631 | 1.019 |
| - GP | A | 0.591 | 0.830 | 0.576 | 0.749 | 0.605 | 0.924 | 0.618 | 0.981 |
| - Kernel | A | 0.611 | 0.885 | 0.589 | 0.796 | 0.637 | 0.981 | 0.648 | 1.056 |
| - ForecastPFN | A | 0.617 | 0.885 | 0.588 | 0.791 | 0.643 | 0.981 | 0.674 | 1.075 |
| - Sawtooth | N | 0.628 | 0.900 | 0.597 | 0.800 | 0.661 | 1.012 | 0.684 | 1.091 |
| - Sinewave | N | 0.628 | 0.899 | 0.594 | 0.799 | 0.677 | 1.032 | 0.676 | 1.070 |
| - Anomaly | N | 0.630 | 0.897 | 0.592 | 0.794 | 0.684 | 1.024 | 0.683 | 1.079 |
| - Step | N | 0.640 | 0.927 | 0.605 | 0.819 | 0.686 | 1.063 | 0.689 | 1.120 |
| - Stochastic Rhythm | N | 0.642 | 0.911 | 0.601 | 0.802 | 0.701 | 1.043 | 0.699 | 1.111 |
| - Spike | N | 0.645 | 0.936 | 0.619 | 0.836 | 0.678 | 1.059 | 0.684 | 1.115 |
| - Cauker | A | 0.656 | 0.928 | 0.605 | 0.810 | 0.728 | 1.084 | 0.729 | 1.132 |
| - SDE (OU Process) | N | **0.729** | **1.031** | **0.684** | **0.916** | **0.799** | **1.184** | **0.789** | **1.225** |
| seasonal_naive | – | 1.000 | 1.000 | 1.000 | 1.000 | 1.000 | 1.000 | 1.000 | 1.000 |

mark reveal consistent gains from our augmentation pipeline yielding a 5.4% relative improvement in overall CRPS and a 3.8% improvement in overall MASE. These gains are present across all forecasting horizons, underscoring the benefit of our complex data augmentation pipeline for extrapolation to real-world data.

**Ablating architectural components.** Results on architectural ablations are provided in Table 9 in the Appendix. These results highlight the importance of our proposed 'weaving' mechanism. Specifically, Table 9 shows that disabling 'weaving' in our main model (d=512, L=10) leads to a performance degradation, increasing the overall CRPS from 0.533 to 0.537. This result supports the hypothesis that enabling bidirectional information flow across layers is beneficial.

Table 2: The impact of data augmentation on model performance, shown by normalized performance values (500k iterations). Individual augmentation effects and probability tuning were not exhaustively explored due to resource constraints.

| Model | Gift-ZS Overall | | Gift-ZS Short | | Gift-ZS Medium | | Gift-ZS Long | |
|---|---|---|---|---|---|---|---|---|
| | CRPS | MASE | CRPS | MASE | CRPS | MASE | CRPS | MASE |
| w/ Aug | **0.577** | **0.842** | **0.563** | **0.763** | **0.566** | **0.900** | **0.631** | **1.019** |
| w/o Aug | 0.610 | 0.875 | 0.582 | 0.783 | 0.617 | 0.963 | 0.643 | 1.059 |
| seasonal_naive | 1.000 | 1.000 | 1.000 | 1.000 | 1.000 | 1.000 | 1.000 | 1.000 |

## 5 CONCLUSION AND FUTURE WORK

We introduce *TempoPFN*, a novel time series foundation model demonstrating that linear RNNs, specifically the *GatedDeltaProduct* architecture, provide a highly efficient and scalable solution for zero-shot forecasting. By enabling parallelizable training, our model processes long sequences without patching or summarization heuristics. TempoPFN is trained exclusively on our open-source synthetic data generation pipeline, which integrates diverse generators and a complex augmentation framework. This synthetic-only approach ensures full reproducibility and eliminates data leakage concerns. On the Gift-Eval benchmark, TempoPFN achieves top-tier competitive performance, surpassing other synthetic-only approaches and the vast majority of models trained on real-world data, establishing linear RNNs as a powerful and scalable alternative to prevailing architectures.

A key limitation of our current work is its focus on univariate time series. Extending our synthetic generation pipeline and state-weaving architecture to the more complex multivariate case represents a primary direction for future work. Additionally, incorporating pre-training on diverse real-world time series datasets could further enhance forecasting accuracy and generalization. Finally, investigating the performance of Linear RNN architectures against Transformer-based models for zero-shot forecasting represents a potential direction for further research.

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

# A  THEORETICAL FRAMEWORK AND DESIGN PRINCIPLES

## A.1  BACKGROUND: PRIOR DATA FITTED NETWORKS (PFNS)

Prior Data Fitted Networks (PFNs) (Müller et al., 2022) represent a paradigm shift in machine learning, moving from learning a single fixed task to learning a *universal inference algorithm*. In this section, we provide a brief overview of the PFN framework to contextualize the methodology used in TempoPFN.

**Definition and Objective.**  A PFN is a neural network $\phi$, with parameters $\theta$, trained to approximate the posterior predictive distribution (PPD) induced by a prior distribution $P(\mathcal{D})$ over datasets. Formally, let a dataset $\mathcal{D} = \{(x_i, y_i)\}_{i=1}^{N}$ be drawn from a prior $P$. The goal of a PFN is to minimize the Kullback-Leibler divergence (or equivalently, the cross-entropy loss) between its output and the true posterior predictive distribution of the prior:

$$\min_{\theta} \mathbb{E}_{\mathcal{D} \sim P} \left[ \sum_{i=1}^{N} -\log p_{\theta}(y_i \mid x_i, \mathcal{D}_{1:i-1}) \right] \tag{2}$$

where $\mathcal{D}_{1:i-1}$ represents the context (history) observed so far.

**In-Context Learning as Bayesian Inference.**  Unlike traditional supervised learning, where the model's weights $\theta$ encode the solution to a specific task (e.g., "predict sales for Company X"), a PFN's weights encode the *algorithm* for solving a class of tasks defined by the prior. At inference time, the PFN performs *In-Context Learning* (ICL): it takes a small dataset of context observations (the history of a time series) and produces predictions for new query points (the future) in a single forward pass. Crucially, this forward pass acts as a fast approximation of Bayesian Inference without the computational cost of MCMC or variational methods (Müller et al., 2022; Hollmann et al., 2023).

**The Role of Synthetic Data.**  The performance of a PFN is fundamentally bounded by the quality and diversity of its prior $P$. Since real-world data is often limited, biased, or private, PFNs are typically trained on vast repositories of *synthetic data* generated from procedural priors. For example, TabPFN (Hollmann et al., 2023) uses Structural Causal Models (SCMs) to generate tabular data, while ForecastPFN (Dooley et al., 2023) uses a mix of trends and seasonalities. Similarly, in methodname the "Prior" is the novel synthetic data pipeline detailed in Appendix C, which generates diverse temporal dynamics using SDEs, GPs, and asymmetric waveforms. Our "Network" is a Linear RNN, namely GatedDeltaProduct (Siems et al., 2025), chosen for its efficiency in handling long sequential contexts compared to the Transformers used in previous PFNs. *By training the sequence model on this synthetic prior, TempoPFN learns to infer the underlying generative process of any unseen time series given its history, enabling zero-shot probabilistic forecasting.*

## A.2  DESIGN PRINCIPLES FOR SYNTHETIC DATA GENERATION

Our selection of synthetic data generators is grounded in the principle of *structural decomposition*. We posit that the manifold of real-world time series can be spanned by four fundamental dynamical properties: *Smoothness*, *Stochastic Volatility* (Roughness), *Temporal Asymmetry*, and *Discontinuities*. Existing synthetic pipelines often over-index on the first (trends/seasonality) (Dooley et al., 2023) while neglecting the latter three. We designed a principled portfolio of generators to act as orthogonal "basis functions" for these properties, ensuring our prior distribution covers the complex dynamics found in downstream tasks.

**Smooth Dynamics (Gaussian Processes).** To capture non-parametric trends and local correlations, we employ Gaussian Processes (GPs). Real-world data is dominated by latent trends that evolve smoothly but unpredictably, such as demographic shifts or climate warming, which cannot be captured by rigid linear or polynomial functions. GPs with RBF or Matérn kernels serve as the standard for modeling such smooth, differentiable manifolds, providing the model with a robust prior for interpolation and extrapolation of continuous trends.

**Stochastic Volatility and Roughness (SDEs).** A critical deficiency in standard synthetic pipelines is the reliance on additive Gaussian noise ($\epsilon \sim \mathcal{N}(0, 1)$), which implies homoscedasticity (constant variance). However, financial and physical systems are inherently *heteroscedastic*, exhibiting

state-dependent volatility and mean-reverting dynamics. To fill this gap, we integrate Stochastic Differential Equations (SDEs), specifically Regime-Switching Ornstein-Uhlenbeck (OU) processes. By explicitly modeling the diffusion term $\sigma(t, y_t)$, we force the model to learn to distinguish between deterministic signal drift and stochastic volatility clustering, a capability essential for accurate uncertainty quantification.

**Asymmetric Periodicity (Sawtooth Waveforms).** Standard sinusoidal generators rely on an assumption of *time-reversal symmetry* (equal rise and fall times). Yet, many physical and economic processes are inherently asymmetric and irreversible: inventory levels deplete gradually and restock instantaneously; capacitors discharge rapidly. We selected the *Sawtooth* wave as the fundamental primitive for asymmetry. Unlike Triangle waves (symmetric) or Square waves (step functions), the Sawtooth explicitly models the gradual-rise/sharp-drop dynamic.

**Discontinuities and Structural Breaks (Spikes/Steps).** Finally, real-world data is rife with instantaneous regime changes: policy shifts, sensor failures, or sudden shocks. All of these violate the smoothness assumptions of GPs and SDEs. To model these structural breaks, we include explicit **Step** and **Spike** generators. Including these non-differentiable primitives ensures the model remains robust to covariate shifts and prevents the "smearing" of distinct regimes into a single average.

By composing these four distinct dynamical behaviors, TempoPFN has access to a *complex prior*, allowing it to generalize zero-shot to unseen time series by identifying the governing combination of dynamics (related to the state-tracking capabilities of GatedDeltaProduct too), rather than memorizing dataset-specific statistics.

### A.3 COMPARISON WITH EXISTING SYNTHETIC STRATEGIES

Freq-Synth (Nochumsohn et al., 2024) and TabPFN-TS (Hoo et al., 2024) are two recent methods that employ only synthetic data training as well. However, their methodologies represent fundamentally different paradigms in synthetic data generation and usage compared to TempoPFN. Freq-Synth adopts a task-specific generation strategy, hence requiring prior knowledge of the target dataset's sampling rate to generate a custom training corpus of harmonic signals (sums of sinusoids) tailored to mitigate data scarcity for that specific task. In contrast, in TempoPFN, we pretrain a single model on a fixed, comprehensive corpus designed to marginalize over diverse temporal dynamics without requiring task-specific data generation. This distinction places TempoPFN in a unique position within the broader landscape of synthetic time series data (Liu et al., 2025a). While most existing synthetic pre-training methods (e.g., Chronos, TimesFM) rely on standard statistical components like GPs or ARMA processes, TempoPFN explicitly expands this design space by introducing novel generators for stochastic volatility (via SDEs) and temporal asymmetry (via sawtooth waves). Interestingly, TabPFN-TS relies on cross-domain adaptation, leveraging a model pre-trained on synthetic tabular data (via structural causal models) to effectively "feature-engineer" time series problems into tabular regression tasks, rather than learning temporal dynamics directly.

Parallel to this work, Graf et al. (2025) introduced *FlowState*, a time series foundation model that also leverages State Space Models (SSMs) for subquadratic computational efficiency. FlowState features an SSM-based encoder combined with a functional basis decoder to enable sampling-rate invariance and continuous-time modeling. While both FlowState and TempoPFN move away from Transformer-based architectures in favor of linear recurrences, our approaches differ fundamentally in their pre-training data paradigms. FlowState is pre-trained on a combination of real-world datasets (subsets of GIFT-Eval and Chronos corpora), whereas TempoPFN establishes the viability of a *purely synthetic* pre-training pipeline. Furthermore, while FlowState focuses on resolution adaptation, TempoPFN focuses on maximizing zero-shot generalization through a diverse synthetic prior designed and utilizing the GatedDeltaProduct for state tracking.

## B ARCHITECTURAL MECHANISMS

### B.1 GATEDDELTAPRODUCT ARCHITECTURE AND STATE TRACKING

To overcome the expressivity limitations of diagonal linear RNNs while retaining linear-time parallel scan computation, TempoPFN uses the *GatedDeltaProduct* recurrence (Siems et al., 2025). Unlike diagonal SSMs such as Mamba (Gu & Dao, 2023) or RWKV (Peng et al., 2023), whose state-

transition matrices are restricted to diagonal structure, GatedDeltaProduct employs a structured non-diagonal transition matrix constructed as a product of generalized Householder matrix updates. This yields a more expressive class of linear operators while keeping both training and inference efficient.

**Recurrence Mechanism.** Each layer maintains a matrix-valued hidden state $\boldsymbol{H}_t \in \mathbb{R}^{d \times n}$, updated via a linear recurrence:

$$\boldsymbol{H}_t = \boldsymbol{A}_t \boldsymbol{H}_{t-1} + \boldsymbol{B}_t, \qquad \boldsymbol{y}_t = \boldsymbol{H}_t \boldsymbol{x}_t. \tag{3}$$

Here, $x_t \in \mathbb{R}^D$ represents the input vector at the current time step $t$, where $D$ is the input dimension (e.g., number of features). $\boldsymbol{H}_t \in \mathbb{R}^N$ is the hidden state vector, representing the compressed memory, where $N$ is the state dimension. $\boldsymbol{A}_t \in \mathbb{R}^{N \times N}$ is the state-transition matrix, which defines how the information from the previous hidden state evolves and is stored in the new state. DeltaProduct utilizes a dense, parameterized matrix $\boldsymbol{A}_t$, which is explicitly constrained to be near orthogonal through its initialization and parameterization scheme (a product of $n_h$ rank-1 Householder-like updates):

$$\boldsymbol{A}_t = g_t \prod_{j=1}^{n_h} \left( \boldsymbol{I} - \beta_{t,j} \boldsymbol{k}_{t,j} \boldsymbol{k}_{t,j}^\top \right), \tag{4}$$

where $g_t \in [0, 1]$ is a forget gate, $\boldsymbol{k}_{t,j}$ are normalized key vectors, and $\beta_{t,j}$ are step sizes predicted from the input. The orthogonality constraint ensures that the transformation applied to the hidden state, $\boldsymbol{H}_{t-1}$, preserves its magnitude. Consequently, the hidden state $\boldsymbol{H}_t$ can effectively maintain its stability and information content over extended time steps, enabling robust state tracking of long-term trends and cyclical patterns in time series forecasting. Because each factor $\boldsymbol{I} - \beta \boldsymbol{k}\boldsymbol{k}^\top$ is a rank-1 update, the full matrix-vector product remains $\mathcal{O}(dn_h)$ and is fully parallelizable via a parallel prefix scan (Yang et al., 2024c). Finally, $\boldsymbol{B}_t \in \mathbb{R}^{N \times D}$ is the input weight matrix. This matrix transforms the current input $x_t$ and integrates it into the hidden state $\boldsymbol{H}_t$.

**Gating.** The gating in GatedDeltaProduct was introduced to embed essential non-linearity in the basic linear recurrence above, therefore enhancing the model's overall expressiveness and selective memory. After the main linear recurrence is performed, its output is processed along two parallel streams: 1) *Main Stream*: The result of the linear recurrence, intended for the final output; 2) *Gate Stream*: The same recurrent output is passed through a non-linear activation function (e.g., the SiLU or Swish function). These two streams are combined via element-wise multiplication (the gate). This operation selectively controls which parts of the recurrent output are emphasized or suppressed, mirroring the functionality of sophisticated recurrent units like the Gated Recurrent Unit (GRU) or the selectivity found in Mamba's architecture.

**State Weaving.** This mechanism was specifically designed for the overall multi-layer structure of the TempoPFN framework where multiple GatedDeltaProduct layers are stacked. Instead of simply discarding the final hidden state, the mechanism *weaves* temporal information across the depth of the model. Specifically, the final hidden state ($\boldsymbol{H}_T$) outputted by the first GatedDeltaProduct layer in the stack is passed forward to serve as the initial hidden state ($\boldsymbol{H}_0$) for the subsequent Gated-DeltaProduct layer. This ensures that deeper layers do not start their recurrence from a blank slate but instead *build upon the aggregated temporal state representations* learned by the shallower layers. This process creates a dense flow of information across both the time dimension (via the recurrence) and the model depth (via the weaving).

**State Tracking and Its Relevance in Time-Series Forecasting.** A central advantage of DeltaProduct is its ability to perform *state tracking*, i.e., maintaining and updating information over long sequences. Diagonal linear RNNs and SSMs (e.g., Mamba, RWKV, GLA) update each hidden dimension independently, which is efficient but severely limits expressivity: they cannot mix coordinates, cannot implement basic tracking functions such as parity or counting (Grazzi et al., 2025), and their states inevitably drift toward zero due to exponential decay. DeltaProduct avoids this failure mode through negative eigenvalues present in the structured non-diagonal transitions, where each factor $(\boldsymbol{I} - \beta \boldsymbol{k}\boldsymbol{k}^\top)$ acts as a reflection or low-rank rotation that preserves geometry and prevents collapse. This capability is crucial for time-series forecasting, where tracking corresponds to maintaining *trend* and *level* information across long contexts. As a result, GatedDeltaProduct layers maintain trend information without attenuation, enabling stable and coherent extrapolation over extended horizons.

# C SYNTHETIC DATA IMPLEMENTATION DETAILS

## C.1 GENERATOR SPECIFICATIONS

**CauKer.** To increase the diversity and structural complexity of our training data, we used the CauKer generator from (Xie et al., 2024). This method produces multivariate time series by sampling from a structural causal model (SCM) where each variable is a Gaussian process. A random Directed Acyclic Graph (DAG) defines the dependencies between nodes, with each node having a maximum number of parents. Root nodes in the DAG are sampled from a GP prior $y_i \sim \mathcal{GP}(m(t), \kappa(t, t'))$, using complex composite kernels $\kappa$ (combined via + or *) and stochastic mean functions $m(t)$ (e.g., linear $at + b$, exponential $a \exp(bt)$, or functions with anomalous impulses). Child nodes then apply nonlinear activation functions (e.g., ReLU, sigmoid, sin) to affine combinations of their parents' values, introducing intricate, non-Gaussian dependencies.

We generated multivariate series with 21 channels and treated each channel as an independent univariate time series. This approach allows us to efficiently create a vast corpus of realistic, interdependent patterns from a single generative process, providing the model with a rich and varied learning signal that encompasses trends, periodicities, and complex nonlinear interactions.

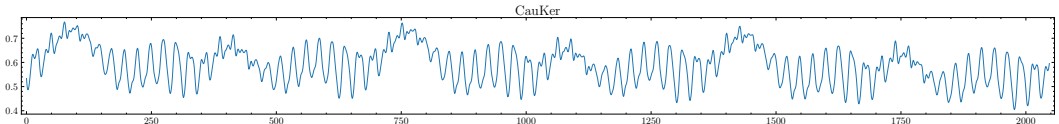

Figure 7: Example time series generated by CauKer

**KernelSynth.** The KernelSynth generator, based on Chronos (Ansari et al., 2024), samples independent univariate time series from Gaussian process priors $y \sim \mathcal{GP}(0, \kappa(t, t'))$. It constructs composite kernels $\kappa$ by randomly combining base kernels (using addition or multiplication) from a large bank. This bank includes periodic kernels (ExpSineSquared($p$) with periods $p$ normalized by series length), stationary kernels (RBF, RationalQuadratic), and noise kernels (WhiteKernel). This method efficiently produces a vast array of smooth and structured series, ideal for learning fundamental temporal representations.

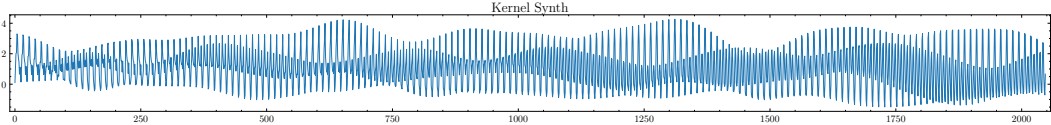

Figure 8: Example time series generated by KernelSynth

**Gaussian Process.** The Gaussian Process generator, inspired by Mamba4Cast (Bhethanabhotla & Swelam, 2024), extends the GP sampling approach with greater complexity and realism. It constructs a composite kernel by combining up to six base kernels from a weighted bank that includes Matern, linear, periodic, and polynomial kernels. The combination logic (addition or multiplication) is also chosen randomly. To generate realistic periodicities, the periods of any periodic kernels are sampled from distributions tailored to the time series' specified frequency (e.g., daily, weekly). Crucially, with a certain probability, we inject **periodic peak spikes** that are aligned with the dominant periodicity of the sampled kernel. This process creates sharp, recurring events on top of the smooth GP trajectory, yielding a wide range of both stationary and non-stationary series with complex covariance structures that mix smooth and abrupt dynamics.

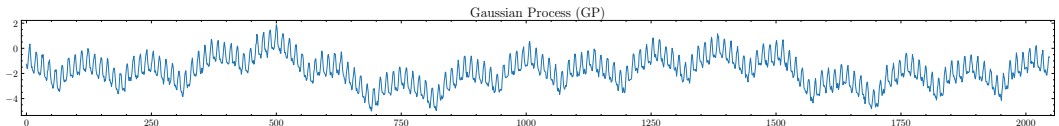

Figure 9: Example time series generated by GP

**ForecastPFN.** The ForecastPFN generator, adapted from Dooley et al. (2023), creates time series with configurable trends, seasonality, and noise patterns. The trend component combines linear and exponential elements multiplicatively for improved stability:

$$\tau(t) = [b + s_l(t + o_l)] \times s_e^{(t+o_e)}, \tag{5}$$

where the exponential base $s_e$ is carefully scaled based on series length and frequency to prevent unbounded growth. The seasonality component is also multiplicative:

$$s(t) = \prod_f \left( 1 + s_f \sum_h \left[ c_{f,h} \sin\left( \frac{2\pi h(t + o_f)}{p_f} \right) + d_{f,h} \cos\left( \frac{2\pi h(t + o_f)}{p_f} \right) \right] \right). \tag{6}$$

The final series values are given by $\tau(t) \cdot s(t) \cdot (1 + n(t))$, where $n(t)$ is Weibull-distributed noise. We enhanced this foundation with a noise injection strategy inspired by Bhethanabhotla & Swelam (2024), incorporating univariate augmentations like time warping, magnitude scaling, damping, and spike injection. A built-in filtering mechanism with retry logic ensures generated series avoid unrealistic spreads or extreme values, guaranteeing robust training data.

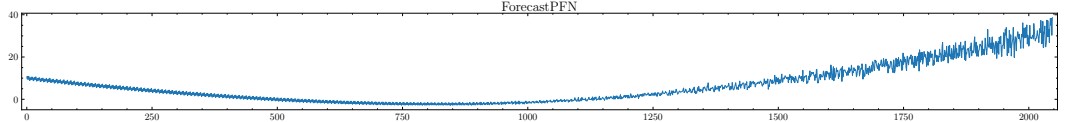

Figure 10: Example time series generated by ForecastPFN

**Sawtooth.** The Sawtooth generator creates univariate series with linear ramping patterns. The core waveform is a sawtooth function: $y_t = A \cdot \text{frac}((t/P) + \phi)$ for upward ramps, or $y_t = A \cdot (1 - \text{frac}((t/P) + \phi))$ for downward ramps (direction chosen randomly). To prevent overly idealised signals, minimal linear trends ($s_l t$) and low-amplitude seasonal components ($a \sin(2\pi t/Q)$) are added. This encourages the model to learn robust representations of trend-dominated series.

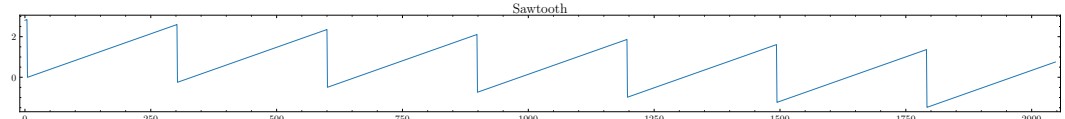

Figure 11: Example time series generated by Sawtooth Generator

**Step Function.** Our Step Function generator constructs complex piecewise constant series by concatenating multiple subseries. Each subseries is generated from a configurable distribution of patterns (stable, gradual trends, spikes, oscillations, random walks) with specific lengths, number of changepoints, step sizes, and drift. The combined series undergoes optional Gaussian smoothing at transitions. Finally, global components like noise, seasonality, a linear trend, and point anomalies are added, creating rich and non-stationary step-like data.

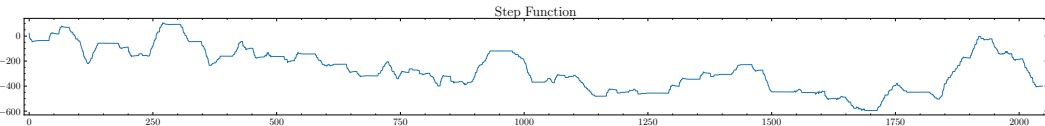

Figure 12: Example time series generated by Step Function

**Anomaly.** The Anomaly generator focuses on outlier detection by producing otherwise constant baseline signals contaminated with periodic spike anomalies. For a given series, all spikes are exclusively positive or negative. Their timing follows patterns (single, clustered, or mixed) with period variance and jitter, while their magnitudes follow defined regimes (constant, trending, cyclical, or correlated random). This provides a controlled environment for learning anomaly detection semantics.

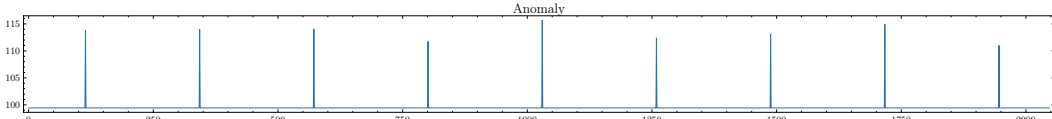

Figure 13: Example time series generated by Anomaly Generator

**Spikes.** The Spikes generator creates series where the primary feature is the spike itself, defined on a flat baseline. Spikes have consistent per-series direction and shape (V-shaped, inverted-V, or chopped variants with plateaus). They are generated in either "burst" (clustered) or "spread" (evenly spaced with defined edge margins) modes. Colored (brown/pink) noise is added probabilistically. This generator is designed to simulate event-driven signals common in domains like healthcare or intrusion detection.

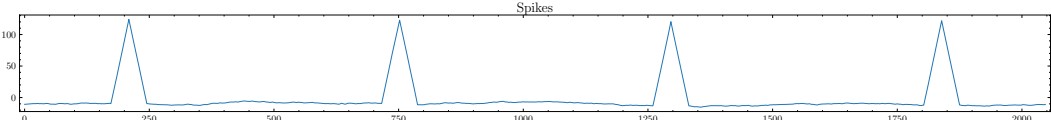

Figure 14: Example time series generated by Spike Generator

**Sine Wave.** Our Sine Wave generator produces complex and non-stationary oscillatory patterns, moving beyond simple periodic signals. It generates a time series by summing 1 to 3 sinusoidal components, each subject to modulation, and then adds a global trend and noise. The underlying model is:

$$y_t = \sum_{i=1}^{N} A_i(t) \sin\left(\phi_i(t)\right) + (at + b) + \epsilon_t$$

Here, $A_i(t)$ represents a time-varying amplitude and $\phi_i(t)$ is a time-varying phase. This is achieved through slow **amplitude and frequency modulation**, where the amplitude and instantaneous frequency of each sine wave are themselves modulated by another low-frequency sinusoid. This technique introduces realistic drifts and warping in the periodic patterns, preventing the signal from being perfectly predictable. A final linear trend $(at + b)$ and Gaussian noise $\epsilon_t$ are added to complete the series.

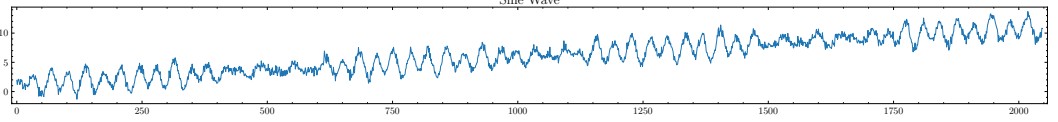

Figure 15: Example time series generated by Sine Wave Generator

**Audio-Inspired Generators.** To generate exceptionally complex and realistic time series, we introduce a family of four novel generators based on procedural audio synthesis techniques, using the `pyo` digital signal processing library. An audio synthesis graph is constructed with various oscillators and modulators, rendered offline, and then resampled to the target time series length. This paradigm allows us to model intricate, dynamic systems.

- **Stochastic Rhythm:** This generator creates multi-layered, event-driven patterns. A base tempo is set, and 3 to 5 rhythmic layers are created on top, each triggering at a random subdivision of the tempo (e.g., twice, three times, etc.). Each trigger fires a percussive envelope controlling a sine wave oscillator, resulting in a complex, polyrhythmic signal ideal for modeling data with recurring, patterned events.
- **Financial Volatility:** This generator mimics financial market dynamics. It combines three components: a slow-moving LFO that acts as the market trend, a Brownian noise source whose amplitude is modulated to create *volatility clustering*, and a triggered, sharp envelope that creates sudden positive or negative *jumps* or shocks.
- **Network Topology:** This generator simulates network traffic data. The signal is a mixture of five components: a base traffic flow (slow LFO), high-frequency noise bursts representing packet traffic, periodic dips from triggered envelopes to model congestion, a high-frequency sine wave for protocol overhead, and large, sharp spikes from filtered noise to simulate DDoS-like attacks.
- **Multi-Scale Fractal:** This generator produces self-similar, fractal-like patterns. A Brownian noise source is passed through a bank of 3 to 6 parallel band-pass filters. The center frequencies of these filters are logarithmically spaced, and each successive filter has a higher attenuation. Summing the outputs creates a signal with structure at multiple time scales.

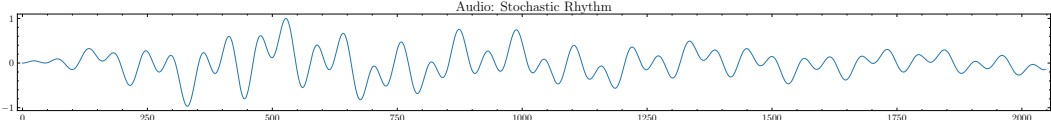

Figure 16: Example time series generated by Audio Stochastic Rhythm

**Stochastic Differential Equations (SDEs).** SDEs provide a principled framework for modeling continuous-time random processes. An SDE specifies the infinitesimal dynamics of a state variable $y_t$ as

$$dy_t = a(y_t, t)dt + b(y_t, t)dW_t, \tag{7}$$

where $a(\cdot, \cdot)$ is the *drift function* governing deterministic trends, $b(\cdot, \cdot)$ is the *diffusion function* controlling random fluctuations, and $W_t$ is a standard Brownian motion. Unlike deterministic differential equations, solutions are random trajectories whose distribution is determined by $(a, b)$ and the distribution of initial conditions.

We adopt the Itô convention of stochastic calculus. This choice is standard in financial mathematics and machine learning because Itô integrals enjoy martingale properties. For simulation, we discretize the SDE on a time grid $\{0, \Delta t, 2\Delta t, \ldots, T\}$ using the Euler–Maruyama scheme:

$$y_{t+\Delta t} = y_t + a(y_t, t)\Delta t + b(y_t, t)\sqrt{\Delta t}Z_t, \quad Z_t \sim \mathcal{N}(0, 1). \tag{8}$$

More advanced schemes such as the Milstein method can reduce bias when the diffusion term depends on $y_t$, but Euler–Maruyama suffices for our purposes.

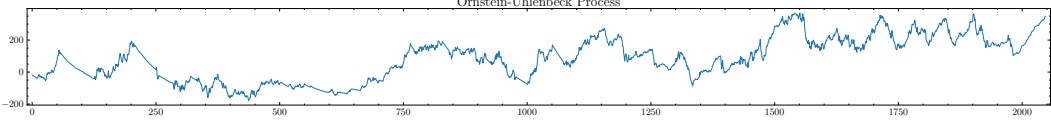

Figure 17: Example time series generated by Ornstein–Uhlenbeck process

Equation $dy_t = \theta(t, r_t)\big(\mu(t, r_t) - y_t\big)dt + \sigma(t, r_t)dW_t$, where $\theta(t, r_t)$ is the mean reversion speed, $\mu(t, r_t)$ the time-varying mean, and $\sigma(t, r_t)$ the volatility, defines the process. In regime $r_t \in \{0, 1\}$, the drift and diffusion coefficients are parameterized as

$$\theta(t, r_t) = \theta^{(r_t)} \cdot (1 + \delta_\theta(t)), \tag{9}$$

$$\mu(t, r_t) = \mu^{(r_t)} + \mu^{\text{trend}}(t) + \mu^{\text{season}}(t), \tag{10}$$

$$\sigma(t, r_t) = \sigma^{(r_t)} \cdot \big(1 + \sigma^{\text{trend}}(t) + \sigma^{\text{season}}(t)\big), \tag{11}$$

where $\delta_\theta(t)$, $\mu^{\text{trend}}(t)$, $\sigma^{\text{trend}}(t)$ are smooth trend functions (e.g., linear, logistic, polynomial), and $\mu^{\text{season}}(t)$, $\sigma^{\text{season}}(t)$ are sinusoidal seasonal components with possible amplitude evolution. Regime switching occurs with probabilities $p_{00}, p_{11} \in [0.85, 0.999]$. The initial state is drawn from $X_0 \sim \mathcal{N}(\mu^{(r_0)}, \sigma^{(r_0)2})$, with $r_0$ chosen uniformly. Each path is subsequently transformed via a global scaling factor $s \sim U[0.1, 50.0]$, global level shift $\ell \sim U[-100, 100]$, and additive Gaussian measurement noise $\epsilon_t \sim \mathcal{N}(0, \sigma_\epsilon^2)$ with $\sigma_\epsilon \in [0, 0.1]$. When long memory is enabled, $W_t$ is replaced with fractional Brownian motion $B_t^H$ with Hurst exponent $H \in [0.3, 0.8]$. Table 4 summarizes the sampling ranges for all parameters used in the generator.

Table 4: Parameter ranges for the Regime-Switching OU generator.

| Parameter | Range / Distribution |
|---|---|
| Integration step size $dt$ | 0.01 |
| Initial value $y_0$ | $\mathcal{N}(0, 2^2)$ |
| Regime 0 mean reversion $\theta^{(0)}$ | $[1.0, 5.0]$ |
| Regime 0 mean $\mu^{(0)}$ | $\mathcal{N}(-2.0, 1.0^2)$ |
| Regime 0 volatility $\sigma^{(0)}$ | $\log \mathcal{N}(\log 0.3, 0.3)$ |
| Regime 0 vol. process $(\kappa_v, \theta_v, \xi_v)$ | $[2.0, 5.0], [0.2, 0.4], [0.1, 0.3]$ |
| Regime 1 mean reversion $\theta^{(1)}$ | $[0.05, 0.5]$ |
| Regime 1 mean $\mu^{(1)}$ | $\mathcal{N}(2.0, 1.0^2)$ |
| Regime 1 volatility $\sigma^{(1)}$ | $\log \mathcal{N}(\log 1.5, 0.5)$ |
| Regime 1 vol. process $(\kappa_v, \theta_v, \xi_v)$ | $[0.5, 2.0], [0.8, 1.2], [0.3, 0.5]$ |
| Regime transition probs $p_{00}, p_{11}$ | $[0.85, 0.999]$ |
| Global level shift $\ell$ | $[-100.0, 100.0]$ |
| Global scale factor $s$ | $[0.1, 50.0]$ |
| Measurement noise std $\sigma_\epsilon$ | $[0.0, 0.1]$ |
| Hurst exponent $H$ | $[0.3, 0.8]$ |
| Seasonal components | 1–3 harmonics |
| Seasonal periods | $\{7.0, 30.0, 90.0, 182.6, 365.25\}$ |
| Seasonal amplitude | $[0.5, 3.0]$ |
| Seasonal phase shift | $[0, 2\pi]$ |
| Seasonal period jitter | $\pm 5\%$ |
| Seasonal amplitude evolution | $[-0.001, 0.001]$ |
| Trend application probs | $\mu : 0.7, \ \theta : 0.2, \ \sigma : 0.3$ |
| Seasonality application probs | $\mu : 0.6, \ \sigma : 0.3$ |

## C.2 SYNTHETIC DATA GENERATION THROUGHPUT

In this section, we present the computational efficiency and resource flexibility of our pipeline. Unlike kernel-based methods such as KernelSynth, which can be computationally intensive due to the cubic $O(T^3)$ complexity of Gaussian Processes, our approach enables high-throughput generation as shown in Table 5.

The benchmarking was conducted on a high-performance system featuring dual AMD EPYC 9334 32-Core Processors (128 threads total) and an NVIDIA L40S GPU. Crucially, *the majority of our synthetic generators run exclusively on the CPU*. The GPU is leveraged primarily for the few neural network-based prior models (e.g. Cauker).

## D TRAINING DETAILS AND HYPERPARAMETERS

**Data Composition and Sampling.** The training corpus consists of approximately 10 million synthetic time series (500k–2M per generator), with batches composed of mixed samples from our generators. We apply higher weights to Cauker and augmented data to promote diversity in the training distribution.

**Dynamic Structure Construction.** Our training uses dynamic, per-sample construction of time series structures. For each training instance, we first randomly sample a total sequence length from

Table 5: Profiling results for synthetic data generation throughput. **N** = Novel prior, **A** = Adapted from open-source.

| Generator | Source | Length | Series / Sec |
|---|---|---|---|
| Cauker | A | 2048 | 0.66 |
| GP | A | 2048 | 7.04 |
| Kernel | A | 2048 | 0.32 |
| ForecastPFN | A | 2048 | 35.49 |
| Sawtooth | N | 2048 | 242.95 |
| Sinewave | N | 2048 | 144.93 |
| Anomaly | N | 2048 | 174.51 |
| Step | N | 2048 | 106.58 |
| Stochastic Rhythm | N | 2048 | 33.46 |
| Spike | N | 2048 | 201.13 |
| SDE (OU Process) | N | 2048 | 13.17 |
| Offline augmentations | N | 2048 | 18.30 |

a weighted distribution that favors longer contexts: $\{128: 0.05, 256: 0.10, 512: 0.10, 1024: 0.10, 1536: 0.15, 2048: 0.50\}$. When length shortening is applied, we use either cutting or subsampling with equal probability (50/50 split). Next, we perform a random history-future split, with forecast horizon lengths sampled from the range specified by the GIFT benchmark. This two-stage sampling creates highly variable training examples that simulate diverse forecasting tasks.

**Data Augmentation.** We apply several augmentation techniques during training: (1) Scaler augmentation with 0.5 probability, randomly selecting among minmax, median, or mean scalers (excluding the main robust scaler); (2) NaN augmentation that injects realistic missing data patterns into the history based on GIFT-Eval statistics.

**Training Infrastructure.** Pretraining uses PyTorch with distributed data parallelism (DDP) across 8–16 NVIDIA A100 or H100 GPUs and mixed precision (`bfloat16`), a requirement for the DeltaProduct implementation (in FLA: `https://github.com/fla-org/flash-linear-attention`).

**Training Protocol.** For pretraining, we employ the AdamW optimizer (Loshchilov & Hutter, 2019) with a weight decay of 0.01 and an effective batch size of approximately 200. No additional regularization techniques—such as dropout or early stopping—are applied. Pretraining is conducted for 4 million iterations using a cosine annealing learning-rate schedule (Loshchilov & Hutter, 2017) with a peak learning rate of $2 \times 10^{-4}$, a warmup ratio of 0.003, and a minimum learning-rate ratio of 0.01. The model is trained using the quantile regression loss, computed independently for each output token across the set of quantile levels $\mathcal{Q} = \{q_1, q_2, \dots, q_m\}$. In our experiments, we set $\mathcal{Q} = \{0.1, 0.2, \dots, 0.9\}$ similarly as in Tirex and TabPFN-TS. The resulting losses are then averaged over all $h$ output tokens in a training sample. Given the true value $y_t$ at time $t$ and its predicted quantile value $\hat{y}_t^{(q)}$ for quantile level $q \in \mathcal{Q}$, the loss is defined as:

$$L = \frac{1}{|\mathcal{Q}| \, h} \sum_{t=1}^{h} \sum_{q \in \mathcal{Q}} \begin{cases} q\left(y_t - \hat{y}_t^{(q)}\right), & \text{if } \hat{y}_t^{(q)} \leq y_t, \\ (1-q)\left(\hat{y}_t^{(q)} - y_t\right), & \text{otherwise.} \end{cases}$$

**Architecture Selection.** We ablated deeper models (8–16 layers) and found no consistent architectural winner. We selected the 10 layer model with embedding dimension of 512.

Table 6: Hyperparameters for main TempoPFN model.

| Category | Parameter | Value |
|---|---|---|
| Model | **Total Parameters** | **34.69** |
| | Embedding size (`embed_size`) | 512 |
| | Encoder layers | 10 |
| | Number of heads (`num_heads`) | 4 |
| | Encoder attention mode | `chunk` |
| | Short convolution kernel size | 32 |
| | State weaving | `True` |
| | Quantiles for loss | [0.1, 0.2, 0.3, 0.4, 0.5, 0.6, 0.7, 0.8, 0.9] |
| Training | Total training series | $\approx 10,000,000$ |
| | Max series length | 2048 |
| | Total training iterations | 4,000,000 |
| | Batch size (per GPU) | 40 |
| | Gradient accumulation steps | 5 |
| | **Effective batch size** | 200 |
| | Peak learning rate | $2 \times 10^{-4}$ |
| | LR scheduler | Cosine annealing |
| | Min learning rate ratio | 0.01 |
| | Warmup ratio | 0.003 |
| Optimization | Optimizer | AdamW |
| | $\beta_1$ | 0.9 |
| | $\beta_2$ | 0.98 |
| | Weight decay | 0.01 |
| | Adam $\epsilon$ | $1 \times 10^{-6}$ |
| | Gradient clipping | 100.0 |
| Augmentations | Length shortening | `True` (cut/subsample: 50/50) |
| | NaN augmentation | `True` |
| | Scaler augmentation prob. | 0.5 (minmax/median/mean) |
| | Batch composition | Mixed (proportions favoring augmented/Cauker) |
| Hardware | GPUs | $8\text{--}16 \times$ A100/H100 |
| | Precision | `bfloat16` |

## D.1 ADDITIONAL EXPERIMENTAL DETAILS AND RESULTS

The results presented in this section are based on ablation studies conducted with our main model architecture.

Table 7: Ablation study of single synthetic priors (trained for **500k iterations**). 'Base Model' uses all priors and augmentations. Lower values are better. **Bold**: best, underline: second-best. Novel priors are our contributions; Adapted are modified open-source versions.

| Ablation | Source | Gift-ZS Overall | | Gift-ZS Short | | Gift-ZS Medium | | Gift-ZS Long | |
|---|---|---|---|---|---|---|---|---|---|
| | | CRPS | MASE | CRPS | MASE | CRPS | MASE | CRPS | MASE |
| Base Model | – | 0.578 | 0.842 | 0.563 | 0.763 | 0.566 | 0.900 | 0.631 | 1.019 |
| + Cauker | Adapted | **0.600** | **0.875** | **0.583** | **0.789** | **0.615** | **0.964** | **0.631** | **1.043** |
| + GP | Adapted | 0.632 | 0.897 | 0.607 | 0.812 | 0.666 | 0.993 | 0.666 | 1.053 |
| + Kernel | Adapted | 0.638 | 0.926 | 0.622 | 0.835 | 0.656 | 1.042 | 0.661 | 1.082 |
| + ForecastPFN | Adapted | 0.715 | 1.027 | 0.695 | 0.918 | 0.760 | 1.172 | 0.726 | 1.206 |
| + SDE (OU Process) | Novel | **0.815** | **1.148** | **0.763** | **1.017** | **0.897** | **1.334** | **0.879** | **1.354** |
| + Sinewave | Novel | 0.868 | 1.223 | 0.854 | 1.113 | 0.901 | 1.375 | 0.872 | 1.397 |
| + Stochastic Rhythm | Novel | 0.953 | 1.337 | 0.940 | 1.252 | 1.004 | 1.472 | 0.938 | 1.440 |
| + Sawtooth | Novel | 1.187 | 1.534 | 1.162 | 1.362 | 1.294 | 1.802 | 1.152 | 1.781 |
| + Spike | Novel | 1.215 | 1.318 | 1.019 | 1.250 | 1.565 | 1.411 | 1.498 | 1.416 |
| + Anomaly | Novel | 1.310 | 1.522 | 1.487 | 1.610 | 1.145 | 1.430 | 1.075 | 1.399 |
| + Step | Novel | 2.199 | 1.702 | 1.272 | 1.280 | 4.325 | 2.398 | 4.693 | 2.549 |
| seasonal_naive | – | 1.000 | 1.000 | 1.000 | 1.000 | 1.000 | 1.000 | 1.000 | 1.000 |

Table 8: Core architectural ablations (trained for **2M iterations**). Base config: $d = 512$, $L = 10$, conv size 16, $H = 4$, weaving enabled, negative eigenvalues allowed. Sorted by overall CRPS. **Bold**: best, underline: second-best.

| Configuration | Gift-ZS Overall | | Gift-ZS Short | | Gift-ZS Medium | | Gift-ZS Long | |
|---|---|---|---|---|---|---|---|---|
| | CRPS | MASE | CRPS | MASE | CRPS | MASE | CRPS | MASE |
| *Ablating Positional Encoding:* | | | | | | | | |
| Base Model (Sin. Pos. Enc. Off) | **0.561** | **0.820** | **0.553** | **0.751** | **0.556** | **0.880** | **0.590** | **0.962** |
| Sinusoidal Positional Encoding | 0.648 | 0.937 | 0.596 | 0.809 | 0.731 | 1.120 | 0.715 | 1.152 |
| *Ablating Number of Householder Matrices (H):* | | | | | | | | |
| H=6 | **0.556** | 0.823 | **0.545** | **0.750** | **0.552** | 0.886 | **0.590** | 0.972 |
| Base Model (H=4) | 0.561 | **0.820** | 0.553 | 0.751 | 0.556 | **0.880** | **0.590** | **0.962** |
| H=2 | 0.562 | 0.822 | 0.549 | 0.750 | 0.560 | 0.892 | 0.598 | 0.964 |
| H=1 (DeltaNet equivalent) | 0.573 | 0.845 | 0.556 | 0.761 | 0.579 | 0.918 | 0.613 | 1.020 |
| *Ablating Negative Eigenvalues and Weaving:* | | | | | | | | |
| Neg. Eig. Off, Weaving On | **0.559** | 0.821 | **0.553** | 0.753 | **0.550** | 0.881 | **0.584** | 0.957 |
| Neg. Eig. Off, Weaving Off | 0.560 | **0.818** | 0.554 | **0.750** | 0.548 | **0.879** | 0.590 | **0.955** |
| Base Model (Neg. Eig. On, Weaving On) | 0.561 | 0.820 | 0.553 | 0.751 | 0.556 | 0.880 | 0.590 | 0.962 |
| *Ablating Convolution Size:* | | | | | | | | |
| Conv. size 32 | **0.559** | **0.816** | **0.543** | **0.737** | 0.566 | 0.897 | 0.594 | 0.968 |
| Base Model (Conv. size 16) | 0.561 | 0.820 | 0.553 | 0.751 | **0.556** | **0.880** | **0.590** | **0.962** |
| seasonal_naive | 1.000 | 1.000 | 1.000 | 1.000 | 1.000 | 1.000 | 1.000 | 1.000 |

Table 9: Ablation of model scale and depth (trained for **4M iterations**). Base Model: $d = 512$, $L = 10$, $H = 4$, conv size 32, weaving/neg eigenvalues on. Compares width vs. depth at constant parameter count. Sorted by overall CRPS. **Bold**: best, underline: second-best.

| Configuration | Gift-ZS Overall | | Gift-ZS Short | | Gift-ZS Medium | | Gift-ZS Long | |
|---|---|---|---|---|---|---|---|---|
| | CRPS | MASE | CRPS | MASE | CRPS | MASE | CRPS | MASE |
| Base Model (d=512, L=10) | **0.533** | **0.788** | **0.532** | 0.727 | **0.523** | **0.840** | **0.544** | **0.912** |
| d=512, L=10, Weaving Off | 0.537 | 0.790 | **0.532** | **0.723** | 0.533 | 0.862 | 0.553 | 0.914 |
| d=384, L=16 (Narrower, Deeper) | 0.539 | 0.792 | 0.532 | 0.727 | 0.533 | 0.850 | 0.563 | 0.921 |
| d=576, L=8 (Wider, Shallower) | 0.540 | 0.794 | 0.536 | 0.732 | 0.529 | 0.849 | 0.561 | 0.921 |
| seasonal_naive | 1.000 | 1.000 | 1.000 | 1.000 | 1.000 | 1.000 | 1.000 | 1.000 |

# E    COMPREHENSIVE QUANTITATIVE ANALYSIS

## E.1    COMPUTATIONAL COMPLEXITY AND EFFICIENCY

Table 11 summarizes the computational characteristics of TempoPFN relative to leading time-series foundation models, given sequence length $T$, horizon $H$, embedding dimension $d$, and layers $L$.

**Training Complexity.** Transformer-based models (Chronos, TimesFM, MOIRAI, TabPFN-TS) require $\mathcal{O}(T^2 d)$ compute and memory due to self-attention, which becomes prohibitive for long context windows. TiRex reduces quadratic memory growth but remains sequential along $T$. In contrast, TempoPFN employs an associative GatedDeltaProduct recurrence, allowing parallel prefix-scan evaluation. This yields linear total work $\mathcal{O}(T L d^2)$ and logarithmic parallel depth $\mathcal{O}(L \log T)$, enabling full sequence-length parallelism.

**Inference Latency.** Autoregressive models such as TiRex and Chronos must unroll $H$ steps to predict a horizon $H$, yielding $\mathcal{O}(H)$ latency. Transformer encoder models also scale their inference cost with $T^2$ even when used non-autoregressively. TempoPFN performs *direct forecasting*: concatenated query tokens allow the entire horizon to be produced in a single forward pass, giving constant $\mathcal{O}(1)$ latency with respect to $H$.

**Memory Usage.** Transformer-based models require caching Key-Value pairs with $\mathcal{O}(T d)$ memory and, in some implementations, up to $\mathcal{O}(T^2)$ activations. TiRex maintains a hidden state of size $\mathcal{O}(T d)$ during training. TempoPFN, being a Linear RNN, compresses the entire past into a single

Table 10: LR scheduler ablation (trained for **2M iterations**). Base architecture: $d = 512$, $L = 10$, $H = 4$, conv size 32, weaving enabled. **WarmupStableDecay**: warmup (0.3%), plateau (90%), cosine decay (9.7%). **CosineWithRestarts**: 4 resets. Sorted by overall CRPS. **Bold**: best, underline: second-best.

| LR Scheduler | Gift-ZS Overall | | Gift-ZS Short | | Gift-ZS Medium | | Gift-ZS Long | |
|---|---|---|---|---|---|---|---|---|
| | CRPS | MASE | CRPS | MASE | CRPS | MASE | CRPS | MASE |
| WarmupStableDecay | **0.554** | **0.812** | **0.544** | **0.740** | **0.550** | 0.877 | **0.584** | 0.956 |
| CosineWithWarmup | 0.559 | 0.817 | 0.552 | 0.751 | **0.550** | **0.873** | 0.588 | **0.955** |
| CosineWithRestarts | 0.559 | 0.820 | 0.552 | 0.755 | 0.552 | 0.874 | 0.585 | 0.953 |
| Cosine (no warmup) | 0.561 | 0.820 | 0.553 | 0.751 | 0.556 | 0.880 | 0.590 | 0.962 |
| seasonal_naive | 1.000 | 1.000 | 1.000 | 1.000 | 1.000 | 1.000 | 1.000 | 1.000 |

hidden state and supports streaming inference with constant $\mathcal{O}(d)$ memory, while still allowing optional $\mathcal{O}(Td)$ state storage if needed for analysis or hybrid decoding.

**Parallelization.** Transformer-based models benefit from substantial batch-level parallelism but cannot eliminate the quadratic attention bottleneck. TiRex provides limited scan-style parallelism. TempoPFN achieves *full* sequence-level parallelization: the entire recurrence is computed via parallel scans, providing both high throughput and sublinear parallel depth.

Overall, TempoPFN combines linear training cost, logarithmic parallel depth, constant-latency forecasting, and streaming memory usage, therefore, providing a zero-shot foundation model tailored for long-context forecasting settings.

Table 11: Comparison of time-series foundation models. Complexities are reported with respect to sequence length $T$, horizon $H$, embedding dimension $d$, and layers $L$. We distinguish total work from parallel depth. TempoPFN benefits from scan-accelerated linear recurrences enabling sublinear depth.

| Model | Params (M) | Training Time | Inference Time | Memory | Parallelization |
|---|---|---|---|---|---|
| TiRex (Auer et al., 2025) | 35 | Work: $\mathcal{O}(TLd^2)$ Depth: $\mathcal{O}(L\log T)$ | AR: $\mathcal{O}(HLd^2)$ Depth: $\mathcal{O}(L\log T)$ | $\mathcal{O}(d)$ (streaming) | Moderate (scan) |
| TabPFN-TS (Hoo et al., 2024) | 11 | $\mathcal{O}(T^2d)$ | $\mathcal{O}(T^2d)$ | $\mathcal{O}(Td)$ | Moderate (attention) |
| TimesFM-2.0 (Das et al., 2024) | 500 | $\mathcal{O}(T^2d)$ | Direct: $\mathcal{O}(T^2d)$ | $\mathcal{O}(Td)$ | High (transformer) |
| Chronos (Ansari et al., 2024) | 9–205 | $\mathcal{O}(T^2d)$ | AR: $\mathcal{O}(H)$ | $\mathcal{O}(Td)$ | High (transformer) |
| MOIRAI-MoE (Liu et al., 2025b) | 14–935 | $\mathcal{O}(T^2d)$ | $\mathcal{O}(T^2d)$ | $\mathcal{O}(Td)$ | High (MoE + transformer) |
| **TempoPFN (ours)** | 35 | Work: $\mathcal{O}(TLd^2)$ Depth: $\mathcal{O}(L\log T)$ | Work: $\mathcal{O}(TLd^2)$ Depth: $\mathcal{O}(L\log T)$ Horizon: $\mathcal{O}(1)$ | $\mathcal{O}(d)$ (streaming) or $\mathcal{O}(Td)$ (cached) | **Full (sequence-parallel)** |

### E.2 CHRONOS-ZS BENCHMARK RESULTS

We evaluate TempoPFN on the **Chronos Zero-Shot** benchmark (Ansari et al., 2024). This benchmark comprises 27 diverse datasets from the GluonTS and Monash repositories, spanning multiple domains (e.g., energy, transport, healthcare) and frequencies. Figure 18 shows the aggregated performance in terms of Normalized and Average Rank for both probabilistic (CRPS) and point (MASE) forecasting.

### E.3 FEV-BENCH RESULTS

To demonstrate generalizability beyond Gift-Eval, we evaluate on the **fev-bench** framework, which standardizes evaluation across 100 diverse forecasting tasks. This benchmark is critical for validating zero-shot performance as it rigorously tracks data leakage and failure rates.

**Metrics.** We report results based on two metrics: MASE and *Scaled Quantile Loss* (SQL). SQL captures calibration quality by evaluating the quality of the entire predictive distribution at each time step. Following the fev-bench protocol, model performance is summarized using two aggregate scores derived from the pairwise error matrices: *Win Rate (%)*, representing the fraction of model

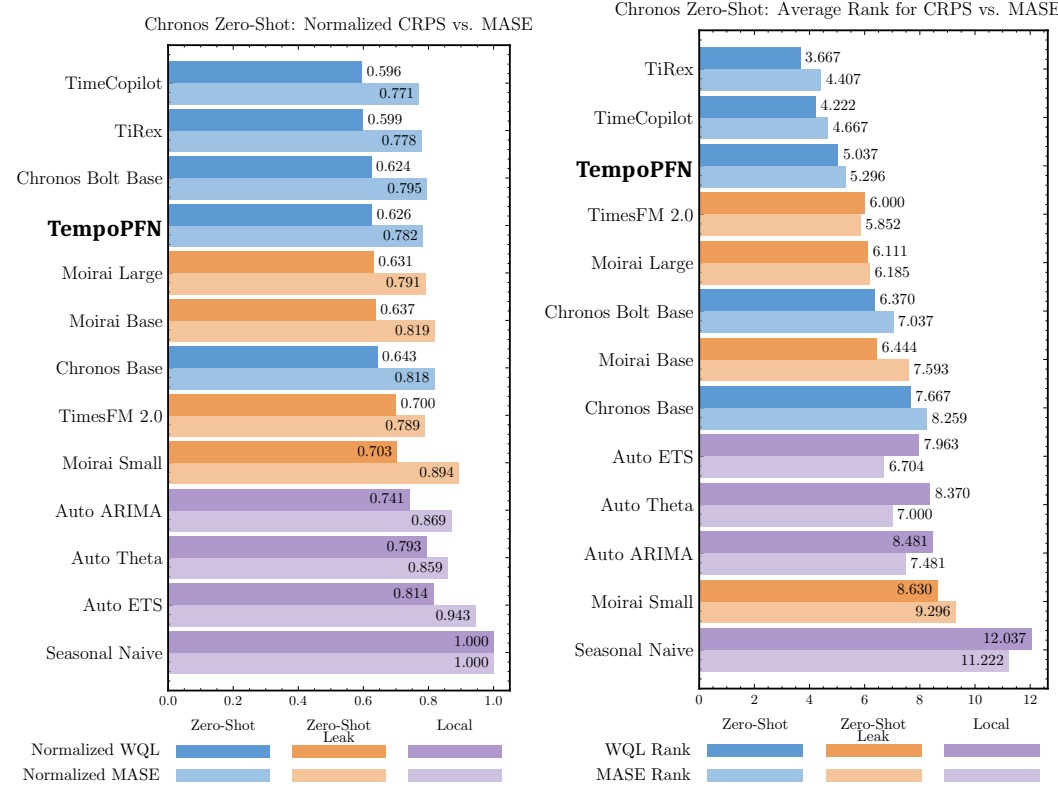

Figure 18: Comparison of TempoPFN performance (4M iterations), against other models on Chronos-Zeroshot benchmark. We compute both normalized and average ranks for CRPS and MASE. Colors represent the class of time series model.

pairs and tasks where the model achieves a lower error than the competitor, and *Skill Score (%)*, a robust measure of relative error reduction compared to the Seasonal Naive baseline.

**Leaderboard Results.** Table 12 presents the leaderboard for MASE, where TempoPFN achieves *Rank 6*. Table 13 presents the leaderboard for SQL, where TempoPFN also achieves *Rank 6*. In both metrics, our model outperforms the other leading synthetic-only baseline, TabPFN-TS (Rank 8 in both). To also visualize relative strengths in probabilistic forecasting, we show in Figure 19 the head-to-head Win Rates and Skill Scores based on SQL.

Table 12: FEV-Bench Leaderboard based on **MASE**. Models are ranked by Win Rate and Skill Score. The **TempoPFN** row is highlighted.

| Rank | Model | Avg. Win Rate (%) | Skill Score (%) | Median Runtime (s) | Leakage (%) | Failed Tasks (%) | Organization | Zero-shot |
|------|-------|-------------------|-----------------|--------------------|-------------|------------------|--------------|-----------|
| 1 | Chronos-2 | 88.0 | 35.5 | 3.57 | 0 | 0 | AWS | ✓ |
| 2 | TiRex | 76.7 | 30.0 | 1.4 | 1 | 0 | NX-AI | ✓ |
| 3 | TimesFM 2.5 | 74.9 | 30.2 | 10.89 | 10 | 0 | Google | ✓ |
| 4 | Toto 1.0 | 66.5 | 28.2 | 77.51 | 8 | 0 | Datadog | ✓ |
| 5 | Moirai 2.0 | 60.5 | 27.3 | 1.9 | 28 | 0 | Salesforce | ✓ |
| **6** | **TempoPFN** | **60.5** | **25.1** | **8.57** | **0** | **0** | **Anonymous** | ✓ |
| 7 | Chronos-Bolt | 60.1 | 26.5 | 1.0 | 0 | 0 | AWS | ✓ |
| 8 | TabPFN-TS | 58.2 | 27.6 | 300.57 | 0 | 2 | Prior Labs | ✓ |
| 9 | Sundial-Base | 52.4 | 24.7 | 33.99 | 1 | 0 | Tsinghua University | ✓ |
| 10 | Stat. Ensemble | 47.1 | 15.7 | 624.45 | 0 | 11 | — | ✗ |
| 11 | AutoARIMA | 35.6 | 11.2 | 120.16 | 0 | 10 | — | ✗ |
| 12 | AutoTheta | 33.6 | 11.0 | 9.27 | 0 | 0 | — | ✗ |
| 13 | AutoETS | 32.6 | 2.3 | 16.24 | 0 | 3 | — | ✗ |
| 14 | Seasonal Naive | 20.0 | 0.0 | 2.32 | 0 | 0 | — | ✗ |
| 15 | Naive | 18.4 | -16.7 | 2.24 | 0 | 0 | — | ✗ |
| 16 | Drift | 14.9 | -18.1 | 2.19 | 0 | 0 | — | ✗ |

Table 13: FEV-Bench Leaderboard based on **Scaled Quantile Loss (SQL)**. Models are ranked by Win Rate and Skill Score. The **TempoPFN** row is highlighted.

| Rank | Model | Avg. Win Rate (%) | Skill Score (%) | Median Runtime (s) | Leakage (%) | Failed Tasks (%) | Organization | Zero-shot |
|---|---|---|---|---|---|---|---|---|
| 1 | Chronos-2 | 91.3 | 47.3 | 3.57 | 0 | 0 | AWS | ✓ |
| 2 | TiRex | 82.4 | 42.6 | 1.4 | 1 | 0 | NX-AI | ✓ |
| 3 | TimesFM 2.5 | 77.3 | 42.2 | 10.89 | 10 | 0 | Google | ✓ |
| 4 | Toto 1.0 | 69.9 | 40.7 | 77.51 | 8 | 0 | Datadog | ✓ |
| 5 | Moirai 2.0 | 63.6 | 39.3 | 1.9 | 28 | 0 | Salesforce | ✓ |
| **6** | **TempoPFN** | **63.4** | **37.8** | **8.57** | **0** | **0** | **Anonymous** | ✓ |
| 7 | Chronos-Bolt | 63.2 | 38.9 | 1.0 | 0 | 0 | AWS | ✓ |
| 8 | TabPFN-TS | 62.0 | 39.6 | 300.57 | 0 | 2 | Prior Labs | ✓ |
| 9 | Sundial-Base | 44.4 | 33.4 | 33.99 | 1 | 0 | Tsinghua University | ✓ |
| 10 | Stat. Ensemble | 43.8 | 20.2 | 624.45 | 0 | 11 | — | ✗ |
| 11 | AutoARIMA | 39.0 | 20.6 | 120.16 | 0 | 10 | — | ✗ |
| 12 | AutoETS | 32.6 | -26.8 | 16.24 | 0 | 3 | — | ✗ |
| 13 | AutoTheta | 25.9 | 5.5 | 9.27 | 0 | 0 | — | ✗ |
| 14 | Seasonal Naive | 19.0 | 0.0 | 2.32 | 0 | 0 | — | ✗ |
| 15 | Naive | 13.2 | -45.4 | 2.24 | 0 | 0 | — | ✗ |
| 16 | Drift | 9.0 | -45.8 | 2.19 | 0 | 0 | — | ✗ |

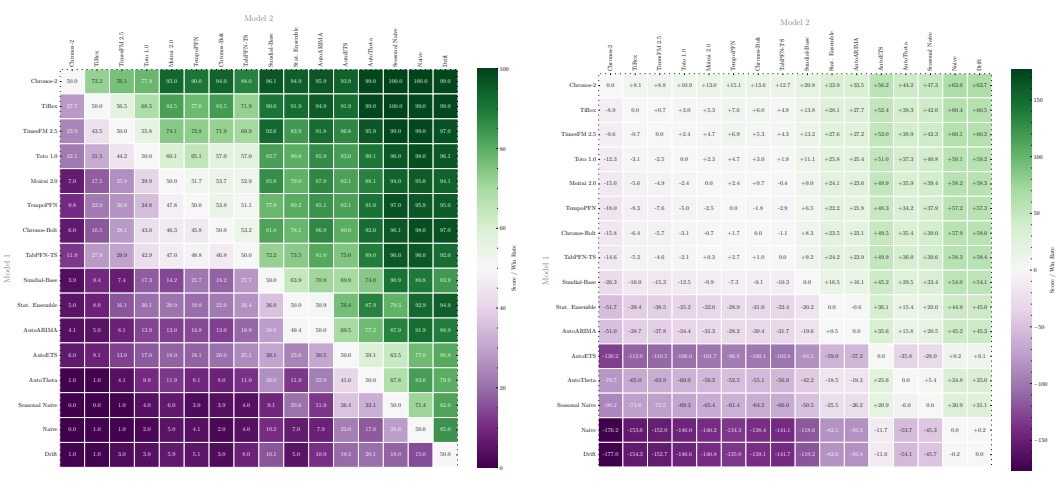

(a) Pairwise Win Rate (SQL) with 95% CIs  (b) Pairwise Skill Score (SQL) with 95% CIs

Figure 19: Head-to-head comparisons on FEV-Bench using **Scaled Quantile Loss (SQL)**. **(a) Win Rate:** Percentage of tasks where the row model achieves lower error than the column model (ties count as half-wins); values >50% indicate the row model is more accurate on average. **(b) Skill Score:** Average relative error reduction of the row model with respect to the column model; positive values indicate error reduction. Brackets indicate 95% confidence intervals estimated via 1000 bootstrap samples.

## E.4 FEATURE-SPACE ALIGNMENT OF REAL AND SYNTHETIC DATA MANIFOLD

To empirically validate that our synthetic pre-training corpus effectively spans the manifold of real-world time series dynamics, we conducted a feature-space analysis comparing our synthetic data against the real-world benchmarks used for evaluation (GIFT-Eval, FEV-Bench, and Chronos).

**Methodology.** We randomly sampled up to 100,000 time series from each of our synthetic generators and the real-world datasets. For each series, we extracted a comprehensive vector of statistical time-series characteristics (including autocorrelation, approximate entropy, trend strength, spikiness, and seasonality metrics) using the `tsfresh` library (Christ et al., 2018). To visualize the relationship between these distributions, we standardized the feature vectors and projected them into a latent space using Uniform Manifold Approximation and Projection (UMAP) (McInnes et al., 2018).

**Analysis.** Figure 20 presents the resulting embeddings in both 2D and 3D projections. The fact that real-world clusters are visible directly *on top* of the synthetic data confirms significant distributional overlap. The synthetic generators do not collapse into a single mode, but instead cover a vast region of the feature space, effectively "underpainting" the real-world benchmarks. This visual evidence supports our hypothesis that a diverse mixture of structurally distinct generators and data augmenta-

tions collectively covers the complex distribution of real-world temporal dynamics, enabling robust zero-shot transfer.

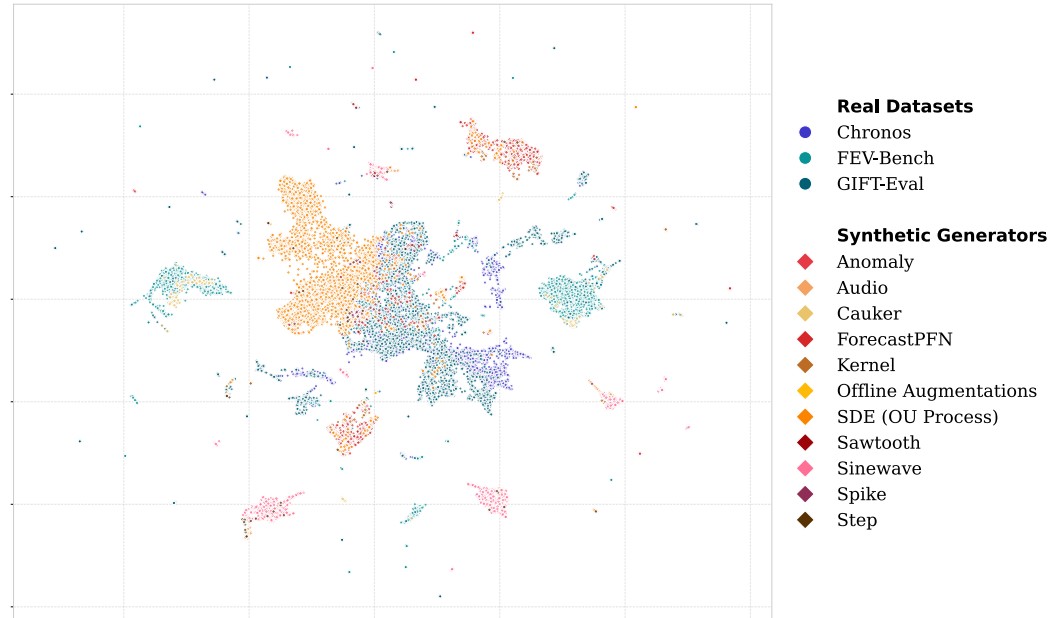

(a) 2D Projection ($N_{neigh} = 5$, $d_{min} = 0.1$, Silhouette Score: -0.054)

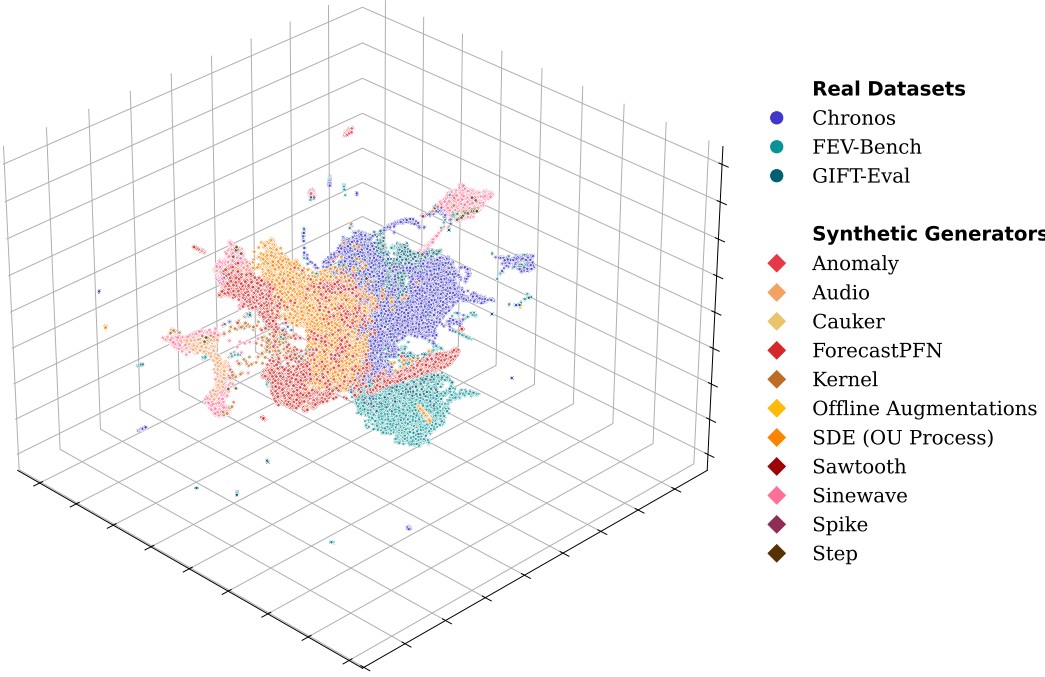

(b) 3D Projection ($N_{neigh} = 5$, $d_{min} = 0.4$, Silhouette Score: -0.001)

Figure 20: **Feature-Space Distribution of Real vs. Synthetic Data.** UMAP projections of time-series features extracted via `tsfresh`. Real-world benchmarks (GIFT-Eval, FEV-Bench, Chronos) are shown in **cold colors** (Blue/Teal) in the foreground, while our synthetic generators are shown in **warm colors** (Red/Orange/Yellow) in the background.

1512
1513

## E.5 QUALITATIVE COMPARISON ON THE GIFT-EVAL BENCHMARK

1514 This section presents qualitative Gift-Eval forecasts in Figure 21, showing the full history (left)
1515 alongside zoomed-in predictions for **TempoPFN**, **TiRex**, and **TabPFN-TS**. Note that evaluation
1516 context lengths vary: **TiRex** uses the full history, **TabPFN-TS** uses 4096 steps, and **TempoPFN**
1517 uses 3072.

1518
1519
1520

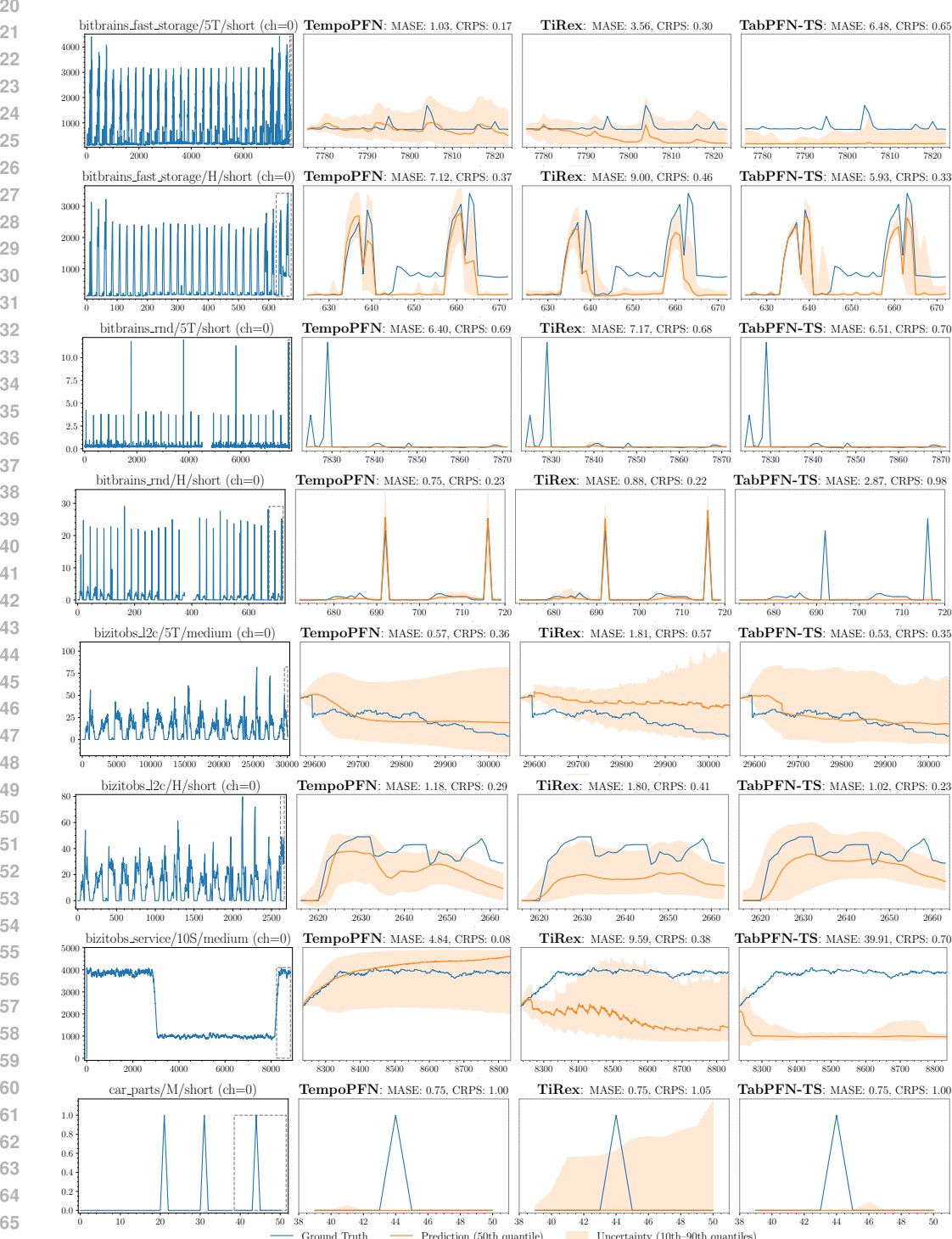

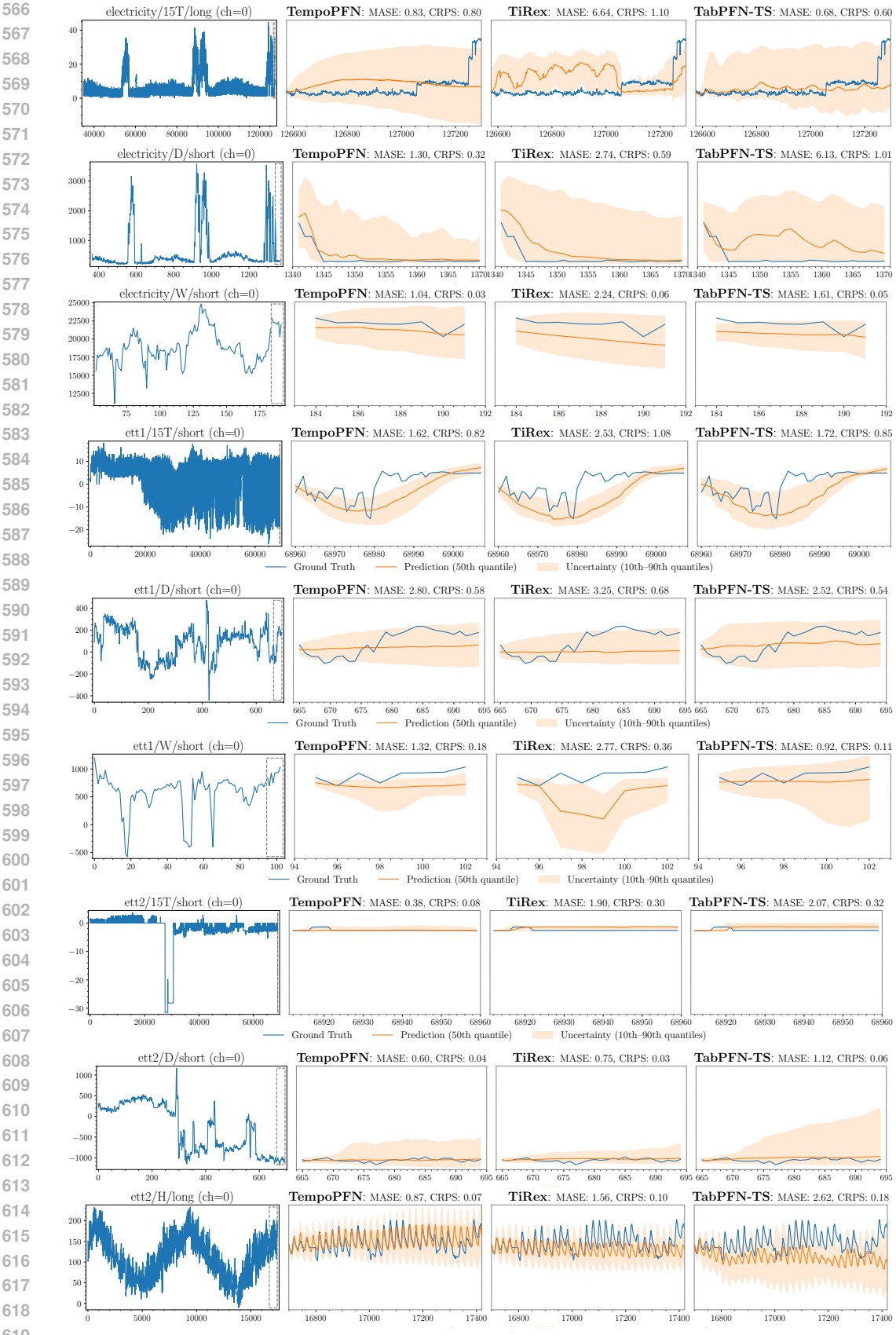

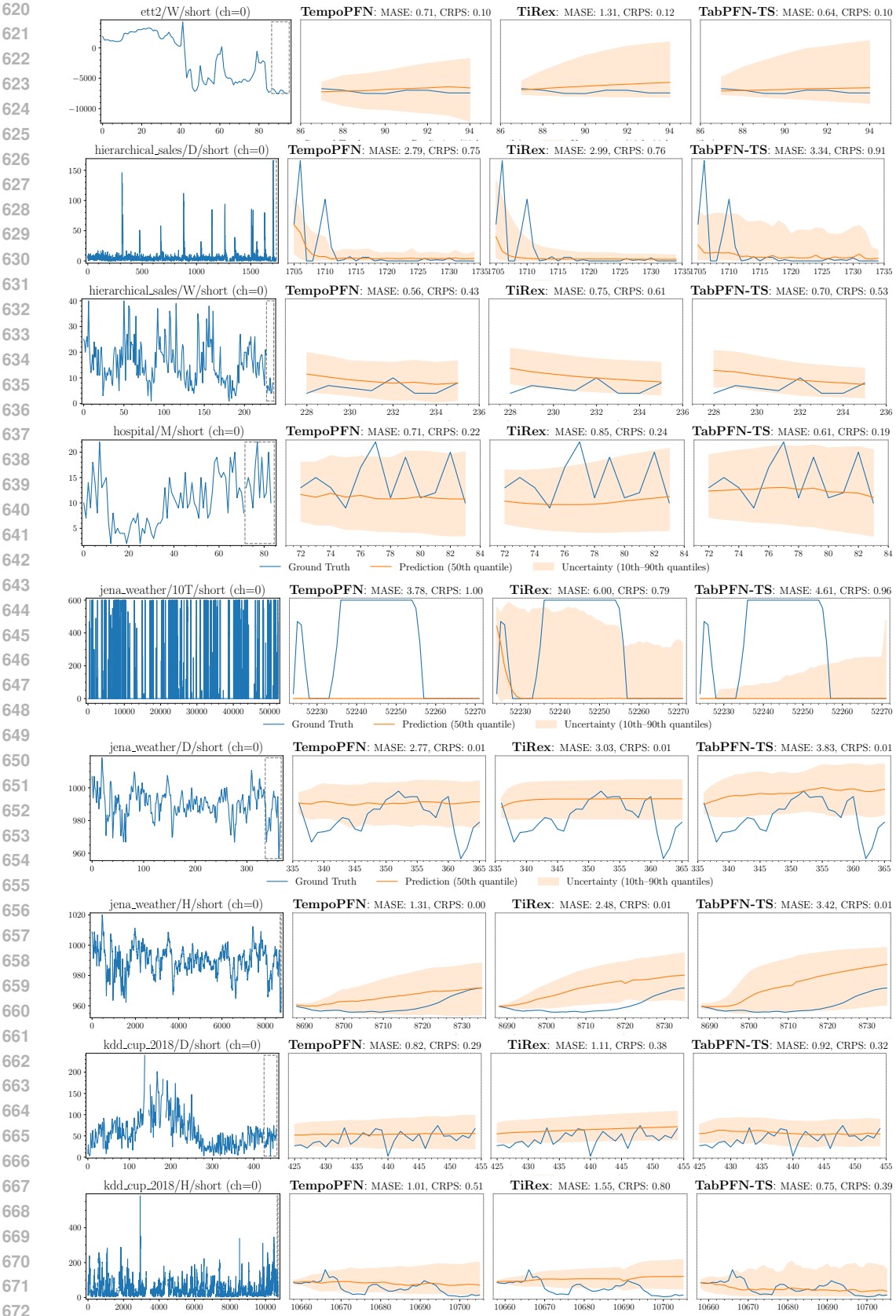

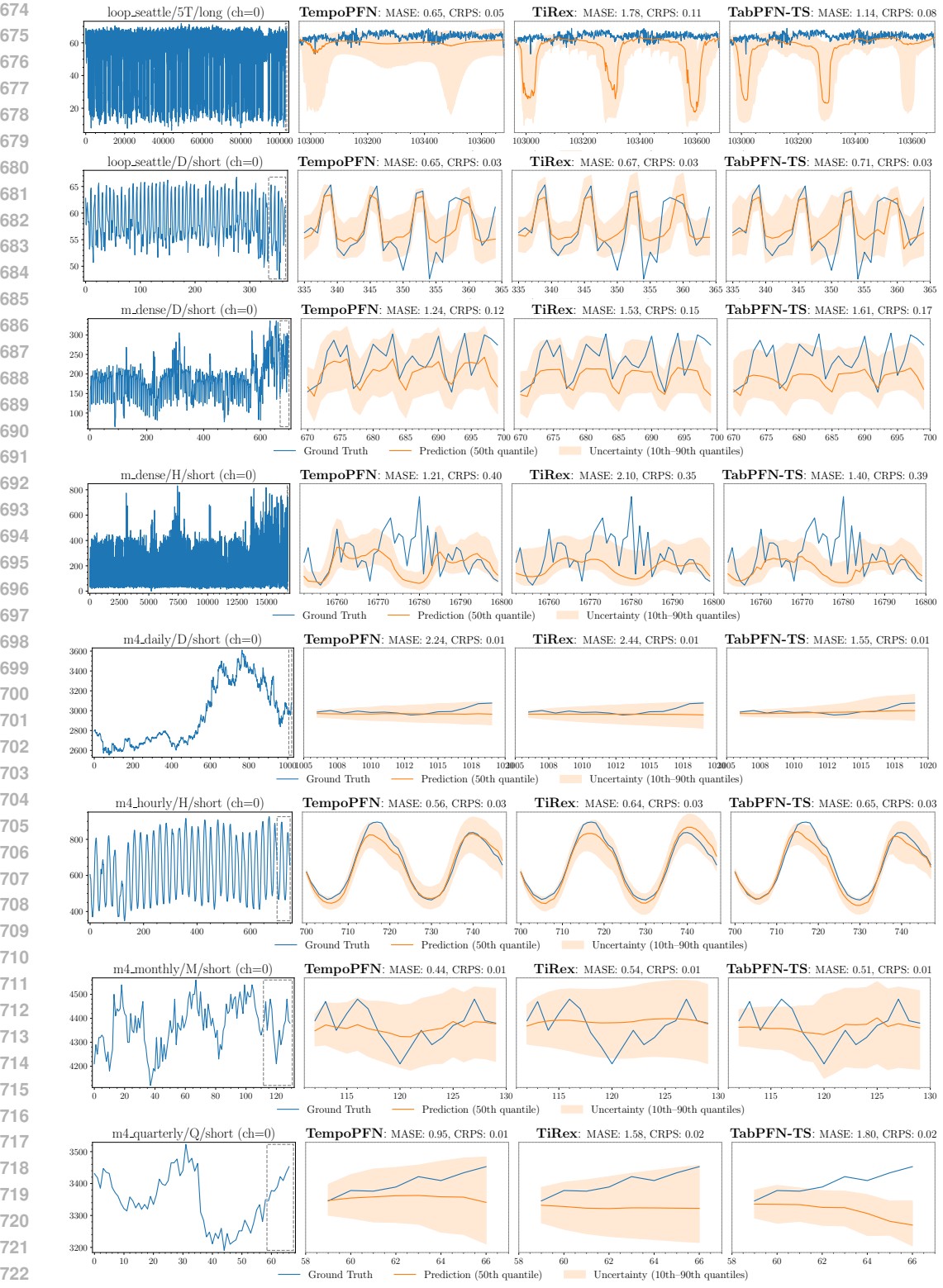

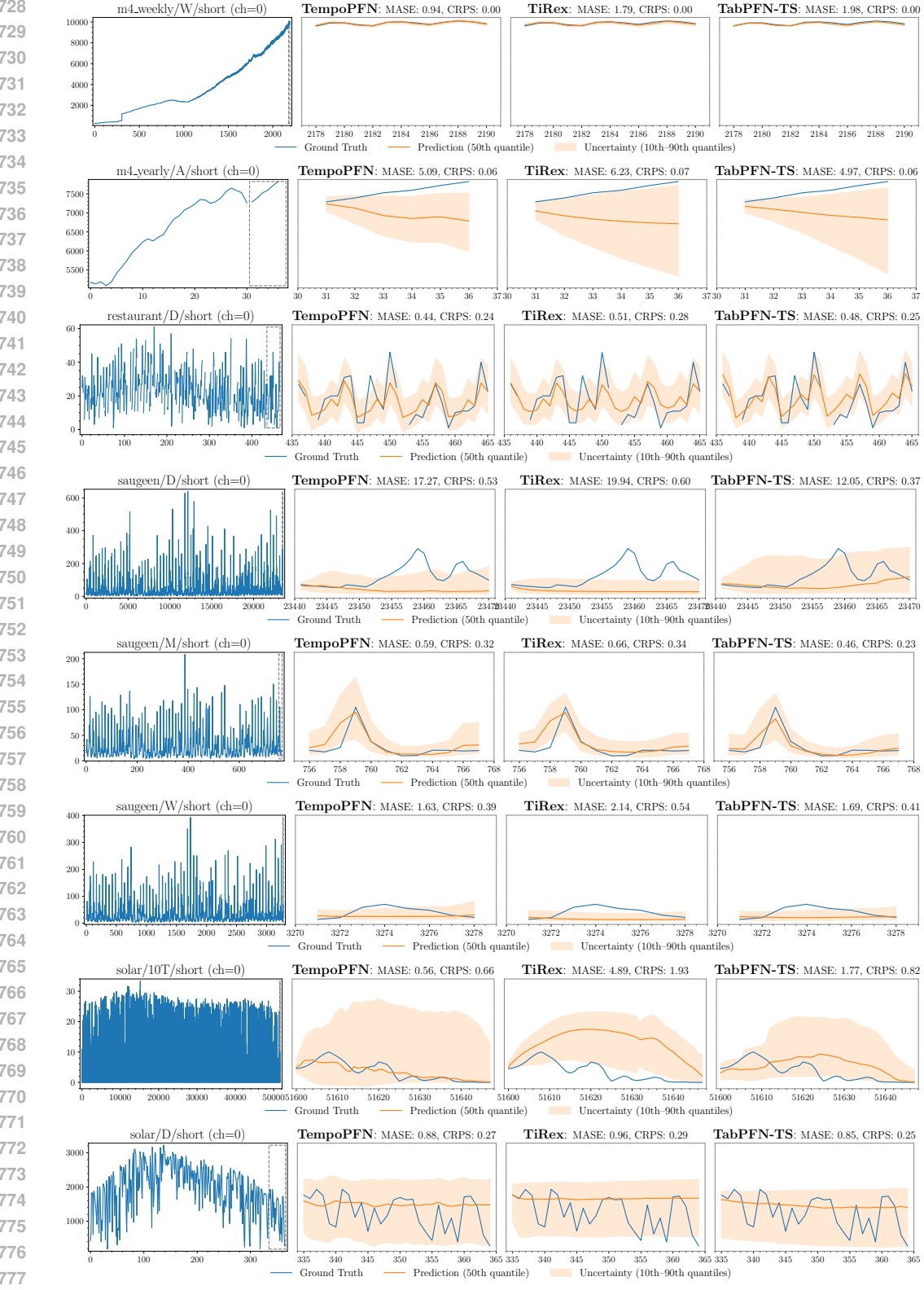

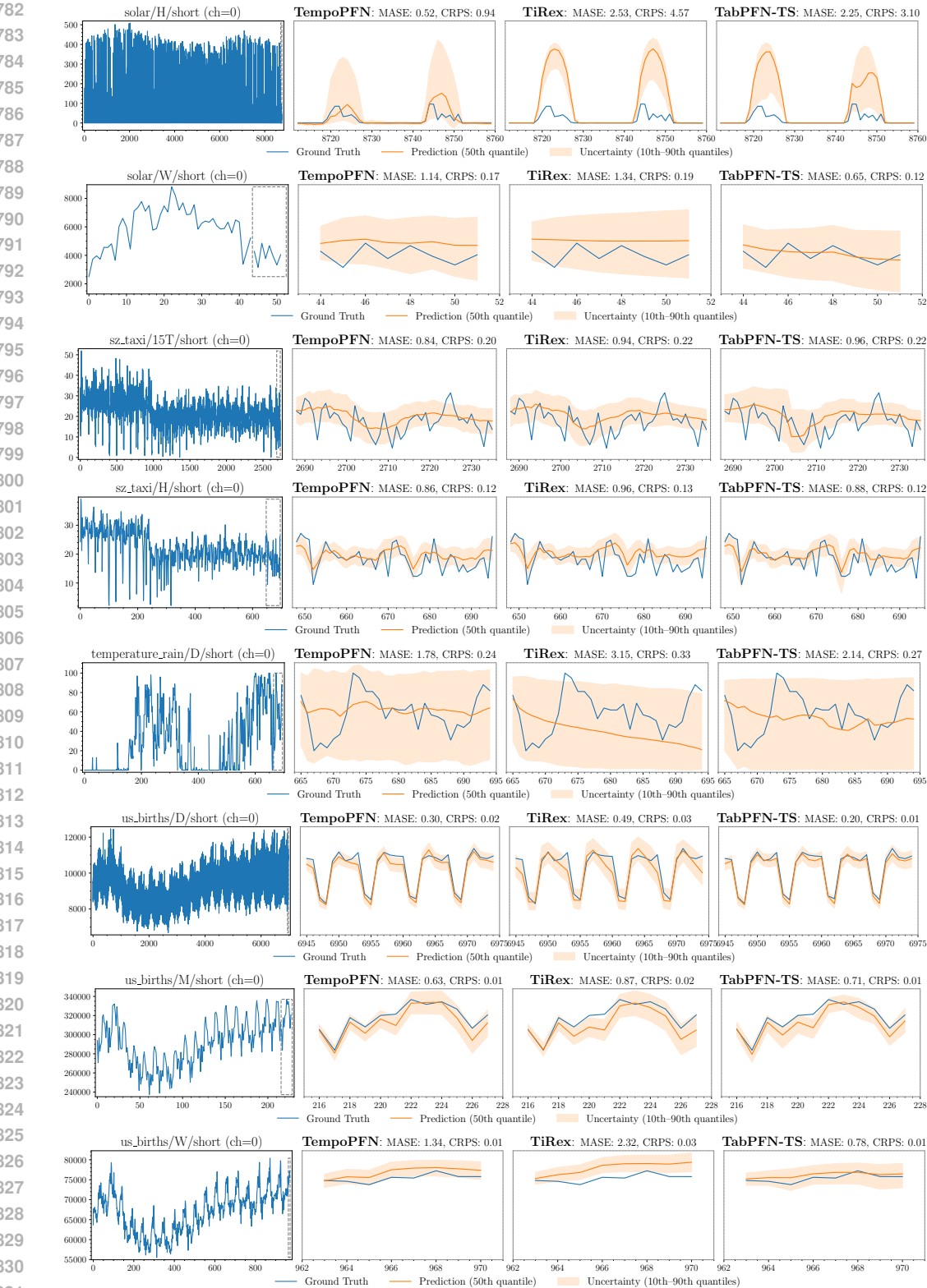

Figure 21: Qualitative comparison between TempoPFN, TiRex and TabPFN-TS on the GIFT-Eval Benchmark. (Left) Total context with prediction window in dashed grey box. (Right) Predictions between TempoPFN, TiRex, and TabPFN-TS.

## E.6 Quantitative Comparison on the Gift-Eval Benchmark

Table 14: Normalized CRPS (Continuous Ranked Probability Score) for various zero-shot models on the GiftEval benchmark. Scores are normalized against a seasonal naive baseline, with values less than 1.0 signifying superior probabilistic forecasts. The top two performing models are highlighted.

| Dataset | TempoPFN | TiRex | FlowState-9.1M | Toto_Open_Base_1.0 | TabPFN-TS | YingLong_50m | Chronos Bolt B | TTM-R2-Finetuned | Moirai L 1.1 | Moirai B 1.1 |
|---|---|---|---|---|---|---|---|---|---|---|
| bitbrains_fast_storage/5T/long | 0.555 | 0.558 | 0.701 | 0.568 | 0.752 | 0.586 | 0.635 | 0.796 | 0.608 | 0.622 |
| bitbrains_fast_storage/5T/medium | 0.516 | 0.516 | 0.651 | 0.525 | 0.792 | 0.537 | 0.631 | 0.690 | 0.531 | 0.553 |
| bitbrains_fast_storage/5T/short | 0.334 | 0.334 | 0.422 | 0.306 | 0.547 | 0.343 | 0.375 | 0.465 | 0.340 | 0.341 |
| bitbrains_fast_storage/H/short | 0.632 | 0.683 | 0.684 | 0.609 | 0.655 | 0.610 | 0.757 | 1.007 | 0.632 | 0.600 |
| bitbrains_rnd/5T/long | 0.602 | 0.541 | 0.619 | 0.501 | 0.697 | 0.563 | 0.643 | 0.648 | 0.577 | 0.566 |
| bitbrains_rnd/5T/medium | 0.530 | 0.500 | 0.619 | 0.537 | 0.701 | 0.546 | 0.517 | 0.670 | 0.508 | 0.527 |
| bitbrains_rnd/5T/short | 0.477 | 0.367 | 0.420 | 0.362 | 0.552 | 0.390 | 0.398 | 0.424 | 0.379 | 0.405 |
| bitbrains_rnd/H/short | 0.523 | 0.522 | 0.535 | 0.477 | 0.597 | 0.546 | 0.502 | 0.718 | 0.455 | 0.467 |
| bizitobs_application/10S/long | 1.041 | 1.142 | 1.152 | 1.148 | 1.062 | 1.315 | 2.383 | 1.261 | 2.053 | 2.624 |
| bizitobs_application/10S/medium | 0.560 | 0.944 | 0.985 | 0.802 | 0.955 | 1.147 | 2.425 | 1.138 | 1.958 | 2.436 |
| bizitobs_application/10S/short | 0.293 | 0.377 | 0.389 | 0.333 | 0.426 | 0.506 | 1.549 | 0.638 | 1.099 | 0.941 |
| bizitobs_l2c/5T/long | 0.885 | 0.917 | 0.355 | 0.821 | 0.472 | 0.848 | 1.138 | 0.373 | 0.783 | 0.854 |
| bizitobs_l2c/5T/medium | 0.681 | 0.684 | 0.411 | 0.606 | 0.502 | 0.690 | 0.856 | 0.475 | 0.788 | 0.730 |
| bizitobs_l2c/5T/short | 0.276 | 0.303 | 0.323 | 0.265 | 0.321 | 0.298 | 0.284 | 0.263 | 0.303 | 0.297 |
| bizitobs_l2c/H/long | 0.333 | 0.297 | 0.263 | 0.392 | 0.311 | 0.401 | 0.295 | 0.736 | 0.638 | 0.526 |
| bizitobs_l2c/H/medium | 0.301 | 0.284 | 0.245 | 0.394 | 0.263 | 0.396 | 0.281 | 0.746 | 0.685 | 0.761 |
| bizitobs_l2c/H/short | 0.381 | 0.413 | 0.362 | 0.382 | 0.402 | 0.516 | 0.363 | 0.516 | 1.073 | 0.946 |
| bizitobs_service/10S/long | 1.043 | 1.001 | 1.001 | 0.955 | 0.965 | 1.220 | 2.111 | 1.095 | 1.946 | 2.151 |
| bizitobs_service/10S/medium | 0.393 | 0.639 | 0.562 | 0.574 | 0.860 | 0.981 | 2.022 | 0.896 | 1.455 | 1.892 |
| bizitobs_service/10S/short | 0.289 | 0.327 | 0.318 | 0.286 | 0.482 | 0.444 | 1.267 | 0.354 | 0.788 | 1.053 |
| car_parts/M/short | 0.590 | 0.576 | 0.585 | 0.522 | 0.563 | 0.817 | 0.578 | 0.641 | 0.685 | 0.580 |
| covid_deaths/D/short | 0.234 | 0.273 | 0.339 | 0.215 | 0.324 | 0.468 | 0.371 | 0.287 | 0.362 | 0.346 |
| electricity/15T/long | 1.010 | 0.656 | 0.666 | 0.761 | 0.716 | 0.744 | 0.748 | 0.690 | 0.878 | 1.022 |
| electricity/15T/medium | 0.907 | 0.655 | 0.671 | 0.762 | 0.734 | 0.754 | 0.734 | 0.683 | 0.913 | 0.940 |
| electricity/15T/short | 0.682 | 0.503 | 0.562 | 0.603 | 0.587 | 0.565 | 0.496 | 0.497 | 0.776 | 0.728 |
| electricity/D/short | 0.639 | 0.525 | 0.529 | 0.566 | 0.603 | 0.560 | 0.531 | 0.560 | 0.666 | 0.583 |
| electricity/H/long | 0.652 | 0.607 | 0.571 | 0.544 | 0.706 | 0.682 | 0.637 | 0.587 | 0.671 | 0.557 |
| electricity/H/medium | 0.685 | 0.617 | 0.603 | 0.589 | 0.688 | 0.677 | 0.636 | 0.615 | 0.685 | 0.645 |
| electricity/H/short | 0.730 | 0.571 | 0.632 | 0.652 | 0.677 | 0.744 | 0.602 | 0.643 | 0.732 | 0.711 |
| electricity/W/short | 0.611 | 0.459 | 0.439 | 0.641 | 0.550 | 0.636 | 0.477 | 0.605 | 0.625 | 0.775 |
| ett1/15T/long | 0.847 | 0.721 | 0.676 | 0.738 | 0.763 | 0.712 | 0.875 | 0.750 | 1.052 | 0.803 |
| ett1/15T/medium | 0.864 | 0.777 | 0.713 | 0.810 | 0.785 | 0.763 | 0.874 | 0.847 | 1.064 | 1.008 |
| ett1/15T/short | 0.753 | 0.662 | 0.670 | 0.670 | 0.691 | 0.704 | 0.654 | 0.724 | 0.936 | 0.800 |
| ett1/D/short | 0.658 | 0.678 | 0.685 | 0.695 | 0.729 | 0.664 | 0.703 | 0.838 | 0.700 | 0.737 |
| ett1/H/long | 0.598 | 0.551 | 0.573 | 0.566 | 0.626 | 0.550 | 0.660 | 0.597 | 0.628 | 0.609 |
| ett1/H/medium | 0.634 | 0.578 | 0.569 | 0.584 | 0.651 | 0.578 | 0.696 | 0.628 | 0.621 | 0.648 |
| ett1/H/short | 0.775 | 0.747 | 0.733 | 0.806 | 0.807 | 0.746 | 0.753 | 0.814 | 0.786 | 0.820 |
| ett1/W/short | 0.827 | 1.000 | 0.786 | 0.843 | 0.911 | 0.931 | 0.949 | 0.906 | 0.834 | 0.837 |
| ett2/15T/long | 0.768 | 0.721 | 0.680 | 0.665 | 0.761 | 0.702 | 0.839 | 0.718 | 0.866 | 1.031 |
| ett2/15T/medium | 0.776 | 0.744 | 0.703 | 0.748 | 0.808 | 0.741 | 0.889 | 0.776 | 0.846 | 0.878 |
| ett2/15T/short | 0.711 | 0.685 | 0.683 | 0.706 | 0.756 | 0.696 | 0.691 | 0.662 | 0.833 | 0.805 |
| ett2/D/short | 0.783 | 0.594 | 0.674 | 0.727 | 0.821 | 0.613 | 0.610 | 0.553 | 0.611 | 0.618 |
| ett2/H/long | 0.498 | 0.549 | 0.536 | 0.521 | 0.669 | 0.514 | 0.562 | 0.504 | 0.601 | 0.529 |
| ett2/H/medium | 0.567 | 0.568 | 0.560 | 0.547 | 0.648 | 0.548 | 0.616 | 0.560 | 0.634 | 0.537 |
| ett2/H/short | 0.730 | 0.744 | 0.710 | 0.725 | 0.821 | 0.730 | 0.713 | 0.742 | 0.775 | 0.807 |
| ett2/W/short | 0.687 | 0.645 | 0.779 | 0.792 | 0.738 | 0.817 | 0.660 | 0.702 | 0.815 | 0.652 |
| hierarchical_sales/D/short | 0.336 | 0.327 | 0.336 | 0.328 | 0.341 | 0.340 | 0.332 | 0.349 | 0.334 | 0.331 |
| hierarchical_sales/W/short | 0.412 | 0.416 | 0.408 | 0.428 | 0.415 | 0.467 | 0.424 | 0.435 | 0.431 | 0.429 |
| hospital/M/short | 0.840 | 0.830 | 0.766 | 0.838 | 0.860 | 0.926 | 0.913 | 0.841 | 0.821 | 0.810 |
| jena_weather/10T/long | 0.240 | 0.220 | 0.214 | 0.210 | 0.224 | 0.254 | 0.269 | 0.286 | 0.325 | 0.297 |
| jena_weather/10T/medium | 0.248 | 0.239 | 0.234 | 0.231 | 0.254 | 0.272 | 0.270 | 0.324 | 0.338 | 0.320 |
| jena_weather/10T/short | 0.204 | 0.200 | 0.185 | 0.172 | 0.219 | 0.238 | 0.210 | 0.285 | 0.331 | 0.345 |
| jena_weather/D/short | 0.239 | 0.218 | 0.226 | 0.240 | 0.224 | 0.233 | 0.214 | 0.343 | 0.243 | 0.236 |
| jena_weather/H/long | 0.147 | 0.134 | 0.159 | 0.136 | 0.245 | 0.145 | 0.147 | 0.449 | 0.145 | 0.156 |
| jena_weather/H/medium | 0.164 | 0.153 | 0.154 | 0.154 | 0.169 | 0.165 | 0.158 | 0.204 | 0.168 | 0.166 |
| jena_weather/H/short | 0.273 | 0.266 | 0.272 | 0.274 | 0.273 | 0.284 | 0.274 | 0.403 | 0.291 | 0.284 |
| kdd_cup_2018/D/short | 0.561 | 0.577 | 0.565 | 0.574 | 0.537 | 0.551 | 0.552 | 0.593 | 0.565 | 0.557 |
| kdd_cup_2018/H/long | 0.476 | 0.353 | 0.489 | 0.488 | 0.510 | 0.475 | 0.320 | 0.507 | 0.404 | 0.446 |
| kdd_cup_2018/H/medium | 0.563 | 0.437 | 0.576 | 0.582 | 0.593 | 0.553 | 0.397 | 0.597 | 0.510 | 0.581 |
| kdd_cup_2018/H/short | 0.701 | 0.498 | 0.697 | 0.736 | 0.763 | 0.694 | 0.450 | 0.758 | 0.661 | 0.710 |
| loop_seattle/5T/long | 0.867 | 0.680 | 0.582 | 0.602 | 0.710 | 0.790 | 1.015 | 0.659 | 0.387 | 0.407 |
| loop_seattle/5T/medium | 0.910 | 0.692 | 0.589 | 0.617 | 0.741 | 0.848 | 0.989 | 0.661 | 0.328 | 0.386 |
| loop_seattle/5T/short | 0.701 | 0.606 | 0.618 | 0.598 | 0.650 | 0.710 | 0.675 | 0.634 | 0.512 | 0.565 |
| loop_seattle/D/short | 0.417 | 0.409 | 0.407 | 0.430 | 0.418 | 0.425 | 0.424 | 0.441 | 0.438 | 0.428 |
| loop_seattle/H/long | 0.405 | 0.328 | 0.338 | 0.345 | 0.334 | 0.364 | 0.406 | 0.360 | 0.397 | 0.425 |
| loop_seattle/H/medium | 0.451 | 0.400 | 0.420 | 0.398 | 0.412 | 0.430 | 0.470 | 0.429 | 0.433 | 0.492 |
| loop_seattle/H/short | 0.669 | 0.569 | 0.583 | 0.609 | 0.608 | 0.609 | 0.624 | 0.642 | 0.634 | 0.709 |
| m4_daily/D/short | 1.048 | 0.827 | 0.998 | 0.898 | 0.928 | 0.975 | 0.865 | 0.970 | 1.244 | 1.646 |
| m4_hourly/H/short | 0.759 | 0.536 | 0.545 | 0.921 | 0.788 | 0.641 | 0.674 | 0.919 | 0.527 | 0.591 |
| m4_monthly/M/short | 0.749 | 0.756 | 0.759 | 0.795 | 0.768 | 0.860 | 0.769 | 0.821 | 0.780 | 0.769 |
| m4_quarterly/Q/short | 0.763 | 0.753 | 0.770 | 0.791 | 0.798 | 0.892 | 0.786 | 0.809 | 0.749 | 0.749 |
| m4_weekly/W/short | 0.689 | 0.589 | 0.602 | 0.806 | 0.608 | 0.681 | 0.627 | 0.722 | 0.764 | 0.792 |
| m4_yearly/A/short | 0.872 | 0.857 | 0.780 | 0.887 | 0.858 | 1.096 | 0.886 | 0.864 | 0.758 | 0.766 |
| m_dense/D/short | 0.291 | 0.292 | 0.308 | 0.328 | 0.269 | 0.354 | 0.304 | 0.311 | 0.420 | 0.458 |
| m_dense/H/long | 0.438 | 0.291 | 0.289 | 0.307 | 0.394 | 0.417 | 0.405 | 0.306 | 0.272 | 0.291 |
| m_dense/H/medium | 0.459 | 0.318 | 0.304 | 0.321 | 0.424 | 0.415 | 0.417 | 0.322 | 0.297 | 0.326 |
| m_dense/H/short | 0.627 | 0.472 | 0.478 | 0.540 | 0.564 | 0.573 | 0.455 | 0.514 | 0.466 | 0.509 |
| restaurant/D/short | 0.381 | 0.377 | 0.378 | 0.439 | 0.388 | 0.401 | 0.390 | 0.397 | 0.399 | 0.393 |
| saugeen/D/short | 0.642 | 0.669 | 0.562 | 0.604 | 0.638 | 0.627 | 0.578 | 0.694 | 0.694 | 0.605 |
| saugeen/M/short | 0.697 | 0.717 | 0.664 | 0.671 | 0.621 | 0.707 | 0.665 | 0.764 | 0.728 | 0.782 |
| saugeen/W/short | 0.546 | 0.477 | 0.486 | 0.531 | 0.539 | 0.512 | 0.494 | 0.606 | 0.586 | 0.576 |
| solar/10T/long | 0.748 | 0.480 | 0.501 | 0.523 | 0.491 | 0.830 | 0.657 | 0.722 | 1.144 | 1.340 |
| solar/10T/medium | 0.664 | 0.549 | 0.544 | 0.539 | 0.498 | 0.819 | 0.666 | 0.759 | 1.140 | 1.270 |
| solar/10T/short | 0.538 | 0.549 | 0.605 | 0.629 | 0.533 | 0.663 | 0.595 | 0.633 | 0.693 | 0.714 |
| solar/D/short | 0.501 | 0.506 | 0.497 | 0.518 | 0.481 | 0.516 | 0.514 | 0.541 | 0.522 | 0.528 |
| solar/H/long | 0.339 | 0.237 | 0.309 | 0.307 | 0.325 | 0.326 | 0.376 | 0.563 | 0.322 | 0.334 |
| solar/H/medium | 0.409 | 0.281 | 0.365 | 0.350 | 0.331 | 0.380 | 0.390 | 0.591 | 0.366 | 0.350 |
| solar/H/short | 0.573 | 0.457 | 0.563 | 0.555 | 0.567 | 0.576 | 0.504 | 0.691 | 0.563 | 0.571 |
| solar/W/short | 0.763 | 0.779 | 0.583 | 0.887 | 0.572 | 1.452 | 0.632 | 0.817 | 1.016 | 1.120 |
| sz_taxi/15T/long | 0.497 | 0.460 | 0.483 | 0.472 | 0.567 | 0.477 | 0.581 | 0.558 | 0.498 | 0.488 |
| sz_taxi/15T/medium | 0.558 | 0.533 | 0.548 | 0.542 | 0.608 | 0.549 | 0.644 | 0.567 | 0.567 | 0.556 |
| sz_taxi/15T/short | 0.669 | 0.649 | 0.656 | 0.657 | 0.676 | 0.658 | 0.654 | 0.682 | 0.696 | 0.690 |
| sz_taxi/H/short | 0.650 | 0.634 | 0.647 | 0.642 | 0.656 | 0.640 | 0.636 | 0.662 | 0.683 | 0.669 |
| temperature_rain/D/short | 0.451 | 0.434 | 0.434 | 0.441 | 0.449 | 0.462 | 0.424 | 0.518 | 0.378 | 0.422 |
| us_births/D/short | 0.169 | 0.179 | 0.198 | 0.217 | 0.133 | 0.258 | 0.217 | 0.171 | 0.224 | 0.223 |
| us_births/M/short | 0.814 | 0.894 | 0.887 | 0.754 | 0.909 | 0.766 | 1.156 | 0.680 | 0.939 | 0.886 |
| us_births/W/short | 0.553 | 0.654 | 0.616 | 0.750 | 0.550 | 0.743 | 0.668 | 0.739 | 0.922 | 0.897 |

Table 15: Normalized MASE scores of different zero-shot models on the GiftEval benchmark. Scores are relative to a seasonal naive baseline, where values below 1.0 indicate better performance. The models achieving the best and second-best scores are highlighted. )

| Dataset | TempoPFN | TiRex | FlowState-9.1M | Toto-Open-Base-1.0 | TTM-R2-Finetuned | TabPFN-TS | Chronos Bolt B | YingLong-50m | Moirai L 1.1 | Moirai B 1.1 |
|---|---|---|---|---|---|---|---|---|---|---|
| bitbrains_fast_storage/5T/long | 0.846 | 0.808 | 0.913 | 0.789 | 0.827 | 1.014 | 0.834 | 0.886 | 0.840 | 0.851 |
| bitbrains_fast_storage/5T/medium | 0.896 | 0.815 | 1.033 | 0.807 | 0.876 | 1.072 | 0.871 | 0.889 | 0.836 | 0.860 |
| bitbrains_fast_storage/5T/short | 0.716 | 0.609 | 0.883 | 0.591 | 0.648 | 0.879 | 0.662 | 0.719 | 0.728 | 0.697 |
| bitbrains_fast_storage/H/short | 0.910 | 0.825 | 0.851 | 0.728 | 0.927 | 0.912 | 0.824 | 0.856 | 0.839 | 0.909 |
| bitbrains_rnd/5T/long | 1.005 | 0.954 | 1.007 | 0.953 | 1.001 | 1.107 | 0.970 | 1.002 | 0.977 | 0.985 |
| bitbrains_rnd/5T/medium | 1.001 | 0.967 | 1.009 | 0.972 | 0.997 | 1.064 | 0.979 | 1.004 | 0.982 | 0.997 |
| bitbrains_rnd/5T/short | 0.940 | 0.845 | 0.996 | 0.837 | 0.889 | 1.030 | 0.865 | 0.915 | 0.888 | 0.923 |
| bitbrains_rnd/H/short | 0.999 | 0.971 | 0.982 | 0.934 | 1.004 | 1.106 | 0.977 | 0.972 | 0.982 | 1.005 |
| bizitobs_application/10S/long | 1.086 | 1.150 | 1.042 | 1.021 | 1.187 | 0.965 | 3.270 | 1.518 | 2.445 | 4.210 |
| bizitobs_application/10S/medium | 0.800 | 1.035 | 0.962 | 0.856 | 1.134 | 0.925 | 3.612 | 1.516 | 2.746 | 4.756 |
| bizitobs_application/10S/short | 0.466 | 0.571 | 0.597 | 0.556 | 0.734 | 0.563 | 2.468 | 0.841 | 2.011 | 2.373 |
| bizitobs_l2c/5T/long | 0.840 | 0.819 | 0.356 | 0.809 | 0.347 | 0.457 | 0.853 | 0.816 | 0.770 | 0.770 |
| bizitobs_l2c/5T/medium | 0.629 | 0.668 | 0.418 | 0.606 | 0.441 | 0.513 | 0.706 | 0.677 | 0.794 | 0.686 |
| bizitobs_l2c/5T/short | 0.278 | 0.299 | 0.317 | 0.262 | 0.250 | 0.311 | 0.282 | 0.292 | 0.289 | 0.295 |
| bizitobs_l2c/H/long | 0.478 | 0.425 | 0.384 | 0.559 | 0.844 | 0.466 | 0.390 | 0.554 | 0.891 | 0.764 |
| bizitobs_l2c/H/medium | 0.403 | 0.358 | 0.312 | 0.501 | 0.753 | 0.324 | 0.328 | 0.512 | 0.828 | 0.874 |
| bizitobs_l2c/H/short | 0.381 | 0.420 | 0.371 | 0.387 | 0.448 | 0.400 | 0.356 | 0.532 | 0.947 | 0.823 |
| bizitobs_service/10S/long | 1.216 | 1.104 | 1.011 | 0.953 | 1.086 | 0.999 | 3.875 | 1.715 | 3.167 | 4.447 |
| bizitobs_service/10S/medium | 1.076 | 0.946 | 0.866 | 0.820 | 0.953 | 0.928 | 3.768 | 1.605 | 2.931 | 4.536 |
| bizitobs_service/10S/short | 0.947 | 0.677 | 0.658 | 0.644 | 0.653 | 0.721 | 2.706 | 0.953 | 1.885 | 2.799 |
| car_parts/M/short | 0.700 | 0.698 | 0.744 | 0.675 | 0.698 | 0.706 | 0.712 | 1.057 | 0.752 | 0.695 |
| covid_deaths/D/short | 0.785 | 0.830 | 0.738 | 0.695 | 0.657 | 0.837 | 0.828 | 0.929 | 0.778 | 0.738 |
| electricity/15T/long | 1.170 | 0.759 | 0.778 | 0.897 | 0.747 | 0.812 | 0.801 | 0.863 | 1.126 | 1.134 |
| electricity/15T/medium | 1.027 | 0.726 | 0.744 | 0.858 | 0.714 | 0.773 | 0.749 | 0.825 | 1.121 | 1.156 |
| electricity/15T/short | 0.751 | 0.557 | 0.630 | 0.667 | 0.530 | 0.670 | 0.545 | 0.628 | 0.996 | 0.897 |
| electricity/D/short | 0.787 | 0.716 | 0.722 | 0.747 | 0.730 | 0.751 | 0.729 | 0.744 | 0.760 | 0.755 |
| electricity/H/long | 0.896 | 0.802 | 0.795 | 0.811 | 0.781 | 0.861 | 0.812 | 0.869 | 0.892 | 0.827 |
| electricity/H/medium | 0.871 | 0.780 | 0.785 | 0.792 | 0.782 | 0.838 | 0.774 | 0.831 | 0.862 | 0.855 |
| electricity/H/short | 0.794 | 0.641 | 0.689 | 0.719 | 0.663 | 0.763 | 0.643 | 0.807 | 0.795 | 0.803 |
| electricity/W/short | 0.751 | 0.691 | 0.653 | 0.857 | 0.729 | 0.740 | 0.707 | 0.832 | 0.857 | 0.919 |
| ett1/15T/long | 1.050 | 0.871 | 0.850 | 0.898 | 0.870 | 0.939 | 0.954 | 0.869 | 1.176 | 0.940 |
| ett1/15T/medium | 0.995 | 0.868 | 0.833 | 0.909 | 0.902 | 0.919 | 0.893 | 0.865 | 1.094 | 1.044 |
| ett1/15T/short | 0.830 | 0.745 | 0.737 | 0.742 | 0.751 | 0.794 | 0.728 | 0.769 | 0.990 | 0.883 |
| ett1/D/short | 0.918 | 0.952 | 0.916 | 0.933 | 1.056 | 0.922 | 0.940 | 0.953 | 0.984 | 0.978 |
| ett1/H/long | 0.937 | 0.886 | 0.906 | 0.926 | 0.941 | 0.996 | 0.916 | 0.909 | 0.981 | 0.933 |
| ett1/H/medium | 0.860 | 0.785 | 0.782 | 0.811 | 0.819 | 0.896 | 0.877 | 0.808 | 0.855 | 0.861 |
| ett1/H/short | 0.875 | 0.849 | 0.834 | 0.885 | 0.869 | 0.908 | 0.847 | 0.850 | 0.875 | 0.906 |
| ett1/W/short | 0.851 | 1.003 | 0.820 | 0.893 | 0.902 | 0.938 | 0.959 | 0.980 | 0.854 | 0.871 |
| ett2/15T/long | 1.014 | 0.907 | 0.855 | 0.860 | 0.918 | 0.965 | 0.928 | 0.908 | 1.126 | 1.284 |
| ett2/15T/medium | 0.923 | 0.859 | 0.806 | 0.878 | 0.872 | 0.933 | 0.877 | 0.865 | 1.008 | 1.046 |
| ett2/15T/short | 0.731 | 0.695 | 0.707 | 0.736 | 0.681 | 0.788 | 0.718 | 0.724 | 0.937 | 0.899 |
| ett2/D/short | 1.648 | 0.917 | 1.053 | 1.161 | 0.868 | 1.029 | 0.951 | 0.943 | 1.036 | 0.942 |
| ett2/H/long | 0.879 | 1.008 | 0.972 | 0.955 | 0.919 | 1.280 | 0.918 | 0.940 | 1.134 | 0.993 |
| ett2/H/medium | 0.843 | 0.846 | 0.838 | 0.821 | 0.831 | 1.008 | 0.830 | 0.816 | 0.953 | 0.832 |
| ett2/H/short | 0.809 | 0.809 | 0.782 | 0.796 | 0.811 | 0.894 | 0.794 | 0.815 | 0.848 | 0.874 |
| ett2/W/short | 1.150 | 1.021 | 1.256 | 1.267 | 1.009 | 0.983 | 0.949 | 1.221 | 1.683 | 1.093 |
| hierarchical_sales/D/short | 0.662 | 0.653 | 0.660 | 0.648 | 0.666 | 0.669 | 0.655 | 0.677 | 0.657 | 0.657 |
| hierarchical_sales/W/short | 0.711 | 0.704 | 0.695 | 0.726 | 0.709 | 0.713 | 0.715 | 0.757 | 0.731 | 0.729 |
| hospital/M/short | 0.834 | 0.829 | 0.824 | 0.851 | 0.815 | 0.830 | 0.860 | 0.899 | 0.834 | 0.842 |
| jena_weather/10T/long | 0.851 | 0.836 | 0.868 | 0.833 | 1.008 | 0.876 | 0.862 | 0.843 | 1.040 | 1.001 |
| jena_weather/10T/medium | 0.867 | 0.842 | 0.873 | 0.835 | 1.001 | 0.874 | 0.852 | 0.888 | 0.969 | 0.994 |
| jena_weather/10T/short | 0.401 | 0.402 | 0.373 | 0.358 | 0.431 | 0.418 | 0.411 | 0.452 | 0.455 | 0.471 |
| jena_weather/D/short | 0.811 | 0.648 | 0.666 | 0.760 | 0.840 | 0.781 | 0.668 | 0.687 | 0.725 | 0.731 |
| jena_weather/H/long | 0.907 | 0.778 | 0.856 | 0.800 | 1.153 | 1.109 | 0.811 | 0.832 | 0.695 | 0.836 |
| jena_weather/H/medium | 0.934 | 0.949 | 0.965 | 0.847 | 0.969 | 1.225 | 0.841 | 0.991 | 1.003 | 0.919 |
| jena_weather/H/short | 0.739 | 0.716 | 0.728 | 0.752 | 0.802 | 0.759 | 0.741 | 0.740 | 0.809 | 0.766 |
| kdd_cup_2018/D/short | 0.803 | 0.820 | 0.817 | 0.810 | 0.800 | 0.784 | 0.799 | 0.782 | 0.802 | 0.802 |
| kdd_cup_2018/H/long | 0.776 | 0.554 | 0.789 | 0.779 | 0.752 | 0.819 | 0.512 | 0.756 | 0.649 | 0.719 |
| kdd_cup_2018/H/medium | 0.753 | 0.561 | 0.759 | 0.754 | 0.721 | 0.790 | 0.490 | 0.721 | 0.668 | 0.735 |
| kdd_cup_2018/H/short | 0.727 | 0.490 | 0.712 | 0.739 | 0.709 | 0.784 | 0.448 | 0.697 | 0.667 | 0.704 |
| loop_seattle/5T/long | 0.952 | 0.783 | 0.660 | 0.678 | 0.697 | 0.805 | 0.990 | 0.882 | 0.445 | 0.473 |
| loop_seattle/5T/medium | 1.004 | 0.796 | 0.671 | 0.697 | 0.701 | 0.836 | 0.985 | 0.949 | 0.390 | 0.453 |
| loop_seattle/5T/short | 0.856 | 0.745 | 0.756 | 0.737 | 0.732 | 0.784 | 0.823 | 0.864 | 0.638 | 0.703 |
| loop_seattle/D/short | 0.518 | 0.505 | 0.505 | 0.534 | 0.519 | 0.524 | 0.521 | 0.541 | 0.529 | 0.521 |
| loop_seattle/H/long | 0.727 | 0.583 | 0.595 | 0.610 | 0.585 | 0.599 | 0.644 | 0.636 | 0.679 | 0.744 |
| loop_seattle/H/medium | 0.721 | 0.634 | 0.662 | 0.628 | 0.623 | 0.661 | 0.688 | 0.684 | 0.675 | 0.770 |
| loop_seattle/H/short | 0.775 | 0.657 | 0.664 | 0.696 | 0.678 | 0.705 | 0.696 | 0.693 | 0.731 | 0.820 |
| m4_daily/D/short | 1.352 | 0.942 | 1.096 | 1.010 | 1.007 | 1.279 | 0.976 | 1.131 | 1.275 | 1.638 |
| m4_hourly/H/short | 0.695 | 0.589 | 0.613 | 0.721 | 0.862 | 0.619 | 0.701 | 0.808 | 0.743 | 0.814 |
| m4_monthly/M/short | 0.733 | 0.732 | 0.732 | 0.780 | 0.751 | 0.760 | 0.753 | 0.836 | 0.776 | 0.757 |
| m4_quarterly/Q/short | 0.733 | 0.725 | 0.720 | 0.766 | 0.730 | 0.764 | 0.764 | 0.875 | 0.712 | 0.712 |
| m4_weekly/W/short | 0.903 | 0.679 | 0.728 | 0.864 | 0.700 | 0.738 | 0.748 | 0.816 | 0.929 | 1.012 |
| m4_yearly/A/short | 0.863 | 0.864 | 0.746 | 0.857 | 0.819 | 0.834 | 0.884 | 1.098 | 0.749 | 0.759 |
| m_dense/D/short | 0.413 | 0.414 | 0.439 | 0.457 | 0.438 | 0.406 | 0.429 | 0.498 | 0.573 | 0.659 |
| m_dense/H/long | 0.751 | 0.491 | 0.504 | 0.528 | 0.497 | 0.692 | 0.635 | 0.666 | 0.471 | 0.497 |
| m_dense/H/medium | 0.660 | 0.464 | 0.441 | 0.464 | 0.452 | 0.633 | 0.561 | 0.567 | 0.436 | 0.468 |
| m_dense/H/short | 0.683 | 0.528 | 0.534 | 0.591 | 0.544 | 0.609 | 0.521 | 0.630 | 0.522 | 0.563 |
| restaurant/D/short | 0.684 | 0.674 | 0.678 | 0.779 | 0.695 | 0.694 | 0.696 | 0.713 | 0.711 | 0.700 |
| saugeen/D/short | 0.917 | 0.933 | 0.802 | 0.869 | 0.878 | 0.922 | 0.832 | 0.888 | 0.964 | 0.853 |
| saugeen/M/short | 0.809 | 0.790 | 0.766 | 0.775 | 0.773 | 0.720 | 0.757 | 0.763 | 0.774 | 0.854 |
| saugeen/W/short | 0.678 | 0.588 | 0.586 | 0.657 | 0.690 | 0.662 | 0.611 | 0.623 | 0.693 | 0.708 |
| solar/10T/long | 1.481 | 0.928 | 0.977 | 1.011 | 1.144 | 1.000 | 1.229 | 1.564 | 2.239 | 2.319 |
| solar/10T/medium | 1.196 | 0.969 | 0.970 | 0.951 | 1.075 | 0.906 | 1.108 | 1.324 | 1.963 | 2.039 |
| solar/10T/short | 0.823 | 0.982 | 0.928 | 0.934 | 0.745 | 0.854 | 0.896 | 1.015 | 1.004 | 0.995 |
| solar/D/short | 0.846 | 0.841 | 0.858 | 0.875 | 0.851 | 0.854 | 0.849 | 0.856 | 0.854 | 0.882 |
| solar/H/long | 0.929 | 0.688 | 0.879 | 0.894 | 1.311 | 1.011 | 0.964 | 0.911 | 0.952 | 0.999 |
| solar/H/medium | 1.061 | 0.792 | 0.937 | 0.926 | 1.323 | 0.924 | 0.996 | 0.993 | 0.981 | 0.954 |
| solar/H/short | 0.962 | 0.759 | 0.916 | 0.870 | 0.926 | 0.943 | 0.854 | 0.922 | 0.919 | 0.938 |
| solar/W/short | 0.804 | 0.830 | 0.570 | 0.970 | 0.678 | 0.539 | 0.666 | 1.415 | 1.041 | 1.129 |
| sz_taxi/15T/long | 0.787 | 0.736 | 0.766 | 0.749 | 0.851 | 0.810 | 0.789 | 0.754 | 0.802 | 0.777 |
| sz_taxi/15T/medium | 0.789 | 0.753 | 0.772 | 0.764 | 0.764 | 0.793 | 0.784 | 0.774 | 0.797 | 0.782 |
| sz_taxi/15T/short | 0.733 | 0.712 | 0.721 | 0.719 | 0.733 | 0.730 | 0.717 | 0.721 | 0.760 | 0.754 |
| sz_taxi/H/short | 0.781 | 0.763 | 0.780 | 0.769 | 0.780 | 0.776 | 0.762 | 0.768 | 0.814 | 0.797 |
| temperature_rain/D/short | 0.689 | 0.666 | 0.671 | 0.679 | 0.702 | 0.685 | 0.648 | 0.725 | 0.597 | 0.651 |
| us_births/D/short | 0.211 | 0.219 | 0.245 | 0.267 | 0.208 | 0.170 | 0.260 | 0.324 | 0.270 | 0.273 |
| us_births/M/short | 0.875 | 0.941 | 0.920 | 0.764 | 0.673 | 0.941 | 1.215 | 0.799 | 1.014 | 0.951 |
| us_births/W/short | 0.583 | 0.687 | 0.651 | 0.790 | 0.729 | 0.572 | 0.696 | 0.789 | 0.940 | 0.921 |

Table 16: CRPS performance summarized by average rank for zero-shot models on the GiftEval benchmark. A lower rank signifies superior probabilistic forecasting performance. The models achieving the first and second-best overall average ranks are highlighted.

| Dataset | TempoPFN | TiRex | FlowState-9.1M | Toto_Open_Base_1.0 | Chronos Bolt B | TabPFN-TS | YingLong_50m | TTM-R2-Finetuned | Moirai L 1.1 | Moirai B 1.1 |
|---|---|---|---|---|---|---|---|---|---|---|
| bitbrains_fast_storage/5T/long | **1.000** | 2.000 | 8.000 | 3.000 | 7.000 | 9.000 | 4.000 | 10.000 | 5.000 | 6.000 |
| bitbrains_fast_storage/5T/medium | **1.000** | 2.000 | 8.000 | 3.000 | 7.000 | 10.000 | 5.000 | 9.000 | 4.000 | 6.000 |
| bitbrains_fast_storage/5T/short | 2.000 | 3.000 | 8.000 | **1.000** | 7.000 | 10.000 | 6.000 | 9.000 | 4.000 | 5.000 |
| bitbrains_fast_storage/H/short | 4.000 | 7.000 | 8.000 | 2.000 | 9.000 | 6.000 | 3.000 | 11.000 | 5.000 | **1.000** |
| bitbrains_rnd/5T/long | 6.000 | 2.000 | 7.000 | **1.000** | 8.000 | 10.000 | 3.000 | 9.000 | 5.000 | 4.000 |
| bitbrains_rnd/5T/medium | 5.000 | **1.000** | 8.000 | 6.000 | 3.000 | 10.000 | 7.000 | 9.000 | 2.000 | 4.000 |
| bitbrains_rnd/5T/short | 9.000 | 2.000 | 7.000 | **1.000** | 5.000 | 10.000 | 4.000 | 8.000 | 3.000 | 6.000 |
| bitbrains_rnd/H/short | 6.000 | 5.000 | 7.000 | 3.000 | 4.000 | 9.000 | 8.000 | 10.000 | **1.000** | 2.000 |
| bizitobs_application/10S/long | 2.000 | 4.000 | 6.000 | 5.000 | 10.000 | 3.000 | 8.000 | 7.000 | 9.000 | 11.000 |
| bizitobs_application/10S/medium | **1.000** | 3.000 | 5.000 | 2.000 | 10.000 | 4.000 | 8.000 | 7.000 | 9.000 | 11.000 |
| bizitobs_application/10S/short | **1.000** | 3.000 | 4.000 | 2.000 | 11.000 | 5.000 | 6.000 | 7.000 | 10.000 | 8.000 |
| bizitobs_l2c/5T/long | 8.000 | 9.000 | **1.000** | 5.000 | 11.000 | 3.000 | 6.000 | 2.000 | 4.000 | 7.000 |
| bizitobs_l2c/5T/medium | 5.000 | 6.000 | **1.000** | 4.000 | 10.000 | 3.000 | 7.000 | 2.000 | 9.000 | 8.000 |
| bizitobs_l2c/5T/short | 3.000 | 7.000 | 10.000 | 2.000 | 4.000 | 9.000 | 6.000 | **1.000** | 8.000 | 5.000 |
| bizitobs_l2c/H/long | 5.000 | 3.000 | **1.000** | 6.000 | 2.000 | 4.000 | 7.000 | 10.000 | 9.000 | 8.000 |
| bizitobs_l2c/H/medium | 5.000 | 4.000 | **1.000** | 6.000 | 3.000 | 2.000 | 7.000 | 9.000 | 8.000 | 10.000 |
| bizitobs_l2c/H/short | 3.000 | 6.000 | **1.000** | 4.000 | 2.000 | 5.000 | 8.000 | 7.000 | 11.000 | 9.000 |
| bizitobs_service/10S/long | 6.000 | 4.000 | 5.000 | **1.000** | 10.000 | 2.000 | 8.000 | 7.000 | 9.000 | 11.000 |
| bizitobs_service/10S/medium | **1.000** | 4.000 | 2.000 | 3.000 | 11.000 | 5.000 | 7.000 | 6.000 | 9.000 | 10.000 |
| bizitobs_service/10S/short | 2.000 | 4.000 | 3.000 | **1.000** | 11.000 | 7.000 | 6.000 | 5.000 | 8.000 | 10.000 |
| car_parts/M/short | 7.000 | 3.000 | 6.000 | **1.000** | 4.000 | 2.000 | 10.000 | 8.000 | 9.000 | 5.000 |
| covid_deaths/D/short | 2.000 | 3.000 | 6.000 | **1.000** | 9.000 | 5.000 | 10.000 | 4.000 | 8.000 | 7.000 |
| electricity/15T/long | 10.000 | **1.000** | 2.000 | 7.000 | 6.000 | 4.000 | 5.000 | 3.000 | 8.000 | 11.000 |
| electricity/15T/medium | 8.000 | **1.000** | 2.000 | 7.000 | 5.000 | 4.000 | 6.000 | 3.000 | 9.000 | 10.000 |
| electricity/15T/short | 8.000 | 3.000 | 4.000 | 7.000 | **1.000** | 6.000 | 5.000 | 2.000 | 10.000 | 9.000 |
| electricity/D/short | 9.000 | **1.000** | 2.000 | 6.000 | 3.000 | 8.000 | 5.000 | 4.000 | 10.000 | 7.000 |
| electricity/H/long | 7.000 | 5.000 | 3.000 | **1.000** | 6.000 | 10.000 | 9.000 | 4.000 | 8.000 | 2.000 |
| electricity/H/medium | 9.000 | 4.000 | 2.000 | **1.000** | 5.000 | 10.000 | 7.000 | 3.000 | 8.000 | 6.000 |
| electricity/H/short | 8.000 | **1.000** | 3.000 | 5.000 | 2.000 | 6.000 | 10.000 | 4.000 | 9.000 | 7.000 |
| electricity/W/short | 6.000 | 2.000 | **1.000** | 9.000 | 3.000 | 4.000 | 8.000 | 5.000 | 7.000 | 10.000 |
| ett1/15T/long | 8.000 | 3.000 | **1.000** | 4.000 | 9.000 | 6.000 | 2.000 | 5.000 | 11.000 | 7.000 |
| ett1/15T/medium | 7.000 | 3.000 | **1.000** | 5.000 | 8.000 | 4.000 | 2.000 | 6.000 | 11.000 | 10.000 |
| ett1/15T/short | 8.000 | 2.000 | 4.000 | 3.000 | **1.000** | 5.000 | 6.000 | 7.000 | 10.000 | 9.000 |
| ett1/D/short | **1.000** | 3.000 | 4.000 | 5.000 | 7.000 | 8.000 | 2.000 | 10.000 | 6.000 | 9.000 |
| ett1/H/long | 6.000 | 2.000 | 4.000 | 3.000 | 10.000 | 8.000 | **1.000** | 5.000 | 9.000 | 7.000 |
| ett1/H/medium | 7.000 | 3.000 | **1.000** | 4.000 | 10.000 | 9.000 | 2.000 | 6.000 | 5.000 | 8.000 |
| ett1/H/short | 5.000 | 3.000 | **1.000** | 7.000 | 4.000 | 8.000 | 2.000 | 9.000 | 6.000 | 10.000 |
| ett1/W/short | 2.000 | 11.000 | **1.000** | 5.000 | 9.000 | 7.000 | 8.000 | 6.000 | 3.000 | 4.000 |
| ett2/15T/long | 7.000 | 5.000 | 2.000 | **1.000** | 8.000 | 6.000 | 3.000 | 4.000 | 9.000 | 11.000 |
| ett2/15T/medium | 6.000 | 3.000 | **1.000** | 4.000 | 10.000 | 7.000 | 2.000 | 5.000 | 8.000 | 9.000 |
| ett2/15T/short | 7.000 | 3.000 | 2.000 | 6.000 | 4.000 | 8.000 | 5.000 | **1.000** | 10.000 | 9.000 |
| ett2/D/short | 9.000 | 2.000 | 7.000 | 8.000 | 3.000 | 10.000 | 5.000 | **1.000** | 4.000 | 6.000 |
| ett2/H/long | **1.000** | 7.000 | 6.000 | 4.000 | 8.000 | 10.000 | 3.000 | 2.000 | 9.000 | 5.000 |
| ett2/H/medium | 6.000 | 7.000 | 4.000 | 2.000 | 8.000 | 10.000 | 3.000 | 5.000 | 9.000 | **1.000** |
| ett2/H/short | 5.000 | 7.000 | **1.000** | 3.000 | 2.000 | 10.000 | 4.000 | 6.000 | 8.000 | 9.000 |
| ett2/W/short | 4.000 | **1.000** | 7.000 | 8.000 | 3.000 | 6.000 | 10.000 | 5.000 | 9.000 | 2.000 |
| hierarchical_sales/D/short | 7.000 | **1.000** | 6.000 | 2.000 | 4.000 | 9.000 | 8.000 | 10.000 | 5.000 | 3.000 |
| hierarchical_sales/W/short | 2.000 | 4.000 | **1.000** | 6.000 | 5.000 | 3.000 | 10.000 | 9.000 | 8.000 | 7.000 |
| hospital/M/short | 6.000 | 4.000 | **1.000** | 5.000 | 9.000 | 8.000 | 10.000 | 7.000 | 3.000 | 2.000 |
| jena_weather/10T/long | 5.000 | 3.000 | 2.000 | **1.000** | 7.000 | 4.000 | 6.000 | 8.000 | 10.000 | 9.000 |
| jena_weather/10T/medium | 4.000 | 3.000 | 2.000 | **1.000** | 6.000 | 5.000 | 7.000 | 9.000 | 10.000 | 8.000 |
| jena_weather/10T/short | 4.000 | 3.000 | 2.000 | **1.000** | 5.000 | 6.000 | 7.000 | 8.000 | 9.000 | 10.000 |
| jena_weather/D/short | 7.000 | 2.000 | 4.000 | 8.000 | **1.000** | 3.000 | 5.000 | 10.000 | 9.000 | 6.000 |
| jena_weather/H/long | 5.000 | **1.000** | 8.000 | 2.000 | 6.000 | 9.000 | 4.000 | 10.000 | 3.000 | 7.000 |
| jena_weather/H/medium | 5.000 | **1.000** | 3.000 | 2.000 | 4.000 | 9.000 | 6.000 | 10.000 | 8.000 | 7.000 |
| jena_weather/H/short | 3.000 | **1.000** | 2.000 | 6.000 | 5.000 | 4.000 | 7.000 | 10.000 | 9.000 | 8.000 |
| kdd_cup_2018/D/short | 5.000 | 9.000 | 7.000 | 8.000 | 3.000 | **1.000** | 2.000 | 10.000 | 6.000 | 4.000 |
| kdd_cup_2018/H/long | 6.000 | 2.000 | 8.000 | 7.000 | **1.000** | 10.000 | 5.000 | 9.000 | 3.000 | 4.000 |
| kdd_cup_2018/H/medium | 5.000 | 2.000 | 6.000 | 8.000 | **1.000** | 9.000 | 4.000 | 10.000 | 3.000 | 7.000 |
| kdd_cup_2018/H/short | 6.000 | 2.000 | 5.000 | 8.000 | **1.000** | 10.000 | 4.000 | 9.000 | 3.000 | 7.000 |
| loop_seattle/5T/long | 9.000 | 6.000 | 3.000 | 4.000 | 11.000 | 7.000 | 8.000 | 5.000 | **1.000** | 2.000 |
| loop_seattle/5T/medium | 9.000 | 6.000 | 3.000 | 4.000 | 10.000 | 7.000 | 8.000 | 5.000 | **1.000** | 2.000 |
| loop_seattle/5T/short | 9.000 | 4.000 | 5.000 | 3.000 | 8.000 | 7.000 | 10.000 | 6.000 | **1.000** | 2.000 |
| loop_seattle/D/short | 3.000 | 2.000 | **1.000** | 8.000 | 5.000 | 4.000 | 6.000 | 10.000 | 9.000 | 7.000 |
| loop_seattle/H/long | 8.000 | **1.000** | 3.000 | 4.000 | 9.000 | 2.000 | 6.000 | 5.000 | 7.000 | 10.000 |
| loop_seattle/H/medium | 8.000 | 2.000 | 4.000 | **1.000** | 9.000 | 3.000 | 6.000 | 5.000 | 7.000 | 10.000 |
| loop_seattle/H/short | 9.000 | **1.000** | 2.000 | 5.000 | 6.000 | 3.000 | 4.000 | 8.000 | 7.000 | 10.000 |
| m4_daily/D/short | 9.000 | **1.000** | 7.000 | 3.000 | 2.000 | 4.000 | 6.000 | 5.000 | 10.000 | 11.000 |
| m4_hourly/H/short | 7.000 | 2.000 | 3.000 | 10.000 | 6.000 | 8.000 | 5.000 | 9.000 | **1.000** | 4.000 |
| m4_monthly/M/short | **1.000** | 2.000 | 3.000 | 8.000 | 6.000 | 4.000 | 10.000 | 9.000 | 7.000 | 5.000 |
| m4_quarterly/Q/short | 4.000 | 3.000 | 5.000 | 7.000 | 6.000 | 8.000 | 10.000 | 9.000 | 1.500 | 1.500 |
| m4_weekly/W/short | 6.000 | **1.000** | 2.000 | 10.000 | 4.000 | 3.000 | 5.000 | 7.000 | 8.000 | 9.000 |
| m4_yearly/A/short | 7.000 | 4.000 | 3.000 | 9.000 | 8.000 | 5.000 | 11.000 | 6.000 | **1.000** | 2.000 |
| m_dense/D/short | 2.000 | 3.000 | 5.000 | 7.000 | 4.000 | **1.000** | 8.000 | 6.000 | 9.000 | 10.000 |
| m_dense/H/long | 10.000 | 3.000 | 2.000 | 6.000 | 8.000 | 7.000 | 9.000 | 5.000 | **1.000** | 4.000 |
| m_dense/H/medium | 10.000 | 3.000 | 2.000 | 4.000 | 8.000 | 9.000 | 7.000 | 5.000 | **1.000** | 6.000 |
| m_dense/H/short | 10.000 | 3.000 | 4.000 | 7.000 | **1.000** | 8.000 | 9.000 | 6.000 | 2.000 | 5.000 |
| restaurant/D/short | 3.000 | **1.000** | 2.000 | 10.000 | 5.000 | 4.000 | 9.000 | 7.000 | 8.000 | 6.000 |
| saugeen/D/short | 7.000 | 8.000 | **1.000** | 3.000 | 2.000 | 6.000 | 5.000 | 10.000 | 9.000 | 4.000 |
| saugeen/M/short | 5.000 | 7.000 | 2.000 | 4.000 | 3.000 | **1.000** | 6.000 | 9.000 | 8.000 | 10.000 |
| saugeen/W/short | 7.000 | **1.000** | 2.000 | 5.000 | 3.000 | 6.000 | 4.000 | 10.000 | 9.000 | 8.000 |
| solar/10T/long | 7.000 | **1.000** | 3.000 | 4.000 | 5.000 | 2.000 | 8.000 | 6.000 | 10.000 | 11.000 |
| solar/10T/medium | 5.000 | 4.000 | 3.000 | 2.000 | 6.000 | **1.000** | 8.000 | 7.000 | 10.000 | 11.000 |
| solar/10T/short | 2.000 | 7.000 | 4.000 | 5.000 | 3.000 | **1.000** | 8.000 | 6.000 | 9.000 | 10.000 |
| solar/D/short | 3.000 | 4.000 | 2.000 | 7.000 | 5.000 | **1.000** | 6.000 | 10.000 | 8.000 | 9.000 |
| solar/H/long | 8.000 | **1.000** | 3.000 | 2.000 | 9.000 | 5.000 | 6.000 | 10.000 | 4.000 | 7.000 |
| solar/H/medium | 9.000 | **1.000** | 5.000 | 4.000 | 8.000 | 2.000 | 7.000 | 10.000 | 6.000 | 3.000 |
| solar/H/short | 8.000 | **1.000** | 5.000 | 3.000 | 2.000 | 6.000 | 9.000 | 10.000 | 4.000 | 7.000 |
| solar/W/short | 4.000 | 5.000 | 2.000 | 7.000 | 3.000 | **1.000** | 11.000 | 6.000 | 9.000 | 10.000 |
| sz_taxi/15T/long | 6.000 | **1.000** | 4.000 | 2.000 | 10.000 | 9.000 | 3.000 | 8.000 | 7.000 | 5.000 |
| sz_taxi/15T/medium | 6.000 | **1.000** | 3.000 | 2.000 | 10.000 | 9.000 | 4.000 | 7.000 | 8.000 | 5.000 |
| sz_taxi/15T/short | 6.000 | **1.000** | 3.000 | 4.000 | 2.000 | 7.000 | 5.000 | 8.000 | 10.000 | 9.000 |
| sz_taxi/H/short | 6.000 | **1.000** | 5.000 | 4.000 | 2.000 | 7.000 | 3.000 | 8.000 | 10.000 | 9.000 |
| temperature_rain/D/short | 8.000 | 5.000 | 4.000 | 6.000 | 3.000 | 7.000 | 9.000 | 10.000 | **1.000** | 2.000 |
| us_births/D/short | 2.000 | 4.000 | 5.000 | 7.000 | 6.000 | **1.000** | 10.000 | 3.000 | 9.000 | 8.000 |
| us_births/M/short | 4.000 | 7.000 | 6.000 | 2.000 | 11.000 | 8.000 | 3.000 | **1.000** | 9.000 | 5.000 |
| us_births/W/short | 2.000 | 4.000 | 3.000 | 8.000 | 5.000 | **1.000** | 7.000 | 6.000 | 10.000 | 9.000 |

Table 17: MASE performance presented as average ranks for zero-shot models across the GiftEval benchmark. A lower average rank indicates consistently higher accuracy across datasets. The two models with the best overall average ranks are highlighted.

| Dataset | TempoPFN | TiRex | FlowState-9.1M | Toto_Open_Base_1.0 | Chronos Bolt B | TTM-R2-Finetuned | TabPFN-TS | YingLong-50m | Moirai L 1.1 | Moirai B 1.1 |
|---|---|---|---|---|---|---|---|---|---|---|
| bitbrains_fast_storage/5T/long | 6.000 | 2.000 | 9.000 | 1.000 | 4.000 | 3.000 | 11.000 | 8.000 | 5.000 | 7.000 |
| bitbrains_fast_storage/5T/medium | 8.000 | 2.000 | 10.000 | 1.000 | 5.000 | 6.000 | 11.000 | 7.000 | 3.000 | 4.000 |
| bitbrains_fast_storage/5T/short | 6.000 | 2.000 | 10.000 | 1.000 | 4.000 | 3.000 | 9.000 | 7.000 | 8.000 | 5.000 |
| bitbrains_fast_storage/H/short | 8.000 | 3.000 | 5.000 | 1.000 | 2.000 | 10.000 | 9.000 | 6.000 | 4.000 | 7.000 |
| bitbrains_rnd/5T/long | 9.000 | 2.000 | 10.000 | 1.000 | 3.000 | 7.000 | 11.000 | 8.000 | 4.000 | 5.000 |
| bitbrains_rnd/5T/medium | 8.000 | 1.000 | 10.000 | 2.000 | 3.000 | 5.000 | 11.000 | 9.000 | 4.000 | 6.000 |
| bitbrains_rnd/5T/short | 8.000 | 2.000 | 9.000 | 1.000 | 3.000 | 5.000 | 11.000 | 6.000 | 4.000 | 7.000 |
| bitbrains_rnd/H/short | 7.000 | 2.000 | 5.000 | 1.000 | 4.000 | 9.000 | 11.000 | 3.000 | 6.000 | 10.000 |
| bizitobs_application/10S/long | 5.000 | 6.000 | 4.000 | 3.000 | 10.000 | 7.000 | 1.000 | 8.000 | 9.000 | 11.000 |
| bizitobs_application/10S/medium | 1.000 | 6.000 | 4.000 | 2.000 | 10.000 | 7.000 | 3.000 | 8.000 | 9.000 | 11.000 |
| bizitobs_application/10S/short | 1.000 | 4.000 | 5.000 | 2.000 | 11.000 | 6.000 | 3.000 | 7.000 | 9.000 | 10.000 |
| bizitobs_l2c/5T/long | 9.000 | 8.000 | 2.000 | 6.000 | 10.000 | 1.000 | 3.000 | 7.000 | 4.500 | 4.500 |
| bizitobs_l2c/5T/medium | 5.000 | 6.000 | 1.000 | 4.000 | 9.000 | 2.000 | 3.000 | 7.000 | 10.000 | 8.000 |
| bizitobs_l2c/5T/short | 3.000 | 8.000 | 10.000 | 2.000 | 4.000 | 1.000 | 9.000 | 6.000 | 5.000 | 7.000 |
| bizitobs_l2c/H/long | 5.000 | 3.000 | 1.000 | 7.000 | 2.000 | 9.000 | 4.000 | 6.000 | 10.000 | 8.000 |
| bizitobs_l2c/H/medium | 5.000 | 4.000 | 1.000 | 6.000 | 3.000 | 8.000 | 2.000 | 7.000 | 9.000 | 10.000 |
| bizitobs_l2c/H/short | 3.000 | 6.000 | 2.000 | 4.000 | 1.000 | 7.000 | 5.000 | 8.000 | 10.000 | 9.000 |
| bizitobs_service/10S/long | 7.000 | 6.000 | 4.000 | 1.000 | 10.000 | 5.000 | 2.000 | 8.000 | 9.000 | 11.000 |
| bizitobs_service/10S/medium | 7.000 | 4.000 | 2.000 | 1.000 | 10.000 | 5.000 | 3.000 | 8.000 | 9.000 | 11.000 |
| bizitobs_service/10S/short | 6.000 | 4.000 | 3.000 | 1.000 | 10.000 | 2.000 | 5.000 | 7.000 | 9.000 | 11.000 |
| car_parts/M/short | 5.000 | 4.000 | 8.000 | 1.000 | 7.000 | 3.000 | 6.000 | 11.000 | 9.000 | 2.000 |
| covid_deaths/D/short | 6.000 | 8.000 | 4.000 | 2.000 | 7.000 | 1.000 | 9.000 | 10.000 | 5.000 | 3.000 |
| electricity/15T/long | 11.000 | 2.000 | 3.000 | 7.000 | 4.000 | 1.000 | 5.000 | 6.000 | 9.000 | 10.000 |
| electricity/15T/medium | 9.000 | 2.000 | 3.000 | 7.000 | 4.000 | 1.000 | 5.000 | 6.000 | 10.000 | 11.000 |
| electricity/15T/short | 8.000 | 3.000 | 5.000 | 6.000 | 2.000 | 1.000 | 7.000 | 4.000 | 10.000 | 9.000 |
| electricity/D/short | 10.000 | 1.000 | 2.000 | 6.000 | 3.000 | 4.000 | 7.000 | 5.000 | 9.000 | 8.000 |
| electricity/H/long | 10.000 | 3.000 | 2.000 | 4.000 | 5.000 | 1.000 | 7.000 | 8.000 | 9.000 | 6.000 |
| electricity/H/medium | 10.000 | 2.000 | 4.000 | 5.000 | 1.000 | 3.000 | 7.000 | 6.000 | 9.000 | 8.000 |
| electricity/H/short | 7.000 | 1.000 | 4.000 | 5.000 | 2.000 | 3.000 | 6.000 | 10.000 | 8.000 | 9.000 |
| electricity/W/short | 6.000 | 2.000 | 1.000 | 8.000 | 3.000 | 4.000 | 5.000 | 7.000 | 9.000 | 10.000 |
| ett1/15T/long | 10.000 | 4.000 | 1.000 | 5.000 | 8.000 | 3.000 | 6.000 | 2.000 | 11.000 | 7.000 |
| ett1/15T/medium | 8.000 | 3.000 | 1.000 | 6.000 | 4.000 | 5.000 | 7.000 | 2.000 | 11.000 | 10.000 |
| ett1/15T/short | 8.000 | 4.000 | 2.000 | 3.000 | 1.000 | 5.000 | 7.000 | 6.000 | 10.000 | 9.000 |
| ett1/D/short | 2.000 | 6.000 | 1.000 | 4.000 | 5.000 | 11.000 | 3.000 | 7.000 | 9.000 | 8.000 |
| ett1/H/long | 7.000 | 1.000 | 2.000 | 5.000 | 4.000 | 8.000 | 10.000 | 3.000 | 9.000 | 6.000 |
| ett1/H/medium | 7.000 | 2.000 | 1.000 | 4.000 | 9.000 | 5.000 | 10.000 | 3.000 | 6.000 | 8.000 |
| ett1/H/short | 6.000 | 3.000 | 1.000 | 8.000 | 2.000 | 5.000 | 10.000 | 4.000 | 7.000 | 9.000 |
| ett1/W/short | 2.000 | 11.000 | 1.000 | 5.000 | 8.000 | 6.000 | 7.000 | 9.000 | 3.000 | 4.000 |
| ett2/15T/long | 9.000 | 3.000 | 1.000 | 2.000 | 6.000 | 5.000 | 7.000 | 4.000 | 10.000 | 11.000 |
| ett2/15T/medium | 7.000 | 2.000 | 1.000 | 6.000 | 5.000 | 4.000 | 8.000 | 3.000 | 10.000 | 11.000 |
| ett2/15T/short | 6.000 | 2.000 | 3.000 | 7.000 | 4.000 | 1.000 | 8.000 | 5.000 | 10.000 | 9.000 |
| ett2/D/short | 11.000 | 2.000 | 9.000 | 10.000 | 5.000 | 1.000 | 7.000 | 4.000 | 8.000 | 3.000 |
| ett2/H/long | 1.000 | 9.000 | 6.000 | 5.000 | 2.000 | 3.000 | 11.000 | 4.000 | 10.000 | 7.000 |
| ett2/H/medium | 7.000 | 8.000 | 6.000 | 2.000 | 3.000 | 4.000 | 11.000 | 1.000 | 9.000 | 5.000 |
| ett2/H/short | 5.000 | 4.000 | 1.000 | 3.000 | 2.000 | 6.000 | 10.000 | 7.000 | 8.000 | 9.000 |
| ett2/W/short | 7.000 | 5.000 | 9.000 | 10.000 | 1.000 | 4.000 | 2.000 | 8.000 | 11.000 | 6.000 |
| hierarchical_sales/D/short | 7.000 | 2.000 | 6.000 | 1.000 | 3.000 | 8.000 | 9.000 | 10.000 | 4.000 | 5.000 |
| hierarchical_sales/W/short | 4.000 | 2.000 | 1.000 | 7.000 | 6.000 | 3.000 | 5.000 | 9.000 | 8.000 | 8.000 |
| hospital/M/short | 6.000 | 3.000 | 2.000 | 8.000 | 9.000 | 1.000 | 4.000 | 10.000 | 5.000 | 7.000 |
| jena_weather/10T/long | 4.000 | 2.000 | 6.000 | 1.000 | 5.000 | 10.000 | 7.000 | 3.000 | 11.000 | 9.000 |
| jena_weather/10T/medium | 4.000 | 2.000 | 5.000 | 1.000 | 3.000 | 11.000 | 6.000 | 7.000 | 8.000 | 9.000 |
| jena_weather/10T/short | 3.000 | 4.000 | 2.000 | 1.000 | 5.000 | 7.000 | 6.000 | 8.000 | 9.000 | 10.000 |
| jena_weather/D/short | 9.000 | 1.000 | 2.000 | 7.000 | 3.000 | 10.000 | 8.000 | 4.000 | 5.000 | 6.000 |
| jena_weather/H/long | 8.000 | 2.000 | 7.000 | 3.000 | 4.000 | 11.000 | 10.000 | 5.000 | 1.000 | 6.000 |
| jena_weather/H/medium | 4.000 | 5.000 | 6.000 | 2.000 | 1.000 | 7.000 | 11.000 | 8.000 | 10.000 | 3.000 |
| jena_weather/H/short | 3.000 | 1.000 | 2.000 | 6.000 | 5.000 | 9.000 | 7.000 | 4.000 | 10.000 | 8.000 |
| kdd_cup_2018/D/short | 7.000 | 10.000 | 9.000 | 8.000 | 3.000 | 4.000 | 2.000 | 1.000 | 5.500 | 5.500 |
| kdd_cup_2018/H/long | 7.000 | 2.000 | 9.000 | 8.000 | 1.000 | 5.000 | 10.000 | 6.000 | 3.000 | 4.000 |
| kdd_cup_2018/H/medium | 7.000 | 2.000 | 9.000 | 8.000 | 1.000 | 4.000 | 10.000 | 5.000 | 3.000 | 6.000 |
| kdd_cup_2018/H/short | 8.000 | 2.000 | 7.000 | 9.000 | 1.000 | 6.000 | 10.000 | 4.000 | 3.000 | 5.000 |
| loop_seattle/5T/long | 9.000 | 6.000 | 3.000 | 4.000 | 10.000 | 5.000 | 7.000 | 8.000 | 1.000 | 2.000 |
| loop_seattle/5T/medium | 11.000 | 6.000 | 3.000 | 4.000 | 9.000 | 5.000 | 7.000 | 8.000 | 1.000 | 2.000 |
| loop_seattle/5T/short | 9.000 | 5.000 | 6.000 | 4.000 | 8.000 | 3.000 | 7.000 | 10.000 | 1.000 | 2.000 |
| loop_seattle/D/short | 3.000 | 1.000 | 2.000 | 9.000 | 5.000 | 4.000 | 7.000 | 10.000 | 8.000 | 6.000 |
| loop_seattle/H/long | 9.000 | 1.000 | 3.000 | 5.000 | 7.000 | 2.000 | 4.000 | 6.000 | 8.000 | 10.000 |
| loop_seattle/H/medium | 9.000 | 3.000 | 5.000 | 2.000 | 8.000 | 1.000 | 4.000 | 7.000 | 6.000 | 10.000 |
| loop_seattle/H/short | 9.000 | 1.000 | 2.000 | 5.000 | 6.000 | 3.000 | 7.000 | 4.000 | 8.000 | 10.000 |
| m4_daily/D/short | 10.000 | 1.000 | 6.000 | 5.000 | 2.000 | 4.000 | 9.000 | 7.000 | 8.000 | 11.000 |
| m4_hourly/H/short | 4.000 | 1.000 | 2.000 | 6.000 | 5.000 | 10.000 | 3.000 | 8.000 | 7.000 | 9.000 |
| m4_monthly/M/short | 3.000 | 2.000 | 1.000 | 9.000 | 5.000 | 4.000 | 7.000 | 10.000 | 8.000 | 6.000 |
| m4_quarterly/Q/short | 6.000 | 4.000 | 3.000 | 9.000 | 7.000 | 5.000 | 8.000 | 10.000 | 1.500 | 1.500 |
| m4_weekly/W/short | 8.000 | 1.000 | 3.000 | 7.000 | 5.000 | 2.000 | 4.000 | 6.000 | 9.000 | 11.000 |
| m4_yearly/A/short | 7.000 | 8.000 | 1.000 | 6.000 | 9.000 | 4.000 | 5.000 | 11.000 | 2.000 | 3.000 |
| m_dense/D/short | 2.000 | 3.000 | 6.000 | 7.000 | 4.000 | 5.000 | 1.000 | 8.000 | 9.000 | 10.000 |
| m_dense/H/long | 10.000 | 2.000 | 5.000 | 6.000 | 7.000 | 4.000 | 9.000 | 8.000 | 1.000 | 3.000 |
| m_dense/H/medium | 10.000 | 4.000 | 2.000 | 5.000 | 7.000 | 3.000 | 9.000 | 8.000 | 1.000 | 6.000 |
| m_dense/H/short | 10.000 | 3.000 | 4.000 | 7.000 | 1.000 | 5.000 | 8.000 | 9.000 | 2.000 | 6.000 |
| restaurant/D/short | 3.000 | 1.000 | 2.000 | 10.000 | 6.000 | 5.000 | 4.000 | 9.000 | 8.000 | 7.000 |
| saugeen/D/short | 7.000 | 9.000 | 1.000 | 4.000 | 2.000 | 5.000 | 8.000 | 6.000 | 10.000 | 3.000 |
| saugeen/M/short | 9.000 | 8.000 | 4.000 | 7.000 | 2.000 | 5.000 | 1.000 | 3.000 | 6.000 | 10.000 |
| saugeen/W/short | 7.000 | 2.000 | 1.000 | 5.000 | 3.000 | 8.000 | 6.000 | 4.000 | 9.000 | 10.000 |
| solar/10T/long | 8.000 | 1.000 | 2.000 | 5.000 | 7.000 | 6.000 | 3.000 | 9.000 | 10.000 | 11.000 |
| solar/10T/medium | 8.000 | 3.000 | 4.000 | 2.000 | 7.000 | 6.000 | 1.000 | 9.000 | 10.000 | 11.000 |
| solar/10T/short | 2.000 | 7.000 | 5.000 | 6.000 | 4.000 | 1.000 | 3.000 | 11.000 | 10.000 | 8.000 |
| solar/D/short | 2.000 | 1.000 | 8.000 | 9.000 | 3.000 | 4.000 | 6.000 | 7.000 | 5.000 | 10.000 |
| solar/H/long | 5.000 | 1.000 | 2.000 | 3.000 | 7.000 | 11.000 | 10.000 | 4.000 | 6.000 | 8.000 |
| solar/H/medium | 10.000 | 1.000 | 4.000 | 3.000 | 8.000 | 11.000 | 2.000 | 7.000 | 6.000 | 5.000 |
| solar/H/short | 10.000 | 1.000 | 4.000 | 3.000 | 2.000 | 7.000 | 9.000 | 6.000 | 5.000 | 8.000 |
| solar/W/short | 5.000 | 6.000 | 2.000 | 7.000 | 3.000 | 4.000 | 1.000 | 11.000 | 9.000 | 10.000 |
| sz_taxi/15T/long | 6.000 | 1.000 | 4.000 | 2.000 | 7.000 | 10.000 | 9.000 | 3.000 | 8.000 | 5.000 |
| sz_taxi/15T/medium | 8.000 | 1.000 | 4.000 | 3.000 | 7.000 | 2.000 | 9.000 | 5.000 | 10.000 | 6.000 |
| sz_taxi/15T/short | 7.000 | 1.000 | 5.000 | 3.000 | 2.000 | 8.000 | 6.000 | 4.000 | 10.000 | 9.000 |
| sz_taxi/H/short | 8.000 | 2.000 | 6.000 | 4.000 | 1.000 | 7.000 | 5.000 | 3.000 | 10.000 | 9.000 |
| temperature_rain/D/short | 8.000 | 4.000 | 5.000 | 6.000 | 2.000 | 9.000 | 7.000 | 10.000 | 1.000 | 3.000 |
| us_births/D/short | 3.000 | 4.000 | 5.000 | 7.000 | 6.000 | 2.000 | 1.000 | 10.000 | 8.000 | 9.000 |
| us_births/M/short | 4.000 | 7.000 | 5.000 | 2.000 | 11.000 | 1.000 | 6.000 | 3.000 | 10.000 | 8.000 |
| us_births/W/short | 2.000 | 4.000 | 3.000 | 8.000 | 5.000 | 6.000 | 1.000 | 7.000 | 10.000 | 9.000 |