# OpenReview forum: "TempoPFN: Synthetic Pre-training of Linear RNNs for Zero-shot Time Series Forecasting"
_ICLR.cc/2026/Conference — Submitted to ICLR 2026_

### Official Review · Reviewer_DAtM · 2025-10-27

**Soundness:** 3
**Presentation:** 3
**Contribution:** 2
**Rating:** 6
**Confidence:** 4

**Summary:**

This work introduces TempoPFN, a foundation model for univariate time series forecasting that makes two key contributions: it is trained entirely on synthetic data, including waveforms, Gaussian Processes (GPs), and Stochastic Differential Equations (SDEs), and it employs a lightweight linear gated recurrent neural network architecture with approximately 35 million parameters and a simple yet effective mechanism. Despite relying solely on synthetic data and a relatively modest model design, TempoPFN achieves competitive zero-shot performance, in comparison to state-of-the-art foundation models trained on real-world datasets.

**Strengths:**

* This work enhances the notion of real-data free training, suggesting that real data can be spanned by a combination of processes.
* A rather simple model, with solid performance. Additionally, this work emphasizes the growing potential in RNN (or SSM) models in a Transformer dominated environment together with works such as Tirex and FlowState.
* Solid results on the GIFT-Eval benchmark, which is well accepted in the TSFM community.
* Interesting combinations of common and custom waveforms, GP, and SDE's.
* Up to date comparisons.

**Weaknesses:**

* The rationale for combining the proposed synthetic components remains insufficiently justified. Given the diversity of these components and their varying parameter ranges, there is a concern that the authors may have curated them specifically to perform well on the chosen benchmark. Although some motivation is provided in Appendix Section A, this explanation lacks depth. To address this concern, the authors are encouraged to: (1) offer a stronger theoretical foundation or principled motivation for how these components interact and contribute to model robustness; (2) include additional evaluations on benchmarks such as LTSF [1,2] or Chronos/FEV [3,4] to demonstrate generalizability; or (3) provide other forms of analysis to convincing why this setup is essential.

It also appears from this work that the inclusion of any synthetic generator may positively contribute to the model’s success. If this is not the case, the authors should clarify why certain generators were chosen over others and demonstrate that the selected components are not arbitrarily beneficial but rather collectively essential and complementary. Without such justification, it remains unclear whether the performance gains stem from the specific design of the synthetic pipeline or simply from the presence of diverse synthetic signals.


* Generation overhead: The authors mention generating 10 million synthetic series but provide no analysis of generation time or comparison to existing methods like KernelSynth [3] or waveform-based approaches, leaving the efficiency of generation unclear.


* This is not the first work to challenge TSFM with real-data free training [5,6]. This work should also discuss the differences to Freq-Synth [6] or other methods in [7].

* This work isn't particularly better for long term forecasting as shown in Tab. 3.

**Questions:**

1) **Major**: Why are the results for TempoPFN in  Fig 4.  (MASE=0.797) different from the Base model in Tab.3 (MASE=0.842)?

2) In light of the first concern, why is sawtooth and sine waveforms used and not square/triangle waveform?

3) **Minor**: Have the authors tried comparing the model's performance trained with real data? for example training from LOTSA or datasets from the Monash repo? It may offer an interesting ablation study, measuring the model's performance alone


[1] Sundial: A Family of Highly Capable Time Series Foundation Models

[2[ Timer: Generative Pre-trained Transformers Are Large Time Series Models

[3] Chronos: Learning the Language of Time Series

[4] TiRex: Zero-Shot Forecasting Across Long and Short Horizons with Enhanced In-Context Learning

[5] From Tables to Time: How TabPFN-v2 Outperforms Specialized Time Series Forecasting Models

[6] Beyond Data Scarcity: A Frequency-Driven Framework for Zero-Shot Forecasting

[7] Empowering Time Series Analysis with Synthetic Data: A Survey and Outlook in the Era of Foundation Models

---

> ### Author Response · Authors · 2025-11-21
> **Official Comment by Authors (Part 1/2)**
>
> Thank you very much for your constructive review and for recognizing the contributions of our work. We appreciate the validation of our linear RNN approach as a computationally efficient alternative in a Transformer-dominated field. We have revised the paper to address your concerns regarding the justification of synthetic components, benchmarking scope, and generation overhead.
>
> ---
>
> >***the authors should clarify why certain generators were chosen over others***
>
> During the construction of the synthetic TS generators, we thought of the manifold of real-world time series as a composition of fundamental abstract behaviors and explicitly selected generators to cover specific dynamics that are often missing in standard priors. For example, we employ Gaussian Processes (GPs) to capture smooth trends and correlations, while our novel Regime-Switching Stochastic Differential Equation (SDE) generator was introduced to model the non-stationary volatility found typically in financial data. Similarly, we utilize sawtooth waves to explicitly model physical asymmetries (gradual rise/sharp drop) that simple sine waves fail to capture. Our ablation studies (Table 3) provide empirical validation of this coverage: they show that removing specific components, such as SDEs, leads to significant performance drops, hinting that these generators cover essential portions of the data distribution.
>
> ---
>
> >***On the generalizability of our synthetic data generator and additional benchmarks***
>
> As the author suggested, to demonstrate that we haven’t curated the set of various synthetic time series generators to perform well on GIFT-Eval, we have expanded our evaluation suite to include the **Chronos Zero-Shot** and the recent **fev-bench** benchmarks. The results of these experiments are now included in the Appendix, demonstrating that TempoPFN’s performance is consistent across different benchmarks and not overfitted to GIFT-Eval. Please refer to Appendix E.2 and Figure 18, where TempoPFN achieves competitive rankings on the Chronos Zero-Shot benchmark. We refer you to Appendix E.3 for the fev-bench leaderboard results, where recent SOTA models such as Chronos-2 and TimesFM-2.5 are also included. For completeness, we are adding the fev-bench leaderboard results below, where the models are ranked by win rate and skill score:
>
> | Rank | Model | Avg. Win Rate (%) | Skill Score (%) | Median Runtime (s) | Leakage (%) | Failed Tasks (%) | Organization | Zero-shot |
> |------|-------|-------------------|-----------------|--------------------|--------------|--------------------|--------------|-----------|
> | 1 | Chronos-2 | 88.0 | 35.5 | 3.57 | 0 | 0 | AWS | True |
> | 2 | TiRex | 76.7 | 30.0 | 1.4 | 1 | 0 | NX-AI | True |
> | 3 | TimesFM 2.5 | 74.9 | 30.2 | 10.89 | 10 | 0 | Google | True |
> | 4 | Toto 1.0 | 66.5 | 28.2 | 77.51 | 8 | 0 | Datadog | True |
> | 5 | Moirai 2.0 | 60.5 | 27.3 | 1.9 | 28 | 0 | Salesforce | True |
> | **6** | **TempoPFN** | **60.5** | **25.1** | **8.57** | **0** | **0** | **Anonymous** | **True** |
> | 7 | Chronos-Bolt | 60.1 | 26.5 | 1.0 | 0 | 0 | AWS | True |
> | 8 | TabPFN-TS | 58.2 | 27.6 | 300.57 | 0 | 2 | Prior Labs | True |
> | 9 | Sundial-Base | 52.4 | 24.7 | 33.99 | 1 | 0 | Tsinghua University | True |
> | 10 | Stat. Ensemble | 47.1 | 15.7 | 624.45 | 0 | 11 | — | False |
> | 11 | AutoARIMA | 35.6 | 11.2 | 120.16 | 0 | 10 | — | False |
> | 12 | AutoTheta | 33.6 | 11.0 | 9.27 | 0 | 0 | — | False |
> | 13 | AutoETS | 32.6 | 2.3 | 16.24 | 0 | 3 | — | False |
> | 14 | Seasonal Naive | 20.0 | 0.0 | 2.32 | 0 | 0 | — | False |
> | 15 | Naive | 18.4 | -16.7 | 2.24 | 0 | 0 | — | False |
> | 16 | Drift | 14.9 | -18.1 | 2.19 | 0 | 0 | — | False |

---

> > ### Author Response · Authors · 2025-11-21
> > **Official Comment by Authors (Part 2/2)**
> >
> > ---
> >
> > >***Generation overhead: The authors mention generating 10 million synthetic series but provide no analysis of generation time or comparison to existing methods like KernelSynth [3] or waveform-based approaches, leaving the efficiency of generation unclear.***
> >
> > Thank you for pointing out this relevant feature. While we initially omitted it for brevity, the synthetic data generation throughput is a key advantage of our pipeline compared to existing generators such as KernelSynth. To this end, we have added a new Appendix C.2 ("Synthetic Data Generation Throughput") including a table that details the generation speed of each generator. Unlike kernel-based methods such as KernelSynth, which can be computationally intensive due to the cubic $O(T^3)$ complexity of Gaussian Processes, our approach enables high-throughput generation as shown in the table (also shown below). Crucially, **the majority of our synthetic generators run exclusively on the CPU**. The GPU is leveraged primarily for the few neural network-based generators (e.g., Cauker). All series were generated with a batch size of 128 and a sequence length of 2048.
> >
> > | | Cauker | GP | Kernel | ForecastPFN | Sawtooth | Sinewave | Anomaly | Step | Stochastic Rhythm | Spike | SDE (OU Process) | Offline augmentations |
> > |---|--------|--------|--------|--------|--------|--------|--------|--------|--------|--------|--------|--------|
> > | **Length** | 2048 | 2048 | 2048 | 2048 | 2048 | 2048 | 2048 | 2048 | 2048 | 2048 | 2048 | 2048 |
> > | **Series / Sec** | 0.66 | 7.04 | 0.32 | 35.49 | 242.95 | 144.93 | 174.51 | 106.58 | 33.46 | 201.13 | 13.17 | 18.30 |
> >
> > ---
> >
> > >***This is not the first work to challenge TSFM with real-data free training [5,6]. This work should also discuss the differences to Freq-Synth [6] or other methods in [7].***
> >
> > Thank you for highlighting the related works, including Freq-Synth and FlowState. We have updated our Appendix B to explicitly contrast them with TempoPFN. While Freq-Synth adopts a task-specific generation paradigm requiring knowledge of the target sampling rate, TempoPFN learns a universal prior suitable for zero-shot transfer. Compared to FlowState, as a concurrent work exploring SSMs, our purely synthetic pre-training approach is a crucial difference from their real-data training regime.
> >
> > ---
> >
> > >***why is sawtooth and sine waveforms used and not square/triangle waveform?***
> >
> > We selected sine waveforms due to their smooth symmetric periodicities, relevant in capturing seasonal trends. On the other hand, sawtooth waves explicitly model temporal non-smooth asymmetry, such as the gradual rise and sharp drop seen in various domains, which symmetric sine waves cannot capture. We did not include Square and Triangle waveforms as we deemed them as redundant primitives. Square waves offer only two-level step dynamics, which are already captured more flexibly by our general Step generator. Triangle waves add little beyond what Sawtooth (ramps) and Sine (smooth periodicity) already span. Moreover, since our synthesis pipeline is compositional, superimposing and modulating these core generated signals, it naturally subsumes square and triangle waves as well, and ensures our portfolio is distinct yet comprehensive. To clarify further our data generation design principles, we have added a new Appendix A.2 ("Design Principles for Synthetic Data Generation") in the revised paper PDF.
> >
> > ---
> >
> > >***Why are the results for TempoPFN in Fig 4. (MASE=0.797) different from the Base model in Tab.3 (MASE=0.842)?*** and ***This work isn't particularly better for long term forecasting as shown in Tab. 3.***
> >
> > We apologize for the confusion these results created. Figure 4 shows the performance of our final model, pretrained for 4 million iterations, whereas Table 3 reports ablation studies where we use a smaller pretraining budget (500k iterations), mainly due to limited compute. Regarding the performance gap in long-horizon tasks, we hypothesize that this is primarily due to our training context window being limited to 2048 steps. We are currently exploring longer contexts to bridge this gap and we will include these updated results if available by the camera-ready deadline.
> >
> > ---
> >
> > Thank you again for your valuable feedback and insights, which have ultimately improved our paper and made our case stronger with the additional experiments on Chronos-ZS and fev-bench. We are happy to discuss any further concerns you may have during this discussion phase. Otherwise, we would really appreciate it if you would consider raising your score.
> >
> > ---
> >
> > *-- References –*
> >
> > Ansari et al. “Chronos-2: From Univariate to Universal Forecasting”. 2025
> >
> > Das et al. “A decoder-only foundation model for time-series forecasting”. ICML 2024
> >
> > Shchur et al. “fev-bench: A Realistic Benchmark for Time Series Forecasting”. 2025

---

> > > ### Comment · Reviewer_DAtM · 2025-11-22
> > >
> > > Thanks for the thorough reply. Most of the concerns were addressed; I therefore revised my score.
> > > I think this paper is valuable to the TSFM community, suggesting that a combination of synthetic data can indeed have the potential to span real data.
> > >
> > > I recommend accept.

---

> > > > ### Author Response · Authors · 2025-11-22
> > > >
> > > > Thank you very much for taking the time to reevaluate our work and for increasing your score. We appreciate your recognition of both the practical value and the methodological contributions of TempoPFN to the TSFM community. Your feedback has helped us significantly strengthen the paper.

---

### Official Review · Reviewer_SoBF · 2025-10-31

**Soundness:** 3
**Presentation:** 2
**Contribution:** 2
**Rating:** 2
**Confidence:** 4

**Summary:**

The authors introduce a time series foundation model (TSFM), called TempoPFN, based on linear recurrent neural networks that is pretrained solely on synthetic data.

**Strengths:**

The model of the authors is trained exclusively on synthetic data which is advantageous for zero-shot generalization on various benchmark tasks.
Also, the authors open-source their data processing and generation pipeline, which the authors can really appreciate, as it allows others to quickly build on top of this work and test out new research ideas.

**Weaknesses:**

Unclear description of the core architecture of the paper:\
Unfortunately, the authors describe their architecture in not much detail on roughly half a page (including a Figure). Instead, the author refer the reader to the paper of the DeltaProduct, which forms the core of this architecture. Since the authors present this as a novel model architecture, it would be good to at least have a better description of this critical building block. There appears to be more text on page 3 introducing and describing the linear RNNs than there is text describing the DeltaProduct. This makes the paper awkward to read, because if the reader is not familiar with the DeltaProduct, he is basically forced to look a the other paper to understand it.

Overly focused on TiRex:\
It feels like the work of the authors is overly focused on the comparison point with TiRex. Alone the term "TiRex" is mentioned over 80 times in the paper! and in a lot of places the reference feels awkward and artificial. Especially given that there are other SSM, or linear RNN-based architectures in the literature, some of which the authors put in their results tables, but don't even mention once in the text.

Extensive pretraining:\
The authors mention that "all ablation experiments were conducted for a limited budget of 500,000 iterations. It is crucial to note that this is only a tenth of the full training schedule for our main model". However, their apparent kryptonite enemy TiRex only pretrains for 500,000 in the first place, so in total ten times less than TempPFN, however, the authors remain silent about this point.

**Questions:**

Why the extreme fixation on TiRex? I find the paper per se nice, but this artificial comparison makes the paper read very awkward an distracts from the merits of the authors' own work.\
Why is so much pretraining needed? Is this the tradeoff one has to take when only dealing with synthetic data in the first place?\
Why was the DeltaProduct architecture chosen as the basis? It remains a bit unclear until the end why the authors didn't opt for any other architecture, such an an SSM-based architecture?

---

> ### Author Response · Authors · 2025-11-21
> **Official Comment by Authors (Part 1/2)**
>
> Thank you for reviewing our paper and for your valuable feedback. We are encouraged that you see the contributions of our work and the potential to facilitate future research, and we are committed to addressing the presentation concerns to make the final paper stronger.
>
> ---
>
> >***Unfortunately, the authors describe their architecture in not much detail on roughly half a page (including a Figure). Instead, the author refer the reader to the paper of the DeltaProduct, which forms the core of this architecture. Since the authors present this as a novel model architecture, it would be good to at least have a better description of this critical building block.***
>
> We acknowledge the fact that the description of the GatedDeltaProduct architecture was too brief and have expanded it in the revision (see Appendix B.1). We will incorporate these details in the main paper upon acceptance when the page limit is higher. GatedDeltaProduct was particularly designed to better capture long-range dependencies with its superior state-tracking capabilities compared to standard SSMs. It maintains the computational efficiency of linear recurrence while effectively overcoming the state-decay and state-explosion issues often found in diagonal SSMs like Mamba.
> 1) **Main blocks**: The core recurrence relation is based on the DeltaProduct architecture, defined by the following linear update rule: $H_t = \mathbf{A} H_{t-1} + \mathbf{B} x_t$. Here, $x_t \in \mathbb{R}^D$ represents the input vector at the current time step $t$, where $D$ is the input dimension (e.g., number of features). $H_t \in \mathbb{R}^N$ is the hidden state vector, representing the compressed memory, where $N$ is the state dimension. $\mathbf{A} \in \mathbb{R}^{N \times N}$ is the state-transition matrix, which defines how the information from the previous hidden state evolves and is stored in the new state. DeltaProduct utilizes a dense, parameterized matrix $\mathbf{A}$, which is explicitly constrained to be near orthogonal through its initialization and parameterization scheme. This orthogonality constraint ensures that the transformation applied to the hidden state, $H_{t-1}$, preserves its magnitude. Consequently, the hidden state $H_t$ can effectively maintain its stability and information content over extended time steps, enabling robust state tracking of long-term trends and cyclical patterns in time series forecasting. Finally, $\mathbf{B} \in \mathbb{R}^{N \times D}$ is the input weight matrix. This matrix transforms the current input $x_t$ and integrates it into the hidden state $h_t$.
> 2) **Gating**: The gating in GatedDeltaProduct was introduced to embed essential non-linearity in the basic linear recurrence above, therefore enhancing the model's overall expressiveness and selective memory. After the main linear recurrence is performed its output is processed along two parallel streams: 1) *Main Stream*: The result of the linear recurrence, intended for the final output; 2) *Gate Stream*: The same recurrent output is passed through a non-linear activation function (e.g., the SiLU or Swish function). These two streams are combined via element-wise multiplication (the gate). This operation selectively controls which parts of the recurrent output are emphasized or suppressed, mirroring the functionality of sophisticated recurrent units like the Gated Recurrent Unit (GRU) or the selectivity found in Mamba's architecture.
> 3) **State Weaving**: This mechanism was specifically designed for the overall multi-layer structure of the TempoPFN framework where multiple GatedDeltaProduct layers are stacked. Instead of simply discarding the final hidden state, the mechanism *weaves* temporal information across the depth of the model. Specifically, the final hidden state ($ \mathbf{H}_T $) outputted by the first GatedDeltaProduct layer in the stack is passed forward to serve as the initial hidden state ($ \mathbf{H}_0 $) for the subsequent GatedDeltaProduct layer. This ensures that deeper layers do not start their recurrence from a blank slate but instead *build upon the aggregated temporal state representations* learned by the shallower layers. This process creates a dense flow of information across both the time dimension (via the recurrence) and the model depth (via the weaving).

---

> > ### Author Response · Authors · 2025-11-21
> > **Official Comment by Authors (Part 2/2)**
> >
> > >***Why was the DeltaProduct architecture chosen as the basis? It remains a bit unclear until the end why the authors didn't opt for any other architecture, such an an SSM-based architecture?***
> >
> > We chose GatedDeltaProduct over other SSMs (like Mamba) specifically for its state-tracking capabilities. Recent theoretical work (Grazzi et al. 2025) shows that diagonal SSMs suffer from hidden state decay, making them less suitable for tasks such as time series forecasting that require access to previous states, e.g., seasonality trend features. GatedDeltaProduct maintains state consistency without decay over long horizons, a crucial property for zero-shot forecasting (see also the last paragraph in Appendix B.1).
> >
> > ---
> >
> > >***Overly focused on TiRex*** and ***this artificial comparison makes the paper read very awkward an distracts from the merits of the authors' own work.***
> >
> > Regarding the observation that "TiRex" is mentioned over 80 times, we wish to provide important context. A quantitative check reveals that the vast majority of these instances occur in the Appendix, specifically within the headers of our comprehensive results tables (Tables 11–14) and the labels of our qualitative comparison plots (Figures 5, 18, and Appendix figures). In these tables and plots, TiRex appears as a column header or label for every metric and subplot, exactly as frequently as other baselines like TabPFN-TS. In the main paper, the frequency is significantly lower. However, we agree that the readability of the main text is paramount. In the revised manuscript, we have further reduced repetitive mentions in the narrative, replacing them with general terms (e.g., "state-of-the-art RNN baselines"), to ensure a smoother flow. We would also like to note that Tirex was the state-of-the-art model for TS forecasting at the time our paper was initially written and is still the state-of-the-art non-linear RNN model for TS. As such, it is a natural baseline to emphasize. Nevertheless, we have added new models such as Chronos-2 (Ansari et al. 2025) or TimesFM-2.5 (Das et al. 2025) in our recent evaluation on the latest FEV-Bench leaderboard (Appendix E.3 and Table 12 and 13).
> >
> > ---
> >
> > >***Why is so much pretraining needed?*** and ***their apparent kryptonite enemy TiRex only pretrains for 500,000 in the first place, so in total ten times less than TempPFN, however, the authors remain silent about this point.***
> >
> > We apologize for the confusion. We would like to clarify that the comparison is fairer when accounting for the effective batch size. While the iteration count differs significantly (500k vs 5M), this overlooks that TiRex utilizes a batch size of 256, whereas TempoPFN uses only 40, resulting in a total of $1.28 \times 10^8$ and $2 \times 10^8$ samples, respectively. This results in a difference of roughly **1.56x**, not 10x. Furthermore, our sequences are frequently subsampled or cut (max 2048), therefore, the total token count is comparable to or even lower than TiRex. We have added the batch size detail in Section 4. In addition, *we have also updated the manuscript with a final model with convolution size increased from 16 to 32 and retrained for 4M iterations*, making the difference with Tirex’ iterations lower (1.25x), which achieved slightly better performance compared to our initial submission (see updated Figure 4). It is worth noting that these results were achieved with *minimal hyperparameter optimization due to computational constraints*. More performance gains might remain accessible through further tuning.
> >
> > ---
> >
> > Thank you again for your valuable feedback and constructive criticism, which have significantly strengthened our paper and its narrative. We believe TempoPFN represents a novel and practical contribution to efficient RNN alternatives to transformer-based time series foundation models. Given the comprehensive rebuttal, the new empirical evidence supporting the synergistic role of our architecture and the synthetic data generator, and the positive feedback from several reviewers, we respectfully request that you consider raising your score to reflect the substantial improvements made to the submission.
> >
> > ---
> >
> > *-- References –*
> >
> > Grazzi et al. “Unlocking State-Tracking in Linear RNNs Through Negative Eigenvalues”. ICLR 2025
> >
> > Ansari et al. “Chronos-2: From Univariate to Universal Forecasting”. 2025
> >
> > Das et al. “A decoder-only foundation model for time-series forecasting”. ICML 2024.

---

### Official Review · Reviewer_oLYK · 2025-10-31

**Soundness:** 3
**Presentation:** 3
**Contribution:** 3
**Rating:** 8
**Confidence:** 3

**Summary:**

This paper introduces TempoPFN, a univariate TSFM based on linear RNNs. It adopts a GatedDeltaProduct architecture that enables fully parallelizable training across sequence lengths. TempoPFN is entirely pre-trained on synthetic data. The data synthesis pipeline incorporates diverse generators, including SDEs, Gaussian processes, and audio synthesis, alongside novel data augmentation methods such as time-varying TSMixup, differentiation, and integration. In zero-shot evaluations on the Gift-Eval benchmark, TempoPFN achieves performance comparable to models trained on real-world data, while demonstrating significantly higher computational efficiency than existing baseline methods.

**Strengths:**

- Unlike the currently dominant Transformer architecture, this paper employs a linear RNN as its framework, achieving performance comparable to models with hundreds of millions of parameters while maintaining under 35M parameters itself.
- The paper introduces a comprehensive approach to data synthesis and augmentation, enabling the complete elimination of real-world data for model training. Combined with the compact model architecture, this significantly lowers the barrier for training and deploying TSFMs.

**Weaknesses:**

- The paper employs GatedDeltaProduct, which is a relatively novel network architecture. However, the paper lacks a clear explanation of this architecture and justification for its selection.
- The authors mention the concepts of "state tracking" and "state weaving".  However, the authors neither define these terms nor clarify their relevance to time series forecasting tasks. This reflects a broader trend in the field, where methodologies from LLM and computer vision are frequently adopted without a formal discussion of their applicability to time series data. In such cases, sequences are often treated as textual or visual analogs without rigorous justification.
- There are some writing flaws: an incomplete sentence at line 194, duplicated references at lines 540 and 544.

**Questions:**

- In Figure 2, the input of TempoPFN includes both historical and future time indices, specifically $t_0, t_1, \cdots, t_5$ in the Figure. Do these time indices contain granular information such as year, month, and day as mentioned in line 176? Given that all data used in this study are synthetically generated, how are these temporal attributes actually assigned?
- At line 202, could the authors elaborate on how TempoPFN achieves probabilistic forecasting for arbitrary future horizons?
- TempoPFN is a foundation model trained entirely on synthetic data, which means the diversity of synthetic data is critically important. Do the authors have a methodology to evaluate how well their proposed data synthesis approach can cover real-world data distributions? Additionally, what guiding principles should inform the construction of such data synthesis methods?

---

> ### Author Response · Authors · 2025-11-21
> **Official Comment by Authors (Part 1/2)**
>
> Thank you for reading our paper in detail, as well as for the positive assessment and score. We appreciate your insightful questions regarding our data synthesis methodology and architectural details, which have helped us clarify the foundations of our work. Below we address your main concerns.
>
> ---
>
> >***the paper lacks a clear explanation of this architecture and justification for its selection.*** and ***clarifying "State Tracking" and "State Weaving"***
>
> We agree with you. We have now clarified these concepts in Appendix B.1 of the revision. There, we explain how our GatedDeltaProduct architecture achieves state-tracking via orthogonal state-transition matrix rotations, offering a significant advantage over the diagonal state-transition matrix as in SSMs, which can suffer from state decay. The linear RNN core of the GatedDeltaProduct architecture is defined by the update rule $h_t = \mathbf{A} h_{t-1} + \mathbf{B} x_t$. In this equation, $\mathbf{A}$ is the *state-transition matrix* that updates the previous *hidden state* $h_{t-1}$, and $\mathbf{B}$ is the *input weight matrix* that incorporates the current input $x_t$. The update is a linear combination of the transformed previous state and the current input, which is then extended with a non-linear gate in the full GatedDeltaProduct layer. Note though, that the main recurrence is still linear, which makes the training/inference highly parallelizable. In sequence models, "State Tracking" refers to the RNN's ability to maintain a hidden state that accurately summarizes the history (trend and level) without decaying over time. We can think of this hidden state as a form of compressed memory that encodes the temporal dynamics of the sequence. State-tracking is particularly important in tasks where the model does not have access to past observations, so it needs to keep track of an internal state in order to accurately reason at the current time step. On the other hand, "State Weaving" is a specific architectural mechanism for passing the final hidden state of one layer in the RNN to the initial state of the next layer, enabling bidirectional information flow across the model depth. We have expanded Section 3.1 to define this clearly, and we will add additional details in this section in the camera-ready version.
>
> ---
>
> >***There are some writing flaws: an incomplete sentence at line 194, duplicated references at lines 540 and 544.***
>
> We apologize for these oversights. We have made a full pass through the text again and corrected the incomplete sentence at line 194 and resolved the duplicated references at lines 540 and 544 in the revised manuscript.
>
> ---
>
> >***In Figure 2, the input of TempoPFN includes both historical and future time indices, specifically in the Figure. Do these time indices contain granular information such as year, month, and day as mentioned in line 176? Given that all data used in this study are synthetically generated, how are these temporal attributes actually assigned?***
>
> To assign granular information (Year, Month, Day) to synthetically generated data, we employ a randomized injection strategy. Since most of our synthetic generators (e.g., SDEs, Waveforms) produce numerical sequences that are inherently frequency-agnostic, we bridge this gap during training by dynamically assigning a random frequency (e.g., hourly, daily, weekly) and a random start timestamp to each generated series. Based on these assigned attributes, we utilize the GluonTS time feature utilities to compute granular timestamp embeddings (encoding attributes such as Year, Month, Day, Hour) of dimension 25. This ensures the model learns to associate specific time feature embeddings with actual signal dynamics, effectively mimicking real-world seasonality.
>
> ---
>
> >***At line 202, could the authors elaborate on how TempoPFN achieves probabilistic forecasting for arbitrary future horizons?***
>
> We apologize for not making this clearer. To forecast a horizon $H$, we generate positional embeddings for the future time steps based on their known time features (e.g., dates) and explicitly concatenate these with the history embeddings into a single unified sequence: [History, Future]. The Linear RNN then processes this sequence in a single forward pass. The recurrent state accumulates temporal dynamics through the history values and continues evolving through the future steps, driven effectively by the time-feature embeddings, to output the 9 equidistant predicted quantile levels ({0.1, 0.2, …, 0.9}; this is the same for TabPFN-TS and Tirex) for all $H$ steps simultaneously. This mechanism allows for flexible horizon lengths bounded only by available memory, avoiding the error accumulation typical of autoregressive decoding. We have added details on the quantile loss computation in Appendix D and we will extend the last paragraph of Section 3.1 in the camera ready copy.

---

> > ### Author Response · Authors · 2025-11-21
> > **Official Comment by Authors (Part 2/2)**
> >
> > >***Do the authors have a methodology to evaluate how well their proposed data synthesis approach can cover real-world data distributions? Additionally, what guiding principles should inform the construction of such data synthesis methods?***
> >
> > Thank you for raising this important point. We have now added details and guidelines on our synthetic data generation pipeline in Appendix A.2 of the revised paper PDF. During the construction of the synthetic TS generators, we thought of the manifold of real-world time series as a composition of fundamental abstract behaviors and explicitly selected generators to cover specific dynamics that are often missing in standard priors. For example, we employ Gaussian Processes (GPs) to capture smooth trends and correlations, while our novel Regime-Switching Stochastic Differential Equation (SDE) generator was introduced to model the non-stationary volatility found typically in financial data. Similarly, we utilize sawtooth waves to explicitly model physical asymmetries (gradual rise/sharp drop) that simple sine waves fail to capture. Our ablation studies (Table 3) provide empirical validation of this coverage: they show that removing specific components, such as SDEs, leads to significant performance drops, hinting that these generators cover essential portions of the data distribution.
> >
> > ---
> >
> > Thank you again for reading our paper in detail and your valuable feedback. We are happy to discuss any further concerns you may have during this discussion phase.

---

> ### Author Response · Authors · 2025-11-28
> **Follow-up on Real-World Distribution Coverage**
>
> Dear Reviewer oLYK,
>
> We are writing to follow up on your question regarding a methodology to evaluate how well the **synthetic data** covers **real-world distributions**.
>
> In our initial response (Part 2/2), we discussed our theoretical design principles ("**Structural Decomposition**"). We have now gone a step further and implemented a **quantitative feature-space analysis** to **empirically validate** this coverage.
>
> We invite you to examine the newly added **Appendix E.4** (*Spanning the Real-World Feature Manifold*) and **Figure 20** in the revised PDF. We extracted comprehensive **statistical time-series features** of both our synthetic generators and the real-world evaluation benchmarks (**GIFT-Eval**, **FEV-Bench**, **Chronos**) and projected these **high-dimensional feature vectors** into **2D and 3D latent spaces** using **UMAP**.
>
> This analysis provides strong visual evidence that our synthetic prior does **not collapse into trivial modes** but instead **successfully spans the complex feature manifold** of real-world time series, directly addressing your query about **data diversity and coverage**.
>
> We thank you again for your valuable insights, which have strengthened our paper, and we welcome any additional questions.

---

### Official Review · Reviewer_8sXE · 2025-10-31

**Soundness:** 2
**Presentation:** 3
**Contribution:** 1
**Rating:** 2
**Confidence:** 5

**Summary:**

The paper TempoPFN addresses the task of zero-shot time series forecasting, which is an importance task in the time series domain, and recently getting popularity. The paper has 2 main contributions. It trains linear RNNs, recently proposed in literature, only on synthetic data and evaluate on a exhaustive benchmark GIFT-Eval. The paper proposes a pipeline to generate a diverse set of synthetic data.

**Strengths:**

1. Presentation of the paper is good.
2. The synthetic data generation pipeline is pretty exhaustive, and covers many types of synthetic data.
3. The author(s) promise to open source the codes and pipelines.

**Weaknesses:**

However, the paper has a few weaknesses.
1. The novelty of the paper is limited. First, linear RNNs are not new, they are adopted from the literature. Training models purely on synthetic data is not new; the paper just creates a more diverse set of synthetic data. Given the idea of TabPFN or ForecastPFN, the proposed work can be tried quite trivially, without much complications.
2. The performance of the model is not up to the mark and marginally better than TabPFN-TS in CRPS and (from 0.544 to 0.536), but worse in MASE (0.771 to 0.797) after including significantly more and diverse set of synthetic data. This questions the efficacy and strength of the proposed work. The method is significantly worse than other methods that use real data alongwith synthetic data.
3. "We developed several new generators to fill gaps in existing approaches and capture specific temporal behaviors." -- There are infinite ways to generate time series, why only these types? Is there any real motivation behind the time series generation?
4. Comment: The paper title is TEMPOPFN where PFN stands for Prior Data Fitted Networks. The paper should discuss PFNs (at least briefly) for the readers who are unaware about it. I did not find any mention of PFNs in the methodology.

**Questions:**

See weaknesses.

---

> ### Author Response · Authors · 2025-11-21
> **Official Comment by Authors (Part 1/2)**
>
> Thank you for your constructive feedback, particularly regarding the justification of our synthetic data generation pipeline and the positioning of our work. Based on your comments, we have significantly revised the manuscript to clarify our contributions. We address your specific concerns below.
>
> ---
>
> >***The novelty of the paper is limited. First, linear RNNs are not new, they are adopted from the literature. Training models purely on synthetic data is not new; the paper just creates a more diverse set of synthetic data. Given the idea of TabPFN or ForecastPFN, the proposed work can be tried quite trivially, without much complications.***
>
> We respectfully disagree with the assessment that our work is “trivial to attempt” given existing literature, as this overlooks the complexity of engineering a high-performance foundation model. *Our novelty spans both architecture and data*. While linear RNNs and synthetic pre-training are known concepts, their successful integration, especially in the time series domain, is not. We propose the first linear RNN (GatedDeltaProduct) capable of achieving close to SOTA zero-shot performance, demonstrating that complex non-linear models are not strictly necessary. We would also like to note that linear RNNs have been around for some time, however, their applicability to large scale language modeling tasks was not possible until recently. Similarly, we believe that TempoPFN is a first step in creating more efficient time series foundation models that are not based on the Transformer architecture. Further on, the synthetic data pipeline itself is a significant contribution. Constructing a diverse, non-trivial synthetic data generator that spans the real-world time series manifold without collapsing into trivial patterns is a substantial engineering challenge. Specific generators in our pipeline, such as the SDE one, are also novel to the best of our knowledge. Finally, unlike all prior work with state-of-the-art performance (Chronos 2, Tirex, TimesFM, Todo, YinLong, TabPFN-TS, etc) whose high-performance prior remains proprietary and closed-source, *we provide the fully open-source*, high-performance synthetic data generator that rivals these alternatives and democratizes research in this domain. We thus expect our work to be enormously influential in the community.
>
> ---
>
> >***The method is significantly worse than other methods that use real data along with synthetic data.***
>
> We acknowledge the performance gap, but we argue that comparing synthetic-only models to real+synthetic models overlooks one of the core scientific contribution of our work: answering *”How far can we push forecasting performance using strictly synthetic priors?”* This distinction is relevant because TempoPFN: 1) eliminates benchmark contamination risk inherent in large "real data" models, 2) requires zero access to private data for deployment in regulated environments, ensuring privacy and compliance, and 3) remains competitive with models trained on gigabytes of real data, demonstrating that well-structured synthetic priors can effectively approximate the manifold of real-world dynamics. In our opinion, this is a positive finding.
>
> ---
>
> >***There are infinite ways to generate time series, why only these types? Is there any real motivation behind the time series generation?***
>
> We treat real-world time series as composites of fundamental properties or functions. Rather than arbitrarily selecting from "infinite" possibilities, we chose a specific set of diverse generator classes to act as essential "basis functions" for the space of time series. For instance, *Gaussian Processes* were employed to cover the manifold of smoothness and trend, whereas *SDEs* were included to capture stochastic volatility and mean-reversion, which standard noise injection fails to replicate. Similarly, we incorporate *Sawtooth waves* to model asymmetric periodicity (e.g., rapid rise/slow decay in inventory systems), a physical characteristic that standard sinusoidal priors cannot represent. In the revised paper PDF, we have added a new section (Appendix A.2) with more details on design principles for our synthetic time series generators.
>
> ---
>
> >***The paper should discuss PFNs (at least briefly) for the readers who are unaware about it.***
>
> We have added a dedicated section in Appendix A.1 providing a formal definition of PFNs and the connection they draw between In-context Learning and Bayesian inference.

---

> > ### Author Response · Authors · 2025-11-21
> > **Official Comment by Authors (Part 2/2)**
> >
> > >***The performance of the model is not up to the mark and marginally better than TabPFN-TS in CRPS and (0.533 to 0.544 to), but worse in MASE (0.771 to 0.788) after including significantly more and diverse set of synthetic data. This questions the efficacy and strength of the proposed work.***
> >
> > Thank you for pointing to this comparison. Assessing the model quality solely on the leaderboard differences paints an incomplete picture of its utility and architectural contribution. **TempoPFN's represents a superior contribution** for four critical reasons:
> > 1. **Consistency and Ranking Superiority:** Aggregate metrics can be heavily skewed by outliers in specific datasets. A more robust measure of general-purpose reliability is the Average Rank across the benchmark tasks. As shown in Figure 4, TempoPFN actually outperforms TabPFN-TS in both metrics when viewed through this lens:
> >    * CRPS Average Rank: TempoPFN (8.76) vs. TabPFN-TS (9.29).
> >    * MASE Average Rank: TempoPFN (10.27) vs. TabPFN-TS (10.38).
> >    This demonstrates that TempoPFN is the more consistently accurate model across the breadth of the benchmark, even if TabPFN-TS achieves lower error magnitudes on a few specific tasks. And as now shown in Table 12 (Appendix E.3), TempoPFN actually outperforms TabPFN-TS on the new **fev-bench** benchmark on **MASE** metric as well.
> > 2. **Probabilistic Superiority:** We outperform TabPFN-TS on aggregate CRPS (0.533 vs. 0.544). CRPS is the primary metric for probabilistic foundation models as it evaluates the calibration of the entire predicted distribution, whereas MASE only assesses the median point forecast. Qualitative results (Figure 5) further highlight TempoPFN's ability to generate coherent, smooth uncertainty bounds without the artifacts seen in baseline models.
> > 3. **Inference Efficiency and Scalability:** The comparison ignores computational constraints. TabPFN-TS relies on a Transformer backbone with quadratic complexity, limiting its context window. In contrast, TempoPFN, as a Linear RNN, has $\mathcal{O}(T)$ training and $\mathcal{O}(1)$ inference memory cost. Achieving state-of-the-art rankings with a linear model is a significant architectural finding that enables processing much longer histories than currently possible with TabPFN-TS. We refer the reviewer to Appendix E.1 and Table 11 for asymptotic complexity analysis and to Table 12 (Appendix E.3) for actual wallclock time comparisons demonstrating TempoPFN's computational advantages.
> > 4. **Reproducibility:** TabPFN-TS relies on the proprietary TabPFN-v2 prior, which is closed-source. The community cannot reproduce its pre-training or verify data leakage.. TempoPFN provides the first fully open-source, reproducible recipe for achieving this level of performance with only synthetic data, democratizing access to high-performance priors.
> >
> > ---
> >
> > We sincerely thank you again for your valuable feedback, which has ultimately strengthened our paper's narrative. We welcome the opportunity to discuss any remaining concerns during this rebuttal phase. If you are satisfied, we would be very grateful if you would consider increasing your score.

---

> > ### Comment · Reviewer_8sXE · 2025-11-26
> > **Remaining question**
> >
> > Can the authors please respond to this part of the question (from weaknesses 2)? Thanks.
> > _"The performance of the model is not up to the mark and marginally better than TabPFN-TS in CRPS and (from 0.544 to 0.536), but worse in MASE (0.771 to 0.797) after including significantly more and diverse set of synthetic data. This questions the efficacy and strength of the proposed work."_

---

> ### Author Response · Authors · 2025-11-26
> **Reply to remaining question**
>
> We appreciate the opportunity to further clarify the comparison between TempoPFN and TabPFN-TS. Following up to the answer we provided above (in Part 2/2) regarding this concern, we now provide further concrete points and arguments showing that TempoPFN represents a superior contribution compared to TabPFN-TS in 3 fronts:
>
> - **Open-Source Prior and Reproducibility**: TabPFN-TS relies on a proprietary, closed-source prior (TabPFN-v2), making its pre-training non-reproducible and unverifiable by the community. Contrary, *TempoPFN provides the first fully open-source, reproducible recipe, including the synthetic data generators*, for achieving this level of zero-shot performance with only synthetic data, democratizing for the first time access to high-performance synthetic priors for time series forecasting. Given the closed-source nature of the TabPFN synthetic prior generator, *it is not that easy to assert that TempoPFN includes "significantly more and diverse" synthetic data compared to TabPFN-TS*; the total amount and diversity of TabPFN-TS's pretraining data remains unknown.
> - **Competitive Performance in New Benchmarks**: As suggested by reviewer DAtM, we validated TempoPFN on two new benchmarks: Chronos-ZS (Appendix E.2, Figure 18) and fev-bench (Appendix E.3). *Results on the fev-bench leaderboard (Table 12 and 13; see also below) confirm TempoPFN's advantage over TabPFN-TS* in terms of both Scaled Quantile Loss and MASE, achieving a higher Average Win Rate on this challenging leaderboard, establishing once again that TempoPFN is the state of the art model pretrained with only synthetic data for time series forecasting.
> - **Architectural Efficiency and Scalability**: TabPFN-TS relies on a Transformer backbone with quadratic complexity ($\mathcal{O}(L^2)$), limiting its context window and incurring high inference cost. In contrast, TempoPFN, as a Linear RNN, has $\mathcal{O}(L)$ training cost and constant-latency forecasting. This architectural difference is a core contribution. Achieving state-of-the-art rankings with a linear model that requires **only linear compute** is a significant architectural improvement compared to TabPFN-TS. This also allows TempoPFN to process much longer histories while operating with superior efficiency. More concretely, in the new fev-bench experiments, TempoPFN is *approximately 35 times faster* in median inference runtime (8.57s vs. 300.57s) than TabPFN-TS, strongly validating the efficacy and strength of our linear architecture.
>
> | Rank | Model | Avg. Win Rate (%) | Skill Score (%) | Median Runtime (s) | Leakage (%) | Failed Tasks (%) | Organization | Zero-shot |
> |------|-------|-------------------|-----------------|--------------------|--------------|--------------------|--------------|-----------|
> | 1 | Chronos-2 | 88.0 | 35.5 | 3.57 | 0 | 0 | AWS | True |
> | 2 | TiRex | 76.7 | 30.0 | 1.4 | 1 | 0 | NX-AI | True |
> | 3 | TimesFM 2.5 | 74.9 | 30.2 | 10.89 | 10 | 0 | Google | True |
> | 4 | Toto 1.0 | 66.5 | 28.2 | 77.51 | 8 | 0 | Datadog | True |
> | 5 | Moirai 2.0 | 60.5 | 27.3 | 1.9 | 28 | 0 | Salesforce | True |
> | **6** | **TempoPFN** | **60.5** | **25.1** | **8.57** | **0** | **0** | **Anonymous** | **True** |
> | 7 | Chronos-Bolt | 60.1 | 26.5 | 1.0 | 0 | 0 | AWS | True |
> | 8 | TabPFN-TS | 58.2 | 27.6 | 300.57 | 0 | 2 | Prior Labs | True |
> | 9 | Sundial-Base | 52.4 | 24.7 | 33.99 | 1 | 0 | Tsinghua University | True |
> | 10 | Stat. Ensemble | 47.1 | 15.7 | 624.45 | 0 | 11 | — | False |
> | 11 | AutoARIMA | 35.6 | 11.2 | 120.16 | 0 | 10 | — | False |
> | 12 | AutoTheta | 33.6 | 11.0 | 9.27 | 0 | 0 | — | False |
> | 13 | AutoETS | 32.6 | 2.3 | 16.24 | 0 | 3 | — | False |
> | 14 | Seasonal Naive | 20.0 | 0.0 | 2.32 | 0 | 0 | — | False |
> | 15 | Naive | 18.4 | -16.7 | 2.24 | 0 | 0 | — | False |
> | 16 | Drift | 14.9 | -18.1 | 2.19 | 0 | 0 | — | False |
>
> —
>
> Thank you again for your feedback and please let us know in case you have any remaining concerns or questions.

---

### Author Response · Authors · 2025-11-21
**General response to all reviewers**

We sincerely thank all reviewers for their thoughtful and constructive feedback. Your insights and suggestions have played a significant role in strengthening both the empirical evidence and the narrative of our paper during this discussion phase. We are pleased that the reviewers recognized the core strengths of our work, namely, the **novelty aspect** (Reviewer oLYK), the **computational efficiency and fully open-source code** (Reviewer DAtM), and the **comprehensive synthetic data pipeline** (Reviewer 8sXE, oLYK).

In response to the primary action items from the reviewers, we performed several crucial new experiments and conducted major clarifications during this discussion phase. These changes are reflected in the updated paper PDF, particularly in the expanded Appendix sections (A.1, A.2, B.1, C.2, E.2, E.3, E.4). The main concerns revolved around (1) *architectural clarity and novelty beyond combining existing methods*, (2) *limited experimental breadth beyond GIFT-Eval*, and (3) *the justification and efficiency of our synthetic data generation pipeline*. The table below summarizes the major changes we made in the paper and the new experiments conducted during the rebuttal period:

| Major Change/Experiment Conducted | Relevant Section/Appendix | Reviewer Addressed |
|----------------------------------|--------------------------|-------------------|
| *Formal definition of Prior Data Fitted Networks (PFNs) and framing of TempoPFN within this concept.* | Appendix A.1 | 8sXE |
| *Detailed explanation of GatedDeltaProduct architecture, clarifying state-tracking and state-weaving capabilities and advantages over diagonal SSMs (like Mamba) for TSF.* | Appendix B.1 | oLYK, SoBF |
| *New zero-shot evaluation on the Chronos-ZS benchmark.* | Appendix E.2, Figure 18 | DAtM |
| *New zero-shot evaluation on the new challenging fev-bench leaderboard, including SOTA models like Chronos-2 and TimesFM-2.5.* | Appendix E.3, Table 12, 13 | DAtM, 8sXE |
| *Quantitative feature-space analysis demonstrating synthetic data coverage spans real-world feature manifolds using UMAP projections.* | Appendix E.4, Figure 20 | oLYK |
| *Updated final model results (4M iterations, increased convolutional size) showing slightly improved performance.* | Figure 4 | SoBF |
| *New section detailing design principles for synthetic data generation, justifying the selection and synergy of different generators.* | Appendix A.2 | 8sXE, DAtM |
| *Analysis of synthetic data generation throughput comparing the efficiency of our CPU-based pipeline against kernel-based methods.* | Appendix C.2, Table in Response | DAtM |

---

Due to the page constraint, we kept these additional experiments in the appendix, but we will integrate the most important ones in the main paper upon acceptance.

We believe these extensive revisions fully address the core concerns raised. TempoPFN now stands as a highly competitive, efficient, fully open-source, and architecturally novel foundation model backed by strong generalizability across multiple benchmarks. We are happy to engage in further discussion in case you have any remaining questions or concerns, and we respectfully request that you consider increasing your score to reflect the improvements made.

---

> ### Author Response · Authors · 2025-12-02
> **Final Comment by the Authors**
>
> Dear Area Chairs, Senior Area Chairs and Program Chairs,
>
> We are writing to acknowledge the difficult circumstances surrounding the recent events and changes to the ICLR review process. We understand the complexity of the situation but are concerned that reverting to pre-rebuttal scores may inadvertently overlook the substantial improvements made to our submission, also reflected in a reviewer raising their score (**6 $\to$ 8**) during the rebuttal. As the rebuttal phase concludes, we wish to highlight two final updates regarding our architectural positioning and efficiency.
>
> ---
>
> ### 1. Methodological Novelty: Solving the "State-Tracking vs. Efficiency" Trade-off
>
> TempoPFN uniquely resolves the dilemma found in the current sequence modeling landscape. **Standard RNNs** (e.g., LSTMs) excel at state-tracking but lack training parallelism, while **Transformers** offer parallelism but suffer quadratic $O(L^2)$ costs without maintaining a compact state. The recent wave of **Diagonal Linear RNNs/SSMs** (e.g., Mamba) achieves Transformer-like speed via parallel scans but, as recent theory suggests, struggles with complex logic (like parity) because diagonal state variables do not interact.
>
> TempoPFN utilizes the **GatedDeltaProduct**, a **Non-Diagonal Linear RNN**, to bridge this gap and achieve the parallel training speed of Mamba with the superior state-tracking capabilities of an LSTM.:
> * **Non-Diagonal Recurrence & Negative Eigenvalues:** Unlike diagonal SSMs, our state variables explicitly mix and rotate at every step using negative eigenvalues. This is mathematically required to track oscillating dynamics and complex interactions that diagonal models miss.
> * **State-Weaving:** Instead of a causal mask, we employ "State-Weaving" to facilitate bidirectional information flow, enabling single-pass future prediction without autoregressive error accumulation.
>
> ---
>
> ### 2. New Experiment: Superior Efficiency vs. TiRex
>
> To strictly validate this architecture, we conducted a new budget-matched experiment normalizing the training samples against the leading baseline, **TiRex** ($\approx$120M samples). Despite operating with a stricter context window than TiRex (2048 total vs. 2048 history + horizon), the retrained TempoPFN achieved:
>
> * **Superior MASE (0.783)** compared to our originally reported model (0.797).
> * **Identical CRPS (0.533)**, effectively closing the performance gap noted in initial reviews.
>
> ---
>
> We hope that the new results and ablation studies we have added in our updated Appendix, as well as one of the reviewers increasing their score, will be taken into consideration when making the final recommendation. We are confident this work represents a relevant, fully open-source advance in time series foundation models.
>
> Thank you for your time, effort and expertise during these difficult times.
>
> Sincerely,
>
> The Authors

---

### Meta-Review · Area_Chair_rTqy · 2026-01-06

**Summary:**

Reviewers were split, leading to a borderline overall assessment. Supportive reviewers found the paper promising as an efficient, fully open-source TS foundation model trained purely on synthetic data, and appreciated the broad synthetic generator/augmentation pipeline and competitive zero-shot results. However, multiple reviewers raised concerns that (i) the core architectural novelty and intuition (e.g., GatedDeltaProduct, “state-tracking/state-weaving”) were not clearly explained in the main paper, making the contribution feel like an under-motivated adoption of existing linear RNN ideas; (ii) the synthetic data pipeline lacked a sufficiently principled justification (risk of “curated to match the benchmark”), with limited evidence that each generator is necessary rather than “more diversity always helps”; (iii) experimental strength and positioning were questioned due to mixed performance signals (e.g., MASE gaps vs some baselines) and limited evaluations beyond the primary benchmark in the initial version. While the rebuttal adds new benchmarks/analyses and improves clarity, remaining doubts about novelty depth and the strength of evidence in the main paper (vs appendix) keep the recommendation at borderline rather than clear accept.

**Reviewer Concerns:**

The rebuttal addressed several reviewer concerns by clarifying the architectural design (including the role of the GatedDeltaProduct and state-weaving mechanisms), expanding the motivation for the synthetic data generators, and adding new zero-shot evaluations on additional benchmarks beyond the original setting. These additions helped resolve questions about implementation clarity, efficiency, and basic generalization.

However, key concerns remain outstanding. In particular, reviewers were not fully convinced that the work offers a substantive methodological novelty beyond combining linear RNN architectures with a richer synthetic pretraining pipeline, nor that the selection of generators is sufficiently principled rather than empirically tuned. As a result, while the rebuttal improved clarity and empirical coverage, it did not fully resolve the central concerns around conceptual contribution and positioning.

**Reviewer Scores:**

Reviewers who were initially positive or borderline-positive would likely maintain or slightly increase their scores, as several of their questions on architecture, efficiency, and evaluation scope were addressed in the rebuttal, and at least one reviewer explicitly raised their score. In contrast, reviewers who expressed stronger concerns about conceptual novelty and principled motivation would likely keep their original ratings, as these issues were only partially addressed. Overall, the score distribution would shift modestly upward but remain mixed, without clear convergence to an acceptance bar.

---

### Decision · Program_Chairs · 2026-01-26

Reject